# A *C. elegans* model of *C9orf72*-associated ALS/FTD uncovers a conserved role for eIF2D in RAN translation

Yoshifumi Sonobe [1,2,3], Jihad Aburas [1,3,4], Gopinath Krishnan[5], Andrew C. Fleming [6], Ghanashyam Ghadge[1,2,3], Priota Islam[7,8], Eleanor C. Warren[7,8], Yuanzheng Gu [9], Mark W. Kankel [9], André E. X. Brown[7,8], Evangelos Kiskinis[6], Tania F. Gendron [10], Fen-Biao Gao [5], Raymond P. Roos [1,2,3,11✉] & Paschalis Kratsios [1,3,4,11✉]

A hexanucleotide repeat expansion GGGGCC in the non-coding region of *C9orf72* is the most common cause of inherited amyotrophic lateral sclerosis (ALS) and frontotemporal dementia (FTD). Toxic dipeptide repeats (DPRs) are synthesized from GGGGCC via repeat-associated non-AUG (RAN) translation. Here, we develop *C. elegans* models that express, either ubiquitously or exclusively in neurons, 75 GGGGCC repeats flanked by intronic *C9orf72* sequence. The worms generate DPRs (poly-glycine-alanine [poly-GA], poly-glycine-proline [poly-GP]) and poly-glycine-arginine [poly-GR]), display neurodegeneration, and exhibit locomotor and lifespan defects. Mutation of a non-canonical translation-initiating codon (CUG) upstream of the repeats selectively reduces poly-GA steady-state levels and ameliorates disease, suggesting poly-GA is pathogenic. Importantly, loss-of-function mutations in the eukaryotic translation initiation factor 2D (*eif-2D/eIF2D*) reduce poly-GA and poly-GP levels, and increase lifespan in both *C. elegans* models. Our in vitro studies in mammalian cells yield similar results. Here, we show a conserved role for *eif-2D/eIF2D* in DPR expression.

[1] University of Chicago Medical Center, 5841S. Maryland Avenue, Chicago, IL 60637, USA. [2] Department of Neurology, University of Chicago Medical Center, 5841S. Maryland Avenue, Chicago, IL 60637, USA. [3] The Grossman Institute for Neuroscience, Quantitative Biology, and Human Behavior, University of Chicago, Chicago, IL, USA. [4] Department of Neurobiology, University of Chicago, Chicago, IL, USA. [5] Department of Neurology, University of Massachusetts Medical School, Worcester, MA 01605, USA. [6] The Ken & Ruth Davee Department of Neurology, Feinberg School of Medicine, Northwestern University, Chicago, USA. [7] MRC London Institute of Medical Sciences, London, UK. [8] Institute of Clinical Sciences, Imperial College London, London, UK. [9] Neuromuscular & Movement Disorders, Biogen, Cambridge, MA 02142, USA. [10] Department of Neuroscience, Mayo Clinic, Jacksonville, FL, USA. [11] These authors contributed equally: Raymond P. Roos, Paschalis Kratsios. ✉email: rroos@neurology.bsd.uchicago.edu; pkratsios@uchicago.edu

The GGGGCC (G4C2) hexanucleotide repeat expansion in the non-coding region of *C9orf72* is the most common monogenic cause of inherited amyotrophic lateral sclerosis (ALS) and frontotemporal dementia (FTD)[1,2], and also causes up to 10% of what appears to be sporadic ALS. The G4C2 repeat expands in patients to hundreds or thousands of copies that vary in number in different cells of the same individual. This genetic insult is thought to cause ALS/FTD via three non-mutually exclusive mechanisms: (1) loss of function due to decreased expression of C9ORF72 protein, (2) RNA toxicity from bidirectionally transcribed sense (GGGGCC) and antisense (CCCCGG) transcripts, and (3) proteotoxicity from dipeptide repeat (DPR) proteins produced from the expanded nucleotide repeats[3].

This study focuses on the molecular mechanisms underlying DPR toxicity. Strong evidence suggests that DPRs are toxic in both cell culture and animal models of disease[4,5]. Despite the presence of the expanded G4C2 repeat in the non-coding region of the RNA and the absence of an AUG initiating codon, DPRs are translated in all three reading frames from both sense and antisense transcripts through a process called repeat-associated non-AUG (RAN) translation[4]. Poly-glycine-alanine (poly-GA), poly-glycine-proline (poly-GP), and poly-glycine-arginine (poly-GR) are produced from sense transcripts, whereas poly-proline-arginine (poly-PR), poly-proline-alanine (poly-PA), and poly-GP are generated from antisense transcripts[6–8]. DPRs are present in neural cells of patients with *C9orf72*-associated ALS and FTD, indicating that RAN translation occurs in vivo[6,7,9,10]. A number of potential causes for DPR toxicity have been proposed, including protein aggregation, impaired nucleocytoplasmic transport, and proteasome dysfunction[5]; however, the relative contribution of each DPR to disease pathogenesis remains unclear.

Recent in vitro and in vivo studies have identified non-canonical translation initiation factors as critical mediators of DPR production. For example, a recent study showed that the ribosomal protein eS25 (RPS25), which is required for efficient translation initiation at the internal ribosome entry site (IRES) of a number of viral and cellular RNAs[11,12], is also required for RAN translation of poly-GP from G4C2 repeats in yeast, fly, and induced pluripotent human stem cell models[13]. In addition, the eukaryotic translation initiation factor eIF3F, which is thought to control translation of hepatitis C viral (HCV) RNA by binding to the viral IRES[14], regulates poly-GP production from G4C2 repeats in HEK293 cells[15]. Moreover, a fly study found that eIF4B and eIF4H are important for RAN translation of poly-GR from G4C2 transcripts[16]. Lastly, knockdown of eIF2A, an unconventional translation initiation factor used by cells under stress conditions[17–19], partially affected poly-GA production in HEK293 cells and neural cells of the chick embryo[20]. Collectively, these and other studies[21,22] suggest that multiple factors are involved in RAN translation of DPRs from G4C2 transcripts; yet, their identity and molecular mechanisms of action remain elusive.

To identify factors involved in RAN translation of DPRs in vivo, we developed two *C. elegans* models for *C9orf72*-associated ALS/FTD. *C. elegans* provides a powerful system for the study of molecular and cellular mechanisms underlying neurodegenerative disorders due to: (1) its short lifespan (~3–4 weeks), (2) its compact nervous system (302 neurons in total), (3) the ability to study neuronal morphology and function with single-cell resolution, (4) the fact that most *C. elegans* genes have human orthologs, and (5) the ease of genomic engineering and transgenesis, which enables the rapid generation of worms that harbor human gene mutations, permitting in vivo modeling of neurodegenerative disorders.

In this study, we initially developed *C. elegans* transgenic animals that carry, under the control of a ubiquitous promoter, 75 copies of the G4C2 repeat flanked by intronic *C9orf72* sequences. Compared to controls, these animals produce DPRs (poly-GA, poly-GP, poly-GR), and display evidence of neurodegeneration, as well as locomotor and lifespan defects. A second set of transgenic worms expressing the 75 G4C2 repeats and the flanking intronic sequences exclusively in neurons displayed similar phenotypes, suggesting that DPR production in this cell type is sufficient to cause disease phenotypes. These *C. elegans* models provide an opportunity to study in vivo the molecular mechanisms underlying DPR production. Through a candidate approach, we identified a role for the non-canonical translation initiation factor *eif-2D* (ortholog of human *eIF2D*) in RAN translation of G4C2 repeats. Genetic removal of *eif-2D* did not affect the formation of G4C2 RNA foci, but prominently decreased poly-GA steady-state levels, mildly affected poly-GP levels, and improved locomotor activity and lifespan in both *C. elegans* models. Supporting the phylogenetic conservation of our *C. elegans* findings, in vitro knockdown of eIF2D in mouse and human cells led to a similar decrease in poly-GA steady-state levels. Together, these observations suggest eIF2D may play an important role in RAN translation from G4C2 repeats, a finding with biomedical implications for *C9orf72*-associated ALS and FTD.

## Results

**Generation of a *C. elegans* model for *C9orf72*-mediated ALS/FTD.** The *C. elegans* genome contains a *C9orf72* ortholog (*alfa-1*), which is involved in lysosomal homeostasis, cellular metabolism, and stress-induced neurodegeneration[23–25], however, *alfa-1* does not contain G4C2 repeats. Therefore, to study the molecular mechanisms underlying DPR production from G4C2 repeats, we generated worms that carry a transgene encoding 75 copies of the G4C2 sequence under the control of a ubiquitous (*snb-1*) promoter[26] (Fig. 1a). The widespread expression of this promoter was independently validated (Supplementary Fig. 1). These 75 G4C2 copies are flanked with intronic sequences normally found upstream (113 nucleotides) and downstream (99 nucleotides) of the G4C2 repeats in the human *C9orf72* gene (Fig. 1a). To monitor the expression of poly-GA, the most amyloidogenic DPR[27,28], a nanoluciferase (nLuc) reporter was placed in the poly-GA reading frame. Hereafter, we will refer to these transgenic animals as C9 *ubi* (Fig. 1a). In parallel, we generated four control *C. elegans* strains: (a) UAG *ubi* worms carry an identical sequence to the C9 *ubi* animals, but the upstream translation initiation codon CUG, which is required for translation of poly-GA in vitro[20,29–31], is mutated to UAG (Fig. 1a), (b) ΔC9 *ubi* worms lack the G4C2 repeats and the intronic sequences flanking the G4C2 repeats (Fig. 1a), (c) ΔG4C2 repeats *ubi* worms lack the G4C2 repeats, but maintain the intronic sequences (Supplementary Fig. 2a), and (d) the C9 ΔnLuc *ubi* worms carry an identical sequence to the C9 *ubi* animals, but lack nLuc (Supplementary Fig. 2a).

Several independent transgenic lines for C9 *ubi*, UAG *ubi*, ΔC9 *ubi*, ΔG4C2 repeats *ubi*, and C9 ΔnLuc *ubi* worms were generated and tested as follows. First, we performed quantitative PCR with reverse transcription (RT-qPCR) and found that mRNA corresponding to intronic sequences that flank the G4C2 repeats is only detected in C9 *ubi*, UAG *ubi*, ΔG4C2 repeats *ubi*, and C9 ΔnLuc *ubi* animals, whereas *nLuc* mRNA is only detected in C9 *ubi*, UAG *ubi*, and ΔG4C2 repeats *ubi* animals (Fig. 1b and Supplementary Fig. 2b, c). Second, we sought to evaluate whether G4C2 transcripts form RNA foci in our transgenic animals, similar to observations made in other *C9orf72* models[32,33]. RNA fluorescent in situ hybridization (FISH) revealed G4C2 RNA foci in head and tail neurons, ventral nerve cord motor neurons, as well as non-neuronal cells in C9 *ubi* and UAG *ubi*, but not in control (ΔC9 *ubi*) animals (Fig. 1c and Supplementary Fig. 3a).

Next, we tested whether DPRs (poly-GA, poly-GP, poly-GR) from the G4C2 sense transcript are produced in the C9 *ubi* animals. We focused on these because the 75 G4C2 repeats are

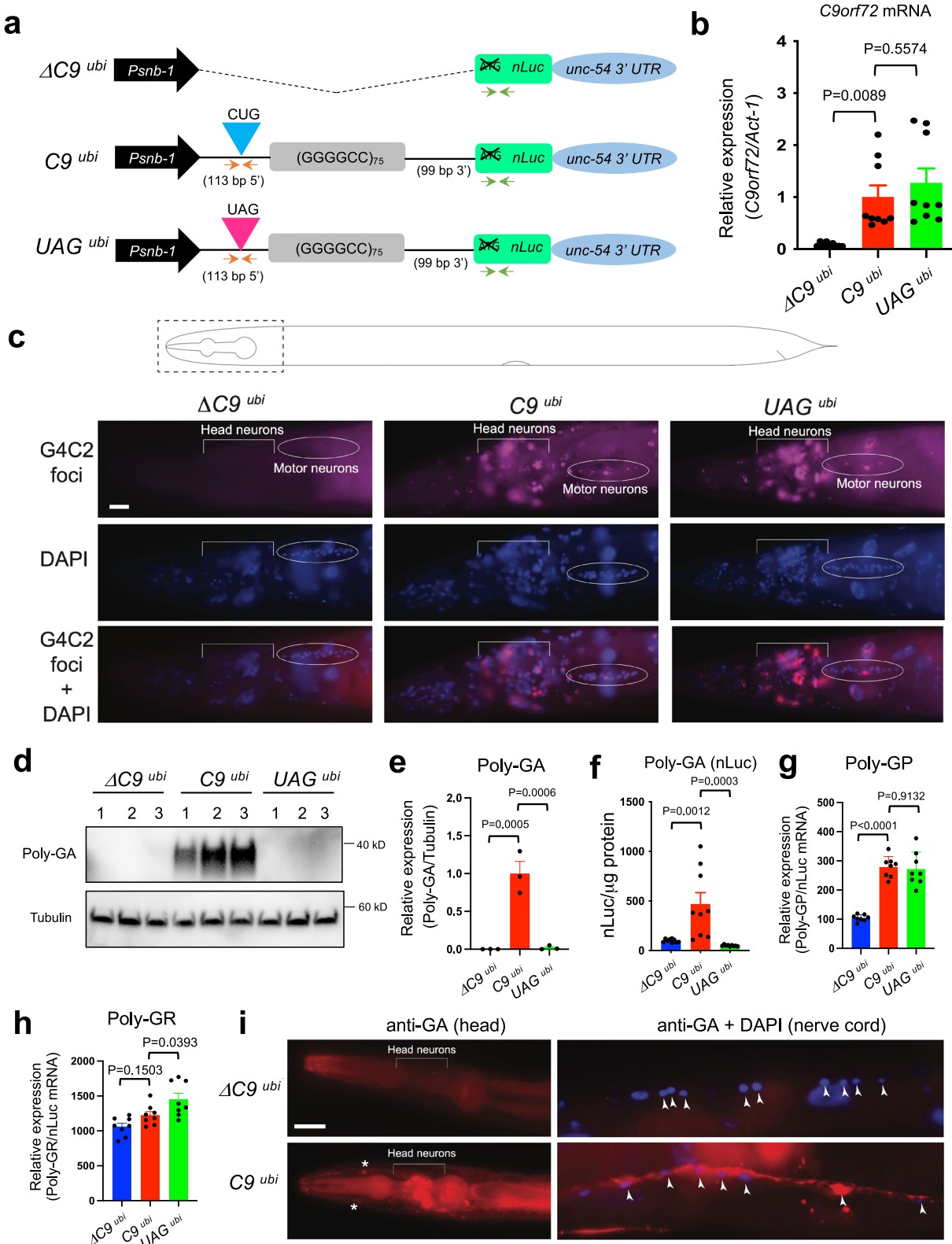

downstream of the *snb-1* promoter, suggesting the sense transcript may be the one primarily transcribed and translated. Western blotting and luciferase assays showed that poly-GA was robustly expressed in *C9 ubi* worms when compared to *ΔC9 ubi*, *UAG ubi*, and *ΔG4C2 repeats ubi* control animals (Fig. 1d–f and Supplementary Fig. 3b). Corroborating these observations, immunofluorescence staining revealed the presence of poly-GA

aggregates in head neurons, ventral nerve cord motor neurons, and non-neuronal cells of *C9 ubi* animals (Fig. 1i). We made similar observations in *C9 ΔnLuc ubi* worms (Supplementary Fig. 3c).

To evaluate potential age-dependent aggregation of poly-GA, we took advantage of the optical transparency of *C. elegans* and generated a GFP version of the *C9 ubi* animals (termed *C9-GFP ubi*)

**Fig. 1 A C. elegans model for C9orf72-mediated ALS/FTD. a** Schematic diagram showing $\Delta C9^{ubi}$, $C9^{ubi}$, and $UAG^{ubi}$ constructs with the snb-1 promoter. The orange and green arrows show primer location to detect C9orf72 and nLuc mRNAs, respectively. nLuc: nanoluciferase; UTR: untranslated region. **b** The G4C2 repeat mRNA (C9orf72 intronic mRNA) and act-1 mRNA were assessed by RT-PCR on three different transgenic lines per genotype: $\Delta C9^{ubi}$ (kasEx154, kasEx259, kasEx260), $C9^{ubi}$ (kasEx153, kasEx261, kasEx262), or $UAG^{ubi}$ (kasEx155, kasEx263, kasEx264) (mean ± s.e.m.). One-way ANOVA with Dunnett's multiple comparisons test was performed. **c** Representative fluorescent images of the head of adult (day 1) $\Delta C9^{ubi}$ (kasIs9), $C9^{ubi}$ (kasIs7) and $UAG^{ubi}$ (kasIs10) animals. $N = 25$. G4C2 RNA foci are visualized with a Quasar 670 probe. Nuclei are demarcated with DAPI (blue). Scale bar = 10 μm. G4C2 RNA foci are detected in head neurons and motor neurons of the retrovesicular ganglion (circled) both in $C9^{ubi}$ and $UAG^{ubi}$ animals. The signal detected on the left of the region indicated "Head neurons" belongs to non-neuronal cells (no neurons reside that far anterior in the C. elegans head). **d, e** Western blotting with poly-GA and α-tubulin antibodies and quantification on three different transgenic lines per genotype: $\Delta C9^{ubi}$ (kasEx154, kasEx259, kasEx260), $C9^{ubi}$ (kasEx153, kasEx261, kasEx262), or $UAG^{ubi}$ (kasEx155, kasEx263, kasEx264) (mean ± s.e.m.). One-way ANOVA with Dunnett's multiple comparisons test was performed. **f** Luciferase assay was performed in worm lysates from the same three transgenic lines per genotype shown in panel e (mean ± s.e.m.). **g, h** ELISA for poly-GP and poly-GR in $\Delta C9^{ubi}$ (kasIs8), $C9^{ubi}$ (kasIs7), and $UAG^{ubi}$ (kasIs10) animals ($n = 4$, mean ± s.e.m.). One-way ANOVA with Dunnett's multiple comparisons test was performed. **i** Immunofluorescent staining for poly-GA (red) in $\Delta C9^{ubi}$ (kasIs9) and $C9^{ubi}$ (kasIs10) adult (day 1) worms. Representative images of the head and ventral nerve cord region taken at ×40 magnification. $N = 20$. Anterior to the left. Motor neuron nuclei in blue (DAPI) shown with arrowheads. Scale bar = 20 μm.

by replacing nLuc with GFP. Similar to mouse C9orf72 models[33,34], we indeed observed an age-dependent poly-GA::GFP aggregation in neurons and non-neuronal cells of $C9\text{-}GFP^{ubi}$ animals (Supplementary Fig. 4).

Although we did not detect poly-GP or poly-GR with western blotting (Supplementary Fig. 5a), more sensitive ELISA assays detected robust expression of poly-GP and modest production of poly-GR in $C9^{ubi}$ worms, above the background level, observed in $\Delta C9^{ubi}$ controls (Fig. 1g, h). The background levels of poly-GP and poly-GR likely reflect the abundance of GP and GR motifs in the C. elegans proteome. Collectively, the above results demonstrate that RAN translation occurs in C. elegans.

Lastly, our findings in $UAG^{ubi}$ animals show that mutation of the non-canonical initiation codon CUG did not affect C9orf72 mRNA levels nor the formation of G4C2 RNA foci (Fig. 1c), but dramatically reduced poly-GA production (Fig. 1d–f). In contrast, the levels of poly-GP and poly-GR were not reduced in $UAG^{ubi}$ animals (Fig. 1g, h). We also confirmed this finding in HEK293 cells in vitro (Supplementary Fig. 5b, c), consistent with a previous report in human induced pluripotent stem cell (iPSC)-derived motor neurons[35]. Interestingly, the nLuc reporter gene is translated in $\Delta G4C2$ repeats $^{ubi}$ animals, which lack G4C2 repeats but carry the intronic C9orf72 sequences including the upstream CUG (Supplementary Fig. 6). However, it remains unclear whether this CUG can initiate translation in the absence of G4C2 repeats.

Importantly, we found that the median survival of $C9^{ubi}$ worms is significantly shorter when compared to $\Delta C9^{ubi}$ and $UAG^{ubi}$ controls ($P < 0.0001$) (Fig. 2a and Supplementary Fig. 7a–c). The lack of an effect on median survival of $UAG^{ubi}$ animals suggests that the G4C2 RNA foci present in these animals do not affect organismal survival. Together, these findings suggest that one or more of the three DPRs (poly-GA, poly-GP, poly-GR) generated from the 75 G4C2 repeats in the $C9^{ubi}$ animals likely contribute to the reduced survival of these animals, thereby establishing a C. elegans model for C9orf72-mediated ALS/FTD.

**$C9^{ubi}$ animals display progressive locomotion defects and motor neuron degeneration.** At the level of animal behavior, we conducted swimming and crawling assays on $C9^{ubi}$ worms. The swimming assay was performed in a longitudinal fashion at juvenile (larval stage 4, L4) and adult (days 2 and 5) stages. By quantifying the rate of body flexion while animals are swimming, we found an age-dependent defect in $C9^{ubi}$ animals compared to three control strains ($\Delta C9^{ubi}$, $UAG^{ubi}$, $\Delta G4C2$ repeats $^{ubi}$) (Fig. 2b and Supplementary Fig. 7d). Next, we performed a high-resolution behavioral analysis of freely crawling adult (day 2) C. elegans using automated worm tracking technology[36,37].

This analysis can quantitate several features related to locomotion (e.g., velocity, crawling amplitude, body curvature). We found multiple locomotor defects in $C9^{ubi}$ animals, such as reduced velocity and impaired body curvature (Supplementary Figs. 8–10). These defects were not present in $\Delta C9^{ubi}$ and $UAG^{ubi}$, and $\Delta G4C2$ repeats $^{ubi}$ controls, indicating that a hallmark of ALS, namely motor deficits, is recapitulated in the $C9^{ubi}$ animal model. Crawling defects were also present in older (adult day 6) $C9^{ubi}$ animals, but not in $\Delta C9^{ubi}$ and $UAG^{ubi}$ controls (Supplementary Videos 1–3). The presence of locomotion and survival defects in $C9^{ubi}$, but not $UAG^{ubi}$ animals is unlikely to be a result of different G4C2 mRNA amounts since both showed comparable mRNA levels (Fig. 1b).

The locomotion defects could be attributed, at least partially, to the degeneration of ventral nerve cord motor neurons, which are essential for C. elegans locomotion. To test this possibility, we fluorescently labeled the cholinergic motor neurons present in the nerve cord of $C9^{ubi}$ and $\Delta C9^{ubi}$ animals with the unc-17/VAChT::gfp transgenic reporter, which allows for single-cell resolution in our analysis (Fig. 2c, d). Next, we quantified the number of cholinergic motor neurons (unc-17/VAChT::gfp positive) in $C9^{ubi}$ and $\Delta C9^{ubi}$ animals at juvenile (L4) and adult (days 2 and 5) stages. Although no differences were observed at L4 and day 2, we found a significant reduction in the number of motor neurons that populate the nerve cord of $C9^{ubi}$ animals at day 5 (Fig. 2e), arguing for progressive motor neuron degeneration. Because the snb-1 promoter is broadly active, we questioned whether other neuron types also degenerate. Although a similar longitudinal analysis did not reveal dopaminergic neuron degeneration (Supplementary Fig. 11), we cannot exclude the possibility that other neuron types in addition to cholinergic neurons also degenerate in $C9^{ubi}$ animals.

**DPR production selectively in neurons is sufficient to cause disease phenotypes in vivo.** We found that poly-GA is present both in neuronal and non-neuronal cells of $C9^{ubi}$ animals because the snb-1 promoter is active in multiple cell types (Fig. 1i and Supplementary Fig. 4a, b). To test whether DPR production from the G4C2 repeats exclusively in neurons is sufficient to cause disease-associated phenotypes, we generated a second set of transgenic animals by replacing the snb-1 promoter with a fragment of the unc-11 promoter (unc-11 prom8) known to be exclusively active in all C. elegans neurons[26]. We refer to these animals as $C9^{neuro}$, $\Delta C9^{neuro}$, and $UAG^{neuro}$ (Fig. 3a). First, we confirmed by RT-qPCR that the G4C2 repeat RNA (with C9orf72 intronic mRNA) and the nLuc mRNA are transcribed in $C9^{neuro}$ and $UAG^{neuro}$ animals at comparable levels (Fig. 3b, c).

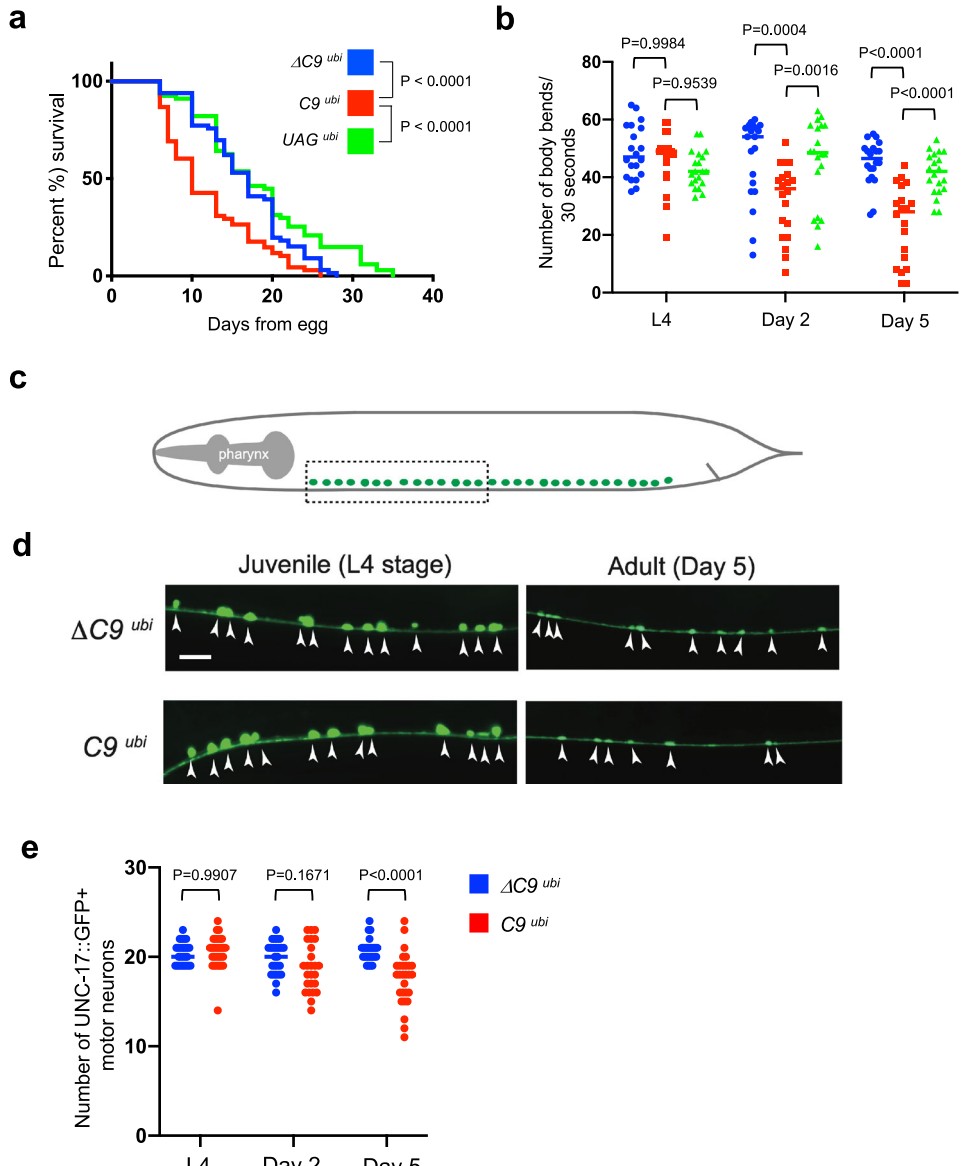

**Fig. 2 Evidence for motor neuron degeneration and synaptic defects in C9 ubi animals. a** Lifespan of ΔC9 ubi (kasEx154, kasEx259, kasEx260), C9 ubi (kasEx153, kasEx261, kasEx262), and UAG ubi (kasEx155, kasEx263, kasEx264) animals. Three different transgenic lines per genotype were used and the results are combined per genotype. ΔC9 ubi, n = 66; C9 ubi, n = 68; UAG ubi, n = 67). See Supplementary Fig. 7. Mantel–Cox log-rank test was performed.
**b** Swimming rates of ΔC9 ubi(kasIs9-blue), C9 ubi(kasIs7-red), and UAG ubi (kasIs10-green) worms at three different stages. n = 20. Two-way ANOVA with Šídák's multiple comparison test was performed. **c** Schematic showing cholinergic motor neurons (green) in the C. elegans ventral nerve cord. Dashed box indicates the region used for imaging and quantification. **d**, **e** Motor neuron cell bodies are visualized with the vsIs48 [unc-17/VAChT::gfp] marker in ΔC9 ubi (KasIs9) and C9 ubi (KasIs7) animals at the indicated stages. Arrowheads indicate motor neuron cell bodies. Scale bar = 10 μm. **e** Quantification of the number of motor neurons at the anterior half of the nerve cord. L4 ΔC9 ubi: n = 25, day 2 ΔC9 ubi: n = 25, day 5 ΔC9 ubi: n = 25, L4 C9 ubi: n = 29, day 2 C9 ubi: n = 25, day 5 ΔC9 ubi: n = 28. Two-way ANOVA with Šídák's multiple comparison test was performed.

Similar to C9 ubi animals, we found through western blotting that poly-GA is robustly produced in C9 neuro animals, but not in control animals lacking G4C2 repeats and intronic sequences (ΔC9 neuro) or in animals having the non-canonical translation initiation codon CUG mutated to UAG (UAG neuro) (Fig. 3d, e). Importantly, we found a dramatic decrease in the median survival of C9 neuro animals when compared to ΔC9 neuro and UAG neuro controls (P-value < 0.0001) (Fig. 3f and Supplementary Fig. 12). Our findings establish C9 neuro animals as a model for C9orf72-mediated ALS/FTD, and further suggest that decreased levels of poly-GA proteins in neurons are sufficient to improve ALS/FTD-associated phenotypes.

**Genetic removal of eif-2D/eIF2D reduces DPR production and ameliorates lifespan and locomotion defects in C9 ubi animals.** The C9 ubi and C9 neuro worm models for C9orf72-mediated ALS/FTD offer a unique opportunity to uncover the molecular mechanisms responsible for RAN-translated DPRs from G4C2 repeats. Our results suggest that RAN translation of one DPR, poly-GA, in both in vivo models (C9 ubi and C9 neuro worms) requires the non-canonical initiation codon CUG (Figs. 1d–h and 3d, e). We, therefore, surveyed the literature for factors known to initiate translation via this codon. Two proteins, the eukaryotic translation initiation factors eIF2A and eIF2D, are known to deliver tRNAs to the CUG codon and initiate translation

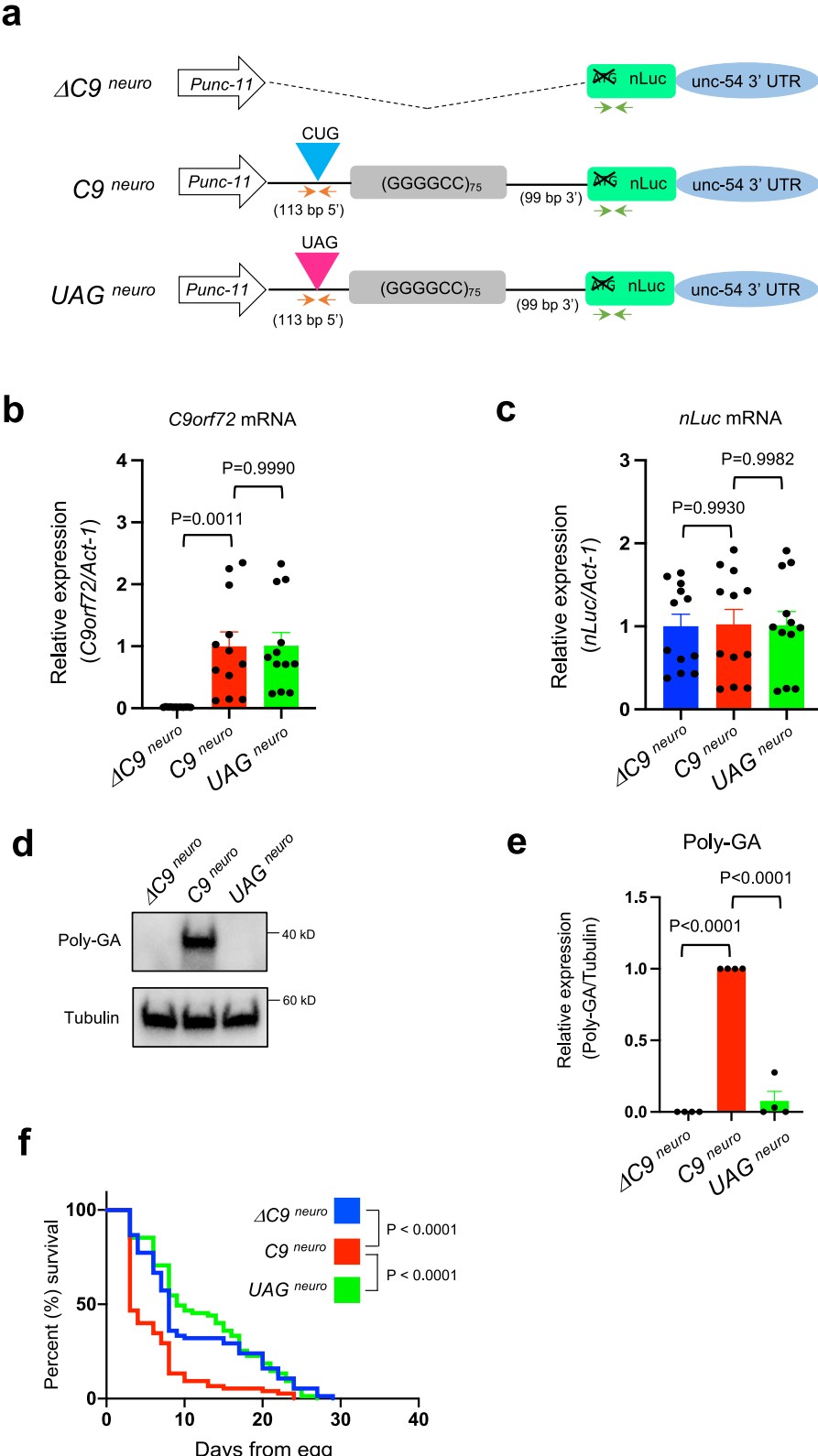

in vitro[38,39]. A single ortholog for each factor is present in the *C. elegans* genome, that is, *eif-2A* for human *eIF2A* and *eif-2D* for human *eIF2D*. We, therefore, questioned whether *eif-2A* and/or *eif-2D* are necessary for DPR expression from the G4C2 repeats. To test this, we obtained mutant animals that carry strong loss-of-function (putative null) alleles for *eif-2A (gk358198)* or *eif-2D (gk904876)*. Both alleles carry missense mutations resulting in a

premature STOP codon: W136Amber for *eif-2A (gk358198)* and R25Opal for *eif-2D (gk904876)* (see "Methods" section). Since these mutant animals are viable and fertile, we crossed them to the *C9 ubi* worms and generated *eif-2A (gk358198); C9 ubi* and *eif-2D (gk904876); C9 ubi* animals. Next, we found via RT-qPCR that the G4C2 repeat RNA with flanking *C9orf72* intronic sequence, as well as the *nLuc* mRNA are normally transcribed in both *eif-2A*

**Fig. 3 DPR expression exclusively in neurons reduces *C. elegans* lifespan. a** Schematic of constructs used in *ΔC9neuro (kasEx159)*, *C9neuro (kasEx157)*, and *UAGneuro (kasEx160)* animals. The orange and green arrows show location of primers used to detect *C9orf72* and nLuc mRNAs, respectively. nLuc: nanoluciferase; UTR: untranslated region. **b** The G4C2 repeat RNA (with *C9orf72* intronic RNA) and *act-1* mRNAs were assessed by RT-PCR. **c** The nLuc and *act-1* mRNAs were assessed by RT-PCR. Experiments were repeated four times using 1 transgenic line per genotype, mean ± s.e.m.). One-way ANOVA with Dunnett's multiple comparisons test was performed. **d** Worm lysates were processed for western blotting and immunostained with poly-GA and α-tubulin antibodies. **e** Quantification of poly-GA expression on western blots. The experiments were repeated three times using 1 transgenic line per genotype, mean ± s.e.m.). One-way ANOVA with Dunnett's multiple comparisons test was performed. **f** Lifespan of *ΔC9 neuro (kasEx159)*, *C9 neuro (kasEx157)*, and *UAG neuro (kasEx160)* animals. The experiments were repeated three times using 1 transgenic line per genotype, mean ± s.e.m. *ΔC9neuro (kasEx159)*, $n = 75$; *C9 neuro (kasEx157)*, $n = 75$; *UAG neuro (kasEx160)*, $n = 75$. See Supplementary Fig. 12. Mantel–Cox log-rank test was performed.

*(gk358198)*; C9 ubi and *eif-2D (gk904876)*; C9 ubi animals (Fig. 4a, b), indicating that loss of *eif-2A* or *eif-2D* does not affect transcription or stability of the repeat RNA. Consistent with these observations, G4C2 RNA foci can be detected in *eif-2D (gk904876)*; UAG ubi, and *eif-2D (gk904876)*; C9 neuro animals (Supplementary Fig. 13). Of note, we found a prominent decrease in poly-GA steady-state levels in *eif-2D (gk904876)*; C9 ubi compared to C9 ubi animals, but no effect in *eif-2A (gk358198)*; C9 ubi animals (Fig. 4c, d), implicating *eif-2D* in poly-GA expression. A rescue assay further established a causal role for *eif-2D* in poly-GA expression (Supplementary Fig. 14). The observed rescue was partial likely due to the mosaic nature of the *eif-2D* rescuing transgene. Interestingly, we observed a modest but statistically significant decrease in poly-GP production in *eif-2D (gk904876)*; C9 ubi animals (Supplementary Fig. 15), suggesting other factors, in addition to *eif-2D*, are also involved in poly-GP translation. We did not observe any effects on poly-GR expression in *eif-2D (gk904876)*; C9 ubi animals (Supplementary Fig. 15).

Since *eif-2D* is necessary for steady-state levels of at least two of the DPRs, loss of *eif-2D* gene activity may improve survival in C9 ubi animals. Indeed, we found that the *eif-2D (gk904876)*; C9 ubi animals display a normal lifespan, similar to ΔC9 ubi and UAG ubi animals (Fig. 4e and Supplementary Figs. 16, 17). Interestingly, the *eif-2D (gk904876)*; UAG ubi animals also display a normal median survival despite producing G4C2 RNA foci (Supplementary Figs. 13a, 17 and 18), suggesting that the improved survival of *eif-2D (gk904876)*; C9 ubi animals are due to decreased DPR (poly-GA and/or poly-GP) expression.

We next asked whether the locomotor defects of C9 ubi animals (detected by automated worm tracking) can be ameliorated upon genetic removal of *eif-2D*, and found this to be the case (Supplementary Figs. 8 and 9). Of note, *eif-2D (gk904876)* single mutants or *eif-2D (gk904876)*; UAG ubi animals did not display any prominent locomotor defects compared to controls (ΔC9 ubi and UAG ubi) (Supplementary Fig. 19). In summary, this behavioral analysis on C9 ubi animals complements our molecular and survival assays (Fig. 4), providing compelling evidence that loss of *eif-2D* ameliorates the pathogenic phenotype associated with RAN translation of DPRs (poly-GA, poly-GP) from *C9orf72* G4C2 nucleotide repeats.

Lastly, we crossed the *eif-2D (gk904876)* mutants to the C9 neuro animals to test whether neuronal *eif-2D* gene activity is critical for poly-GA production and animal survival. Although transcription of the G4C2 repeats (*C9orf72* intronic mRNA) increased in *eif-2D (gk904876)*; C9 neuro for unclear reasons (Fig. 4f, g), we found a prominent decrease in poly-GA production in these animals (Fig. 4h, i). Importantly, loss of *eif-2D* restored survival of C9 neuro animals to levels similar to ΔC9 neuro and *eif-2D (gk904876)* control animals (Fig. 4j and Supplementary Fig. 20). Similar to the case of *eif-2D (gk904876)*;UAG ubi, the *eif-2D (gk904876)*; UAG neuro animals also display a normal median survival (Supplementary Fig. 21).

***eif-2D/eIF2D* is broadly expressed and controls poly-GA steady-state levels independently of G4C2 repeat length.** Eukaryotic translation initiation factors are often expressed broadly in many tissues, but the expression pattern of *C. elegans eif-2D* and its vertebrate orthologs remains unknown. Through CRISPR/Cas9 genome editing (SunyBiotech), we engineered an endogenous reporter allele (*syb3364*) by introducing the gfp-3xFlag sequence before the stop codon of *eif-2d* (Fig. 5a and see "Methods" section). This reporter showed cytoplasmic expression (consistent with the role of *eif-2D* in translation) in most, if not all, *C. elegans* tissues (Fig. 5a, b).

Despite its broad expression, *eif-2D* does not appear to play a role in canonical translation because the loss of *eif-2D* gene activity in *C. elegans* did not affect the expression of an AUG-initiated green fluorescent protein (GFP) transgene or endogenous reporter alleles for two AUG-initiated transcription factors known to be expressed in *C. elegans* ventral nerve cord motor neurons (Supplementary Fig. 22). These findings together with the normal lifespan and locomotion pattern of *eif-2d* mutants (described above) suggest *eif-2D* does not broadly affect canonical translation in *C. elegans*, albeit we cannot exclude the possibility of *eif-2D* being required for translation of other AUG-initiated transcripts.

Next, we sought to determine whether *eif-2D* controls poly-GA steady-state levels independently of G4C2 repeat length. To test this, we generated transgenic animals that carry 5 G4C2 repeats, referred to as *C9 5xG4C2 ubi*, in which nLuc is in the poly-GA frame (Supplementary Fig. 23). Although the RNA levels of intronic *C9orf72* and *nLuc* were similar between *C9 5xG4C2 ubi* and *eif-2d (-)*; *C9 5xG4C2 ubi* animals, a luciferase assay showed that the level of *nLuc* (indicative of poly-GA expression) was significantly decreased in the *eif-2d (-)*; *C9 5xG4C2 ubi* animals (Supplementary Fig. 23). These results together with the data on *C9 ubi* animals that carry 75 G4C2 repeats (Figs. 4 and 5) suggest that *eif-2D* controls poly-GA steady-state levels independently of G4C2 repeat length.

**The SUI1 domain of *eif-2D/eIF2D* is required for poly-GA expression in *C. elegans*.** To obtain mechanistic insights, we interrogated the protein domains of *C. elegans* eIF-2D. Similar to human eIF-2D, we found a PUA domain at the N terminus and a SUI1 domain at the C-terminus of *C. elegans* eIF-2D (Fig. 5a). In eukaryotes, the SUI1 domain is found in translation initiation factors eIF1 and DENR[38]. Because SUI1 is important for accurate translation initiation codon recognition by eIF1[40,41], we hypothesized that the SUI1 domain of *C. elegans* eIF-2D is required for DPR translation. To test this, we first employed CRISPR/Cas9 genome editing and selectively replaced the SUI1 domain with a 3xFLAG epitope (Fig. 5a). Homozygous animals lacking the SUI1 domain (termed *eif-2D ΔSUI1*) expressed a truncated eIF-2D protein (Supplementary Fig. 24) and were viable. When crossed to C9 ubi, we observed a prominent reduction in poly-GA expression with no change in the levels of *C9orf72* intronic

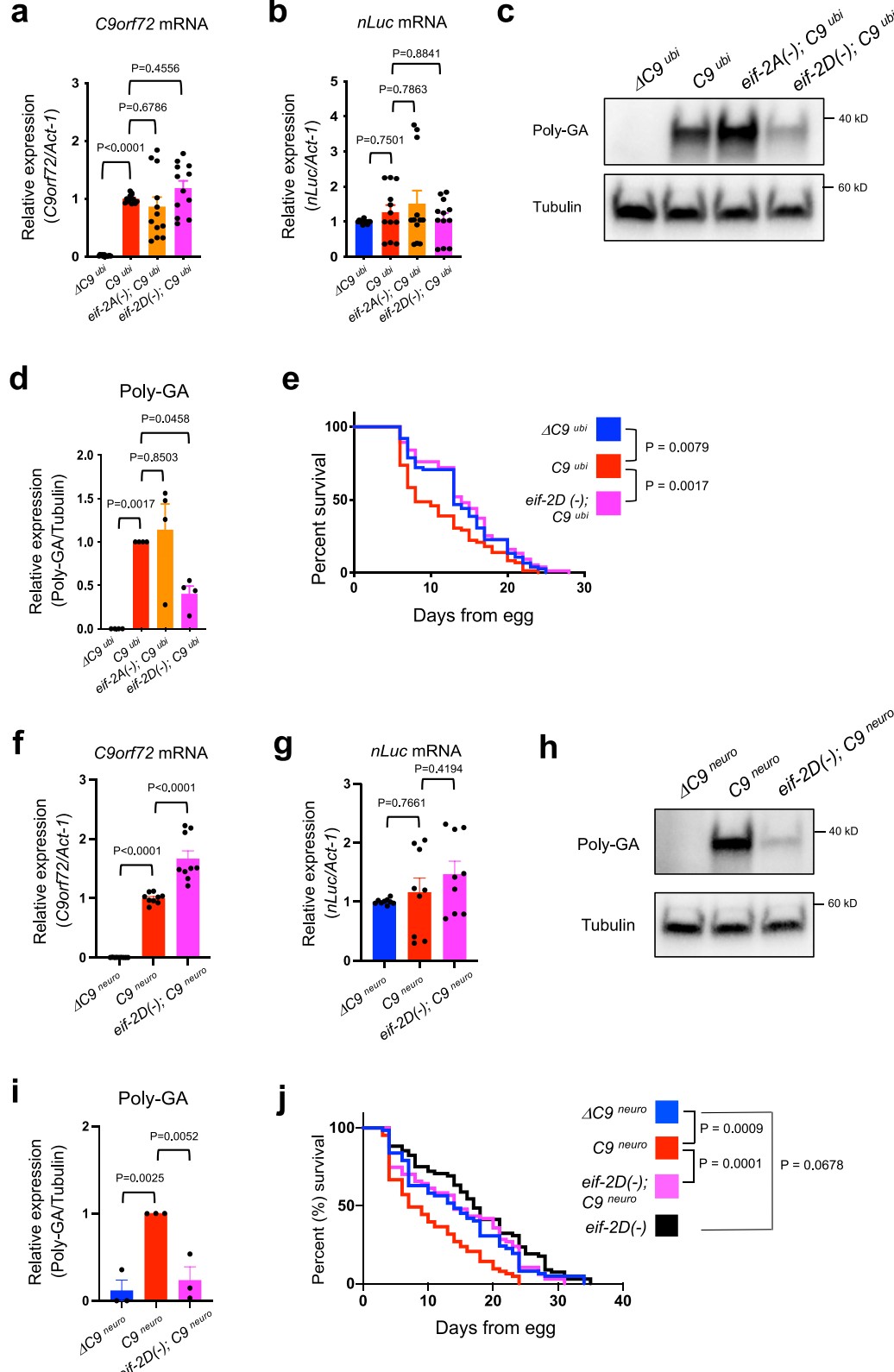

mRNA (Fig. 5c–e). Second, we tested animals carrying the *gk561128* allele that generates a nonsense mutation (Q512Ochre) within the SUI1 domain (Fig. 5a). Again, we observed a strong reduction in poly-GA expression, but no effect on *C9orf72* intronic mRNA (Fig. 5a, f–h). These observations strongly suggest the SUI1 domain of eIF-2D has a key role in poly-GA translation.

**Knockdown of *eIF2D* suppresses poly-GA production in mammalian cells.** The prominent reduction of poly-GA in *eif-2D* mutants suggests that *C. elegans* eIF-2D plays a crucial role in poly-GA translation (Figs. 4 and 5). To extend these findings to mammalian cells, we transfected mouse NSC-34 motor neuron-like cells with a monocistronic construct carrying 75 G4C2

**Fig. 4 Genetic removal of *eif-2D/eIF2D* in C9 *ubi* and C9 *neuro* animals suppresses poly-GA expression and ameliorates lifespan defects. a, f** The G4C2 repeat RNA (primers detect *C9orf72* intronic RNA) and *act-1* mRNAs were assessed (RT-PCR) on mixed stage Δ*C9 ubi* (*kasEx154*), C9 *ubi* (*kasEx262*), *eif-2A* (*gk358198*); C9 *ubi* (*kasEx262*), and *eif-2D* (*gk904876*); C9 *ubi* (*kasEx262*) animals. **b, g** The nLuc and *act-1* mRNAs were assessed by RT-PCR. **c, h** Worm lysates were processed for western blotting and immunostained with poly-GA and α-tubulin antibodies. **d, i** Quantification of poly-GA expression on western blots. The experiments were repeated four times in **a**, **b**, and **d**, and three times in **f**, **g**, and **i**, respectively, using 1 transgenic line per genotype (mean ± s.e.m.). One-way ANOVA with Dunnett's multiple comparisons test was performed (**e**, **j**). **e** Lifespan of Δ*C9 ubi* (*kasEx154*), C9 *ubi* (*kasEx262*), and *eif-2D* (*gk904876*); C9 *ubi* (*kasEx262*) animals. Transgenic lines used: Δ*C9ubi* (*kasEx154*), n = 75; C9*ubi* (*kasEx262*), n = 72, *eif-2D* (*gk904876*); C9 *ubi*, n = 75). **j** Lifespan of Δ*C9 neuro* (*kasEx159*), C9 *neuro* (*kasEx157*), *eif-2D* (*gk904876*); C9 *neuro* (*kasEx157*), and *eif-2D* (*gk904876*) worms. Transgenic lines used: Δ*C9 neuro* (*kasEx159*), n = 62; C9 *neuro* (*KasEx157*), n = 63; *eif-2D* (*gk904876*); C9 *neuro* (*kasEx157*, n = 67; *eif-2D* (*gk904876*): n = 68). Mantel–Cox log-rank test was performed.

repeats and flanking *C9orf72* intronic sequences (Fig. 6a), similar to the one used to generate the C9 *ubi* animals (Fig. 1a). shRNA-mediated knockdown of eIF2D in NSC-34 cells resulted in a strong reduction of poly-GA (Fig. 6b–d), similar to the effects observed with *C. elegans eif-2D* mutants. Compared to control shRNA, eIF2D shRNA did not affect poly-GP and poly-GR steady-state levels in these cells (Fig. 6e, f).

Next, we employed an in vitro platform to differentiate iPSC-derived motor neurons from a healthy individual and two ALS patients with G4C2 repeat expansions (see "Methods" section), (Supplementary Fig. 25). Despite using a sensitive ELISA method, we were unable to detect poly-GA expression above background levels in the iPSC-derived motor neurons from these specific ALS patients, preventing us from testing the role of eIF2D in poly-GA translation in these cells (Supplementary Fig. 26). Future experiments with additional iPSC lines from other ALS patients carrying long G4C2 repeats could help clarify the role of human eIF2D in poly-GA production. However, we did detect poly-GR and poly-GP in these iPSC-derived motor neurons. Similar to our findings in *C. elegans* and NSC-34 cells, siRNA-mediated knockdown of eIF2D had no effect on poly-GR (Supplementary Fig. 26). As in NSC-34 cells, we also observed no difference in poly-GP expression after eIF2D knockdown (Supplementary Fig. 26), which together with the small effect on poly-GP steady-state levels observed in *C. elegans eif-2D* mutants suggest that multiple translation initiation factors control poly-GP expression (see "Discussion" section and Table 1).

Our data with monocistronic G4C2 constructs in *C. elegans* (Figs. 4 and 5), and mouse NSC-34 cells (Fig. 6a–f) are complemented with in vitro data in human HEK293 cells transfected with a bicistronic construct carrying 75 G4C2 repeats and flanking intronic sequences in the second cistron (Fig. 6g and Table 1). To monitor the expression of different DPRs, we generated three different versions of this construct by inserting nLuc in the poly-GA, poly-GP, or poly-GR reading frame (Fig. 6g). In the transfected cells, we robustly detected all three DPRs (Fig. 6h–m). Similar to our findings in *C. elegans eif-2D* mutants, eIF2D knockdown decreased poly-GA and poly-GP (Fig. 6h–k), whereas eIF2D overexpression increased DPR production (Supplementary Fig. 27). Importantly, the protein levels of fLuc, which is expressed by the first cistron via canonical AUG-dependent translation, were similar in cells transfected with either control or eIF2D shRNAs (Fig. 6n), suggesting a specific role for eIF2D in non-canonical translation in human cells. Interestingly, we also observed an effect on poly-GR upon eIF2D knockdown in HEK293 cells transfected with the bicistronic construct (Fig. 6k). We surmise the way DPR translation is initiated in this construct is likely quite different from the monocistronic constructs used in the worm and NSC-34 cells, and from the endogenous human *C9ORF72* locus. Taken together, our in vivo (*C. elegans*) and in vitro (NSC-34, HEK293) data uncovered a conserved role for eIF2D in controlling the steady-state levels of at least one DPR (poly-

GA). Table 1 summarizes the observed effects on DPRs upon *eif-2D*/eIF2D knockdown in vivo (*C. elegans*) and in vitro (NSC-34, HEK293, human iPSC-derived motor neurons).

**eif-2D/eIF2D does not modulate phosphorylation of eukaryotic initiation factor-2α (eIF2α).** A growing body of evidence suggests that phosphorylation of eIF2α as a result of activation of the integrated stress response (ISR) leads to increases in DPR production[29,42]. We, therefore, wondered whether eIF2D is involved in stress-activated phosphorylation of eIF2α. However, we found no significant differences in phosphorylated eIF2α levels between wild-type and *eif-2D* mutant animals carrying the C9 *ubi* transgene (Supplementary Fig. 28). Furthermore, eIF2D knockdown does not affect the level of p-eIF2α in HEK293 cells following administration of a known ER stressor (sodium arsenite) (Supplementary Fig. 29). Lastly, administration of the proteasome inhibitor MG132 in this cellular system did not affect the results of eIF2D knockdown on RAN translation (Supplementary Fig. 30). Altogether, our findings suggest *eif-2D*/eIF2D controls DPR steady-state levels without altering the levels of p-eIF2α.

## Discussion

The discovery of RAN translation of *C9orf72* nucleotide repeats combined with the reported toxicity of DPRs in model systems have focused the ALS/FTD field on this unconventional form of translation, as well as on DPRs as potential therapeutic targets. To study the molecular mechanisms underlying RAN translation, we initially developed a *C. elegans* model that carries a transgene encoding 75 copies of the G4C2 repeat flanked by intronic *C9orf72* sequences under the control of a ubiquitous promoter. These C9 *ubi* animals produce three DPRs from the sense transcript, namely poly-GA, poly-GP, and poly-GR, and show signs of neurodegeneration, as well as locomotor and lifespan defects. Such phenotypes are likely due to DPR toxicity because mutation of a non-canonical CUG initiation codon selectively decreased poly-GA production and ameliorated locomotor and lifespan defects in C9 *ubi* animals without affecting *C9orf72* mRNA levels or the formation of G4C2 RNA foci. Moreover, a second *C. elegans* model (C9 *neuro*) that carries a similar transgene exclusively expressed in neurons displayed a similar phenotype. Importantly, we found that three loss-of-function mutations in the eukaryotic translation initiation factor *eif-2D/eIF2D* suppressed DPR expression (poly-GA, poly-GP) and ameliorated lifespan defects in both C9 *ubi* and C9 *neuro* models. Mechanistically, we found the conserved SUI1 domain at the C-terminus of eIF-2D/eIF2D is required for DPR (poly-GA) expression. Lastly, our in vitro findings in mouse and human cells combined with our *C. elegans* data (Table 1) suggest a conserved role of *eif-2D*/eIF2D in RAN translation.

**The C9 *ubi* and C9 *neuro* worm models offer insights into DPR toxicity.** An unresolved issue in the field of ALS and FTD

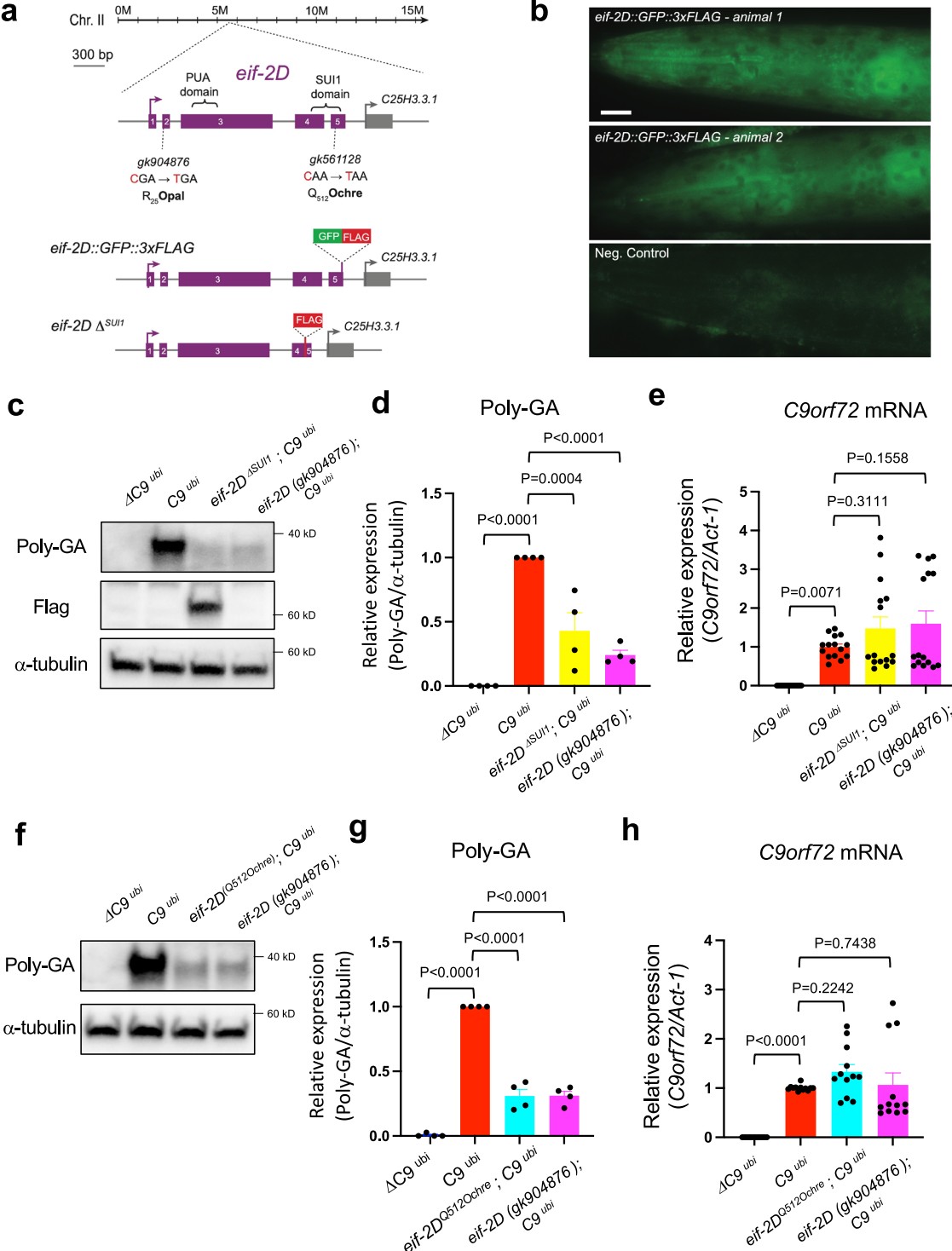

**Fig. 5 The SUI1 domain of eIF-2D is required for poly-GA production. a** Schematic of the *eif-2D* genetic locus. Missense mutations (*gk904876, gk561128*) and CRISPR/Cas9-engineered alleles *(syb3364[eif-2D::GFP::3xFLAG], syb3432[eif-2D ^ΔSUI1])* are shown. The PUA and SUI1 domain are indicated with brackets. **b** An endogenous reporter allele (*syb3364*) shows cytoplasmic *eif-2D* expression. Dark oval or circular shapes of various sizes are nuclei of different cell types. Representative images of the head of two animals (L4 stage) are shown. Negative control is a wild-type (N2) animal (L4). Scale bar = 10 μm. **c, f** Worm lysates were processed for western blotting and immunostained with poly-GA and α-tubulin antibodies. **d, g** Quantification of poly-GA expression on western blots. The experiments were repeated four times. Mean ± s.e.m. One-way ANOVA with Dunnett's multiple comparisons test was performed. **e, h** The G4C2 repeat RNA with *C9orf72* intronic RNA, and *act-1* mRNAs were assessed by RT-PCR. The experiments were repeated four times in **e** and five times in **h**. Mean ± s.e.m. One-way ANOVA with Dunnett's multiple comparisons test was performed.

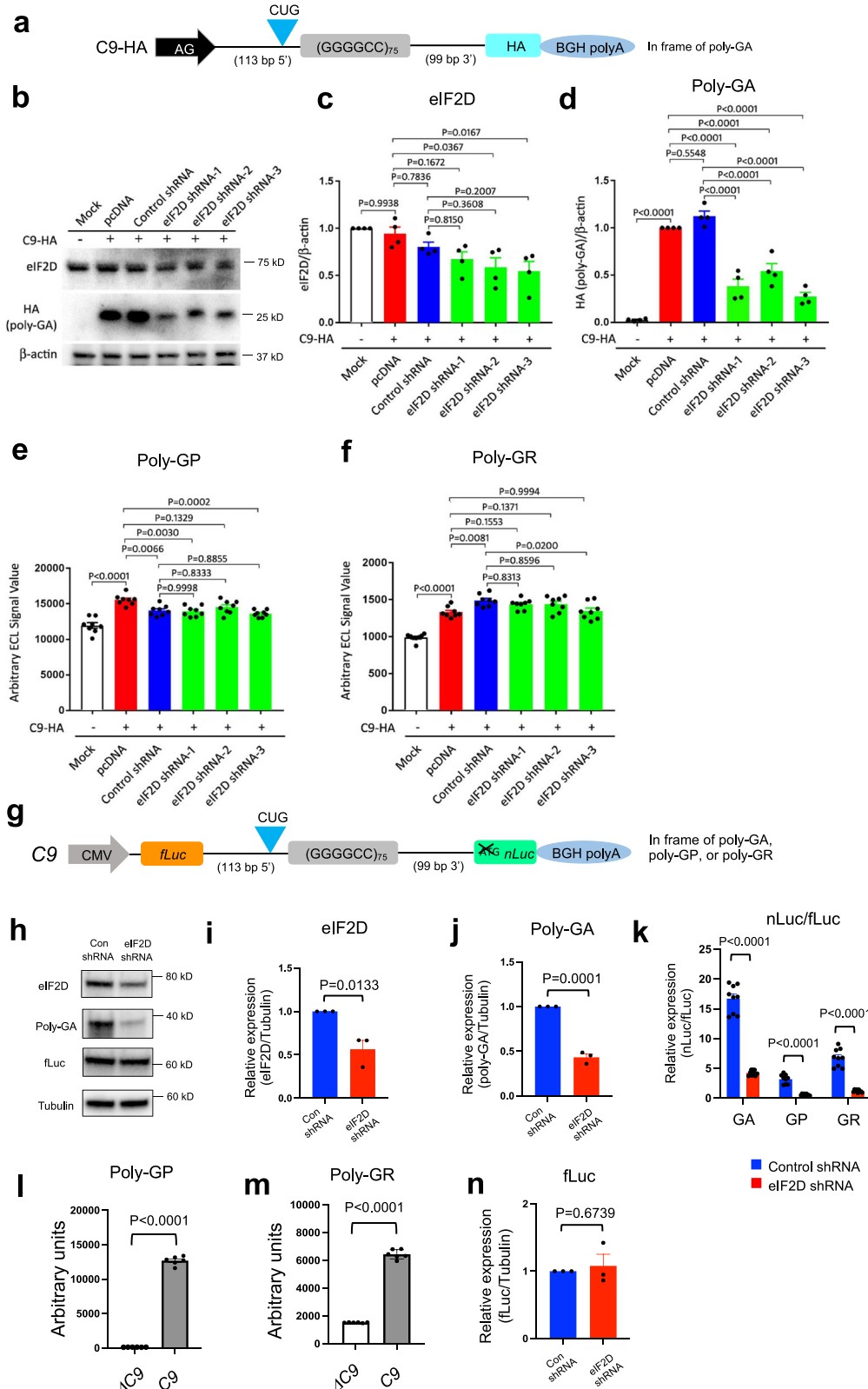

research is the relative contribution of each of the five DPRs (poly-GA, poly-GP, poly-GR, poly-PR, poly-PA) to disease pathogenesis. In this paper, two lines of evidence indicate that poly-GA, the most readily detected DPR in the brain and spinal cord of ALS/FTD patients with *C9orf72* expanded repeats[43,44], significantly contributes to disease phenotypes in *C. elegans*: (1) poly-GA aggregates and appears to be the most abundantly

expressed DPR in our worm models, and (2) mutation of the non-canonical CUG initiation codon selectively decreases poly-GA production, and ameliorates locomotor and lifespan defects in *C9*[ubi] and *C9*[neuro] animals. Importantly, these in vivo results significantly extend previous reports demonstrating that this same CUG is required in vitro for poly-GA translation[20,29–31]. Of note, mutation of this CUG to UAG both

**Fig. 6 Knockdown of eIF2D suppresses DPR expression in NSC-34 and HEK293 cells.** Monocistronic or bicistronic constructs were cotransfected along with either control or anti-eIF2D shRNA and cultured for 48 h. **a** Schematic diagram showing C9-HA construct. AG promoter: CMV enhanced chicken β-actin promoter. HA is in poly-GA frame. FLAG and Myc are in poly-GR and poly-GP frames, respectively, but are not schematically depicted. BGH bovine growth hormone. **b** A representative western blot of NSC-34 lysates to detect poly-GA, eIF2D, and β-actin. **c, d** Quantification of eIF2D (**c**) and poly-GA (**d**) expression from the western blots. **e–f** Quantification of poly-GP (**e**) and poly-GR (**f**) by ELISA in NSC-34 cells. The experiments were repeated four times. mean ± s.e.m. Turkey's multiple comparisons test was used to calculate P-values. One-way ANOVA with Tukey's multiple comparisons test was performed. **g** Schematic diagram showing the bicistronic constructs. CMV cytomegalovirus promoter; fLuc firefly luciferase; nLuc nanoluciferase; BGH bovine growth hormone. **h** The cell lysates were processed for western blotting and immunostained with poly-GA, fLuc, eIF2D, and α-tubulin antibodies. **i, j, n** Quantification of eIF2D (**i**), poly-GA(**j**), and fLuc (**n**) on western blots. The experiments were repeated three times. mean ± s.e.m. Two-tailed unpaired t-test was performed. **k** The levels of luciferase activity were assessed by dual-luciferase assays. The experiments were repeated three times. mean ± s.e.m. **l, m** Poly-GP and poly-GR detected with ELISA in HEK293 cells transfected with a ΔC9 or C9 bicistronic construct. The experiments were repeated three times. mean ± s.e.m. Two-tailed unpaired t-test was performed.

**Table 1 Summary of effects on DPRs with *eif-2D*/eIF2D decrease.**

| Species | G$_4$C$_2$ repeats | eIF2D | Poly-GA | Poly-GP | Poly-GR |
|---|---|---|---|---|---|
| *C. elegans* | Monocistronic construct, 75 G4C2 | Null allele | ↓↓ | ↓ | NS |
| NSC34 cells | Monocistronic construct, 75 G4C2 | KD | ↓↓ | NS | NS |
| HEK293 cells | Bicistronic construct, 75 G4C2 | KD | ↓↓ | ↓↓ | ↓↓ |
| iPSC-derived motor neurons | Endogenous *C9ORF72* locus, >24 G4C2 | KD | ND | NS | NS |

↓↓: >50% decrease.
↓: <50% decrease.
KD knockdown, NS not significant, ND not detected.

in *C. elegans* (Fig. 1) and an in vitro cellular system (Supplementary Fig. 5) did not affect poly-GP and poly-GR translation, suggesting different near-cognate translation initiation codons are used for each DPR. Indeed, an AGG located 1 nt upstream of the repeat has been reported to initiate translation of poly-GR[31]. Despite the above findings, we cannot rule out the possibility that a frameshift occurs in DPR translation upon CUG mutation, as found by other investigators[30].

In support of our *C. elegans* findings, recent studies in mice reported neurodegeneration, motor deficits, and cognitive defects following ubiquitous expression of poly-GA[45], as well as behavioral abnormalities following spinal cord expression of poly-GA[46]. A prevailing hypothesis is that poly-GA causes cellular toxicity through proteasomal impairment and protein sequestration[5]. A recent publication reports that poly-GA spreads cell-to-cell in an in vitro system and inhibits proteasomal function both cell-autonomously and non-cell-autonomously, leading to ubiquitination of the nuclear localization signal of TDP-43 and its subsequent cytoplasmic mislocalization[47]. Importantly, treatment with an antibody against poly-GA ameliorated TDP-43 mislocalization[47], as well as toxicity in mouse models of mutant *C9orf72*-induced ALS/FTD[48]. A zebrafish model that produces poly-GA did not exhibit motor axon toxicity[49], however, this may be due to low poly-GA expression.

Our C9 $^{ubi}$ worm model also produces poly-GP and poly-GR. poly-GP is thought to be relatively nontoxic[5] compared to poly-GR, which has been demonstrated to be toxic in mouse neurons[34]. In *C. elegans*, codon-optimized constructs driving expression of poly-GR or poly-PR caused toxicity in muscles and neurons[50]. Furthermore, initiation of poly-GR translation has been reported to occur at the same upstream CUG codon used for poly-GA production, but with subsequent frameshifting into the GR reading frame[30]. However, two recent papers state that translation initiation of poly-GR did not occur from this upstream CUG codon[31,35]. Because we detected poly-GR in our *C. elegans* model (Fig. 1 and Supplementary Fig. 15), it is possible that poly-GR contributes to the observed disease phenotypes. Supporting this notion, very low levels of poly-GR in neurons have been reported to impair synaptic function and induce behavioral deficits in both *Drosophila* and mice[34].

In addition to what appears to be DPR toxicity in our *C. elegans* models, there remains the question as to whether the repeat RNA itself rather than, or in addition to, its protein product could be toxic[51]. A role for RNA toxicity does not appear to be the case in our nematode experiments since mutation of the non-canonical initiation codon CUG to UAG led to a significant decrease in poly-GA expression and rescue of worm survival to wild-type levels, but had no effect on *C9orf72* mRNA levels nor G4C2 RNA foci formation. It is possible, however, that the CUG > UAG mutation, albeit minimal in nature, can affect RNA structure.

One question that arises is whether our transgenic C9 $^{ubi}$ and C9 $^{neuro}$ animals constitute an appropriate model for the patient with *C9orf72* expanded repeats. It is clear that the RNA template (repeat RNA plus flanking intronic sequence) produced in our *C. elegans* animals differs from the RNA template actually expressed in the central nervous system of ALS/FTD patients with a *C9orf72* expanded repeat. The human G4C2 expanded repeat varies in length in patients and even in different cells within the same patient. The latter finding suggests that a number of potential toxicities—besides one related to DPR production—may be present in the human disease state, including toxicity related to repeat RNA and/or synthesis of other DPRs. Toxicity may also occur as a consequence of a decrease in the authentic *C9orf72* gene product, consistent with a recent study reporting that a decrease in C9ORF72 protein leads to a decrease in autophagic clearance of toxic accumulations of the DPRs[31].

***eif-2D*/eIF2D regulates poly-GA expression from G4C2 nucleotide repeats.** eIF2D is known to initiate translation in vitro at both AUG and non-AUG initiation codons[38,52]. The present study demonstrates a key role of eIF2D in poly-GA expression since eIF2D knockdown in two cellular systems (mouse NSC-34 and human HEK293 cells) and three different *eif-2D* loss-of-function alleles in *C. elegans* led to a prominent decline in poly-GA steady-state levels (Table 1). It is noteworthy that genetic removal of *eif-2D* in *C. elegans* does not appear to have an effect on (a) canonical AUG-initiated translation of three tested mRNAs (Supplementary Fig. 22), (b) animal locomotion,

and (c) lifespan. The absence of detrimental phenotypes in *C. elegans* animals globally lacking *eif-2D*/eIF2D makes this factor an attractive therapeutic target.

What is the mechanism through which *eif-2D*/eIF2D controls poly-GA steady-state levels? We and others found that poly-GA translation is initiated from a CUG (−24nt upstream of G4C2)[20,29–31], and a recent study showed that Met-tRNA is used for initiation of poly-GA translation at this CUG[31]. A previous study showed that both Met-tRNA and non-Met-tRNA are delivered by eIF2D to the P-site of the 40S ribosomal subunit during translation initiation at the HCV IRES in HEK293 cells[38]. Additional cryo-electron microscopy results showed that the SUI1 domain of eIF2D contacts the codon–anticodon duplex on the HCV IRES, and also contacts Met-tRNA, presumably stabilizing its orientation[53,54]. Besides eIF2D, the SUI1 domain is also present in two other non-canonical initiation factors, eIF1 and DENR[38]. It has been proposed that eIF2D may act by changing the conformation of the 40S ribosomal subunit in a manner similar to eIF1, but unlike eIF1, eIF2D stabilizes tRNA binding within the P-site[38]. Using genome engineering in *C. elegans*, we found the SUI1 domain is required for poly-GA steady-state levels. This observation together with the similar effects on DPR expression observed in *eif-2D* (-); C9 ^ubi^ and UAG ^ubi^ animals favor a model in which eIF-2D acts at the CUG codon to deliver the Met-tRNA (which is necessary for poly-GA initiation) to the P-site of the 40S ribosomal subunit. However, it is important to acknowledge that eIF2D has additional functions besides translation initiation. For example, eIF2D was implicated in recycling 40S ribosomal subunits[55–57], which enabled ribosomes and mRNAs to participate in multiple rounds of translation, leading to efficient protein expression. It is thus possible that genetic removal of *eif-2D* in our worm models caused inefficient DPR production by more than one mechanism.

**Multiple factors are involved in DPR translation from G4C2 nucleotide repeats.** As mentioned in Introduction, recent studies suggest multiple factors (e.g., RPS25, eIF3F, eIF4B, eIF4H) are involved in RAN translation of DPRs from G4C2 transcripts. It remains unclear though to what extent the production of each DPR is controlled by these factors, in part because studies often focus on a single DPR. We provide evidence both in *C. elegans* and two cell culture systems (mouse NSC-34 and human HEK293 cells) that *eif-2D*/eIF2D controls the steady-state levels of poly-GA. Moreover, our *C. elegans* data suggest a minor role for *eif-2D*/eIF2D in poly-GP expression and no effect on poly-GR, suggesting that the production of each DPR may involve additional factors. Consistent with this notion, RPS25 knockdown in vitro reduced poly-GA by 90%, poly-GP by 50%, and poly-GR by 30%[13]. A promising candidate to work together with eIF2D is DENR, another initiation factor shown in a recent fly study to act redundantly with eIF2D during development and integrated stress response[58]. Of note, it is possible that these factors do not control each other's expression based on our findings with a *C. elegans* ortholog of eIF4B and eIF-2D (Supplementary Fig. 31). Collectively, these and other studies[16] suggest that a number of different factors play critical roles in RAN translation of DPRs.

Our previous study demonstrated that knockdown of eIF2A in HEK293 cells and neural cells in the chick embryo led to a modest (30%) decrease in poly-GA expression from G4C2 expanded repeats[20]. A recent study also implicated eIF2A in RAN translation from CCUG and CAGG repeats in myotonic dystrophy type 2 (DM2)[59]. In the present study, however, *eif-2A*/eIF2A deficiency had no effect on poly-GA expression in *C. elegans*, while eIF2D knockdown in NSC-34 and HEK293 cells and *eif-2D* knock-out in *C. elegans* had a prominent effect on

poly-GA expression. These results suggest that eIF2D plays a more important role in RAN translation than eIF2A. It is possible, however, that eIF2A also plays a significant role in RAN translation from G4C2 expanded repeats, but only upon cellular stress since it can translate a subset of mRNAs when eIF2α is phosphorylated[17,18,60]. We note that eIF2D does not control the levels of phosphorylated eIF2α in *C. elegans* and HEK293 cells (Supplementary Figs. 27 and 28).

**RAN translation occurs in *C. elegans* and offers the opportunity to discover conserved regulators.** Two previous studies reported DPR-associated toxicity in *C. elegans* by using either a codon-optimized construct that carries no G4C2 repeats, or a construct that carries 66 G4C2 repeats[22,50]. The latter study strongly indicated that RAN translation occurs in the worm. However, the non-canonical (non-AUG) initiating codon and translation initiation factor remained elusive. By generating *C. elegans* animals that carry 75 G4C2 repeats flanked by intronic *C9orf72* sequences and subsequently blocking poly-GA expression by mutating a non-canonical initiation codon (CUG) upstream of the repeats, we advanced the molecular understanding of how RAN translation occurs in the worm. This finding in *C. elegans*, together with previous reports in yeast and flies[16,22], strongly suggest that the molecular machinery required for RAN translation is evolutionary conserved, offering an opportunity to use simple model systems for the discovery of RAN translation regulators.

In summary, we leveraged the specific strengths of the *C. elegans* model and discovered that the non-canonical initiation factor *eif-2D*/eIF2D is required for DPR expression (poly-GA, poly-GP) from expanded G4C2 repeats in the *C9orf72* gene. Our in vitro findings argue for a conserved role of *eif-2D*/eIF2D in poly-GA expression, which could be potentially used to pursue directions for *C9orf72*-associated ALS and FTD therapy. In fact, because RAN translation of other nucleotide repeat expansions (not G4C2) occurs in several neurodegenerative diseases[4], eIF2D could be investigated as a therapeutic target in these disorders as well.

## Methods

**C. elegans strains.** Worms were grown at 15, 20, or 25 °C on nematode growth media (NGM) plates supplied with *E. coli* OP50 as food source[61]. Mutant alleles used in this study include *eif-2A* (gk358198), *eif-2D* (gk904876), and *eif-2D* (gk561128), which were backcrossed three times to the wild-type (N2) strain. The following strains were generated using CRISPR/Cas9 genome editing by Suny-Biotech. The strain PHX3364 with genotype *eif-2D* (syb3364) is an endogenous reporter for *eif-2d* because the GFPnovo2-3xFLAG sequence has been inserted before the stop codon. The strain PHX3432 with genotype *eif-2D* (syb3432) carries a replacement of the SUI1 domain with a 3xFLAG sequence. A complete list of all primers used in this study is provided in Supplementary Table 1.

**Generation of transgenic C. elegans animals.** The template for the generation of the plasmids used in the C9 ^ubi^, ΔC9 ^ubi^, UAG ^ubi^ and C9 5xG4C2 ^ubi^ animals has been previously described in an in vitro study[20]. In C9 ^ubi^ animals, the plasmid has 75 G4C2 repeats with 113 nucleotides 5′ and 99 nucleotides 3′ of intronic sequences that normally flank the G4C2 repeats in the human *C9orf72* gene; this sequence is followed by nanoluciferase (nLuc) reporter in the reading frame of poly-GA, which is followed by the *unc-54* 3′ UTR. In ΔC9 ^ubi^ animals, the construct lacks the G4C2 repeats and the flanking intronic sequences. In UAG ^ubi^ animals, the construct is identical but the non-canonical translation-initiating codon CUG is mutated to UAG. The promoter sequences of the *snb-1* and *unc-11* genes are shown in Supplementary Data 1. The *unc-54* 3′ UTR was cloned into the XhoI-SacI site of the plasmids.

To generate the ΔG4C2 repeats ^ubi^ construct, ΔG4C2-1-F/R and ΔG4C2-2-F/R fragments were phosphorylated, annealed, and then ligated into restriction sites of HindIII and NotI of the C9 ^ubi^ construct. To generate the C9 ΔnLuc ^ubi^ construct, the NotI-ΔnLuc-BamHI-F/R fragment was inserted into restriction sites of NotI and BamHI of the C9 ^ubi^ construct. For the 3xFlag-*eif-2D* ^ubi^ construct, the 3xFlag sequence (synthesized by Integrated DNA Technologies) was inserted into restriction sites of HindIII and NotI, and *eif-2D* cDNA was inserted into the restriction sites of NotI and Bam of the C9 ^ubi^ construct. For the C9-GFP ^ubi^

construct, GFP novo2[62] was inserted into restriction sites of *Not*I and *Bam*HI of the C9 *ubi* construct.

The C9, UAG, ΔC9, ΔG4C2 repeats, C9 ΔnLuc, C9 5xG4C2, or C9-GFP constructs were microinjected into the *C. elegans* gonad at 100 ng/µl along with a myo-2:GFP or myo-2:mCherry plasmid (3 ng/µl), and a pBlueScript plasmid (20 ng/µl) (Supplementary Data 2). Multiple transgenic lines per construct were established (Supplementary Data 3). The resulting C9 *ubi*, ΔC9 *ubi*, and UAG *ubi* animals, as well as the resulting C9 *neuro*, ΔC9 *neuro*, and UAG *neuro* animals, carried these transgenes as extrachromosomal arrays. The exact extrachromosomal lines used in each experiment are indicated in figure legends and Supplementary Data 3. We also generated integrated versions of the ΔC9 *ubi*, C9 *ubi*, and UAG *ubi* constructs (Supplementary Data 3). One integrated line for each was obtained and outcrossed five times to a wild-type (N2) strain. In Supplementary Data 3, we also provide details on transgenes: *vsIs48 [unc-17::gfp]* and *otIs181 [dat-1::mCherry +ttx-3::mCherry]*.

**Lifespan assay.** The lifespan assay was performed as previously described[22]. Worms were transferred at the L4 larval stage to NGM plates coated with OP50 bacteria in the presence or absence of 50 µM 5-fluoro-2'deoxyuridine (FUDR, Sigma). Worms were kept at 20 °C and scored every 2 or 3 days. If the worms did not respond to prodding the head with the wire spatula, they were scored as dead worms. In the assay with FUDR, the worms were moved to new plates before bacterial food was depleted. In the assay without FUDR, the worms were moved to new plates every 2 days. If we were unable to find the same number of worms on the plate after 2 days, they were scored as censored worms. Similar results were obtained when lifespan assays were performed with (Figs. 2a, 3f, and 4e, 4j) or without (Supplementary Figs. 17 and 21) FUDR. The day of the timed egg-laying was considered day 0 for lifespan analysis. The experiments were replicated two to three times. The exact number of biological replicates, censored animals, dead animals, and worms used in each trial is shown in Supplementary Figs. 7, 12, 16, 17, 20, and 21. The Mantel–Cox log-rank test was performed for statistical analysis.

**Microscopy.** Worms were anesthetized using 100 mM of sodium azide (NaN₃) and mounted on 5% agarose pads on glass slides. Images were taken using an automated fluorescence microscope (Zeiss, Axio Imager Z2) equipped with a Zeiss Axiocam 503 mono using the ZEN software (Version 2.3.69.1000, Blue edition). Representative images are shown following max-projection of 2–5 Z-stack images (each ~0.5 µm) using the maximum intensity projection type. Image reconstruction was performed using Image J software (version 2.0.0-rc-59/1.51k)[63].

**Quantification of microscopy images.** The number of neurons expressing the respective fluorescent reporter was quantified manually using Image J software (version 2.0.0-rc-59/1.51k)[63]. Quantification was performed in a blinded fashion. Details related to the fluorescent reporter, as well as the number and stage of animals analyzed are provided in the respective figure legends.

**Immunocytochemistry**

*C. elegans.* Animals of the respective genotypes were grown at 20 °C on nematode growth media (NGM). We followed an immunocytochemical staining procedure described previously[64]. In brief, worms were prepared for staining following the freeze-crack procedure and subsequently fixed in ice-cold acetone (5 min) and ice-cold methanol (5 min). Worms were transferred using a Pasteur pipette from slides to a 50 ml conical tube that contained 40 ml 1 × PBS. Following brief centrifugation (2 min, 1800 × g), worms were pelleted and 1 × PBS was removed. Next, worms were incubated with 300 µl of blocking solution (1 × PBS, 0.2% Gelatin, 0.25% Triton X100) for 30 min at room temperature (rolling agitation). Following removal of the blocking solution, worms were incubated overnight with a mouse anti-poly-GA antibody (1:25 in PGT solution, EMD Millipore). Next, the primary antibody solution was removed and worms were washed five times with washing solution (1 × PBS, 0.25% Triton X100). Worms were incubated with an Alexa Fluor 594 goat anti-mouse IgG secondary antibody (1:1000 in PGT solution, A-11020, Molecular Probes) for 3 h at room temperature. Following 5 washes, worms were mounted on a glass slide and examined with an automated fluorescence microscope (Zeiss, AXIO Imager Z1 Stand).

*iPSC-derived motor neurons.* Motor neurons were seeded onto a preplated mouse glial monolayer. After 50 days in culture, MNs were fixed with 4% paraformaldehyde for 15 min at RT. Cells were then washed with PBS and blocked/permeabilized for 2 h in PBS containing 0.25% Triton and 10% normal donkey serum (Jackson ImmunoResearch, Cat # 017-000-001). Fixed MNs were then incubated overnight at 4 °C with primary antibodies: ChAT (goat, 1:150, Millipore, AB144P), ISLET 1/2 (mouse, 1:100, DSHB, 39.4D5), and MAP2 (chicken, 1:5000, Abcam, Ab5392). The following day, PBS + 0.1% Triton was used for 3–10 min washes. Samples were then incubated with secondary antibodies conjugated to Alexa488 (1:500, Life Technologies A-11055), Alexa555 (1:500, Life Technologies A-31570), and Alexa647 (1:500, Molecular Probes/Jackson ImmunoResearch), while cell nuclei were stained with DNA labeling Hoechst (1:500, Life Technologies, H21492).

**RNA fluorescent in situ hybridization (FISH).** RNA FISH was carried out as previously described[65]. For each genotype, well-fed worms at various stages and grown on 4–5 Petri dishes were fixed with fixation buffer (3.7% formaldehyde in PBS) for 1 h, and then resuspended in 70% ethanol overnight. The samples were washed with wash buffer (10% deionized formamide and 2 × SSC) and hybridized with 5 µM RNA probe (CCCCGGCCCCGGCCCCGG) conjugated with Quasar 670 at the 3' end (LGC Biosearch Technologies) in hybridization buffer (10% (w/w) dextran sulfate, 10% deionized formamide, and 2 × SSC) overnight at 37 °C. The samples were then washed with wash buffer for 30 min at 37 °C, and subsequently resuspended with 5 ng/ml DAPI diluted in wash buffer for 30 min at 37 °C. After washing with 2 × SSC, the samples were resuspended in GLOX buffer (0.4% glucose, 10 mM Tris-HCl, pH8.0, 2 × SSC, 3.7 µg/ml glucose oxidase, and >300 units catalase), placed onto a slide, and examined with an automated fluorescence microscope (Zeiss, AXIO Imager Z1 Stand).

**Automated worm tracking.** Worms were maintained as mixed stage populations by chunking on NGM plates with *E. coli* OP50 as the food source. Worms were bleached and the eggs were allowed to hatch in M9 buffer to arrest as L1 larvae. L1 larvae were refed on OP50 and allowed to grow to day 2 or adulthood. On the day of tracking, five worms were picked from the incubated plates to each of the imaging plates (see below) and allowed to habituate for 30 min before recording for 15 min. Imaging plates were 35 mm plates with 3.5 ml of low-peptone (0.013% Difco Bacto) NGM agar (2% Bio/Agar, BioGene) to limit bacteria growth. Plates are stored at 4 °C for at least 2 days before use. Imaging plates were seeded with 50 µl of a 1:10 dilution of OP50 in M9 the day before tracking and left to dry overnight with the lid on at room temperature.

**Behavioral feature extraction and analysis.** All videos were analyzed using Tierpsy Tracker (Version 228) to extract each worm's position and posture over time[36]. These postural data were then converted into a set of behavioral features selected from a large set of features as previously described[36]. For each strain comparison, we performed unpaired two-sample *t*-tests independently for each feature. The false discovery rate (FDR) was controlled at 5% across all strain and feature comparisons using the Benjamini Yekutieli procedure. The *P*-value threshold for this analysis (FDR 5%) was 0.0014.

**Swimming behavior.** Animals were grown at 15 °C until they were assayed at three different stages (L4, day 2, day 5).

Swimming rates were measured with 20 animals per genotype. Individual animals were placed on 40 µl of temperature-equilibrated M9 buffer on an agar plate. After 1 min of acclimatization, the number of changes in the direction of bending at the mid-body region was recorded for 30 s. Animals were chosen at random by assaying the first worm in the field of view.

**In vitro studies with NSC-34 motor neuron-like cells.** shRNA plasmids against mouse eIF2D were constructed by inserting annealed oligos into *Sgr*AI and *Eco*RI sites of pFB-U6-shRNA-CMV-GFP vector[66]. NSC-34 cells in a 6-well plate were transfected with either C9-3T plasmid alone or in combination with eIF2D shRNA using transfection reagent Attractene (Qiagen) according to the manufacturer's protocol and cultured for 48 h. The NSC-34 cells are a gracious gift from Dr. Neil Cashman[67].

**In vitro studies with HEK293 cells.** Bicistronic plasmids were constructed, as previously described[20], with firefly luciferase (fLuc) in the first cistron and C9orf72 sequences in the second cistron, with or without a UAG replacing the CUG 24 nts upstream of the 75 G4C2 repeats.

shRNA plasmids against human eIF2D were prepared using previously published methods[20]. In brief, oligonucleotides with an siRNA sequence were cloned into the *Bam*HI and *Hin*dIII sites of psilencer 2.1-U6 neo vector (ThermoFisher Scientific) according to the manufacturer's protocol. The control shRNA vector was provided in this kit. The eIF2D-V5 Tag plasmid was prepared by inserting human eIF2D cDNA (RC201726, Origene) into restriction sites of *Eco*RI and *Not*I of pcDNA6. The GFP-V5 Tag plasmid was prepared by inserting GFP sequence from pFB-U6-shRNA-CMV-GFP plasmids[66] into restriction sites of *Eco*RI and *Not*I of pcDNA6.

For western blotting (see below), HEK293 cells (ATCC, CRL-1573) were plated in 6-well plates at 2 × 10⁵ per well and cotransfected with 0.5 µg of bicistronic plasmids along with either 2.5 µg control shRNA or anti-eIF2D shRNA using Lipofectamine LTX (ThermoFisher Scientific). For luciferase assays (see below), the cells were plated in 24-well plates at 5 × 10⁴ per well and cotransfected with 100 ng of the bicistronic plasmid along with 500 ng of either control shRNA or anti-eIF2D shRNA using Lipofectamine LTX (ThermoFisher Scientific). For the eIF2D overexpression assay, cells were cotransfected with 20 ng of bicistronic plasmids along with 380 ng of either GFP-V5 Tag or eIF2D-V5 Tag plasmids using Lipofectamine LTX. The cells were cultured for 48 h in DMEM supplemented with 10% FBS, 2 mM L-Glutamine, 100 U/ml Penicillin and 100 µg/ml Streptomycin, and then processed.

**In vitro differentiation of motor neurons from human induced pluripotent stem cells (iPSC).** Motor neuron cultures were prepared as described by Ortega et al.[68]. Briefly, induced pluripotent stem cells (iPSCs) were dissociated using Accutase (Innovative Cell Technologies) and plated at a final density of 100,000 cells/cm[2] in mTESR1 (STEMCELL) supplemented with 10 μM ROCK inhibitor (DNSK International). Twenty-four hours later, the iPSC culture medium was changed to a differentiation medium (50% DMEM:F12, 50% Neurobasal, non-essential amino acids (NEAA), Glutamax, N2, and B27) (all from ThermoFisher Scientific). For the first 6 days, the differentiation culture medium was changed daily and supplemented with 10 μM SB432542 (DNSK International), 100 nM LDN-193189 (DNSK International), 1 μM Retinoic Acid (Sigma), and 1 μM SAG (DNSK International) to induce spinal neural progenitors. For the next 7 days, the differentiation culture medium was changed daily and supplemented with 5 μM DAPT (DNSK International), 4 μM SU5402 (DNSK International), 1 μM Retinoic Acid, and 1 μM SAG to generate spinal motor neurons. Following the differentiation, cells were dissociated with TrypLE Express (ThermoFisher Scientific) with DNase I (Worthington) and plated onto plastic culture plates coated with Matrigel (Corning). Motor neurons were fed every other day with MN medium (Neurobasal medium which is supplemented with Glutamax, N2, B27, NEAA, Hyclone FBS (Cytiva), 0.2 μg/ml Ascorbic Acid (Sigma), and 10 ng/ml each BDNF, CNTF, and GDNF) (all from R&D Systems)).

*siRNA knockdown experiments.* After 35 days in culture, motor neurons were transfected with a siRNA specific to eIF2D mRNA or a scrambled control. First, a lipofectamine-siRNA (ThermoFisher Scientific) complex was prepared separately for both the targeting and nontargeting siRNA in OptiMEM with incubation at room temperature for 20 min. After this period, the lipofectamine RNAiMAX (ThermoFisher Scientific)-siRNA solution was brought up to the appropriate volume with MN culture medium. The culture medium in the dish was then aspirated and replaced with a lipofectamine–siRNA solution at a final concentration of 60 pmol siRNA in 1.5 ml medium per 1,000,000 cells. Twenty-four hours later the medium was replaced with a normal MN medium. This procedure was repeated 2 more times at 40 and 45 days in culture. After 50 days in culture, motor neurons were lysed with RIPA buffer (10 mM Tris-HCL pH 8.0, 140 mM NaCl, 1 mM EDTA, 0.1% SDS) supplemented with protease inhibitors (Millipore) on ice for 10 min and sonicated using a microneedle sonicator. After removing the RIPA-insoluble fraction by centrifugation, the RIPA-soluble fraction was assessed for levels of the DPRs.

*ALS patient iPSC line information.* Control (18A) line previously characterized for pluripotency[69]. Harvard University; RRID:CVCL_8993. Female, 48 years old.

C9 #1 (TALS9-11.2) line obtained from Target ALS. Cat# ND50008; RRID:CVCL_FA04. Female, 61 years old. Pluripotency certificate is provided in Supplementary File 1. This line is previously described in Ortega et al.[68], where repeat-primed PCR confirmed a large number of G4C2 repeats.

C9 #2 (ND10689) iPSC line was reprogrammed by and received from the Ichida Lab (University of Southern California). B-lymphocytes were originally purchased from Coriell. Cat# ND10689; RRID:CVCL_BE81. Female, 51 years old. Pluripotency status and presence of a G4C2 repeat expansion were previously described in Shi et al.[70]. In the latter study, the G4C2 repeat size is ~14 kb for this line. Repeat-primed PCR in Ortega et al.[68] also confirmed a large number of G4C2 repeats.

**Western blotting.** Lysates of worms were prepared by sonication in buffer consisting of 4% urea and 0.5% SDS with 1 × Halt[TM] Protease and Phosphatase inhibitor Cocktail (ThermoFisher Scientific). NSC-34 cell lysates were prepared using RIPA buffer (50 mM Tris-HCl, pH 7.5; 150 mM NaCL; 1% Triton-X 100; 0.1% SDS; 0.5% sodium deoxycholate; 2 mM EDTA and 10% glycerol) containing protease inhibitor cocktail (Sigma) and phosphatase inhibitor cocktail (Sigma). HEK293 cell lysates were prepared using RIPA buffer containing 1 × Halt[TM] Protease and Phosphatase inhibitor Cocktail. These lysates were subjected to electrophoresis on 4–20% SDS polyacrylamide gels (Mini-PROTEAN TGX Gels, BIO-RAD), and then transferred to Amersham Hybond P 0.45 μm PVDF membranes (GE Healthcare). The membrane was blocked with 1% non-fat skim milk in Tris-buffered saline containing 0.05% Tween-20 for 1 h at room temperature, and then incubated overnight at 4 °C with primary antibodies against poly-GA (1:1000, MABN889, EMD Millipore), fLuc (1:1000, G7451, Promega), FLAG (1:300, F7425, Sigma), FLAG (1:1000, 9A3, Cell Signaling Technology), GFP (1:1000, D5.1, Cell Signaling Technology), V5-Tag (1:1000, D3H8Q, Cell Signaling Technology), HA (1:1000, C29F4, Cell Signaling Technology), human eIF2D (1:2000, ab108218, Abcam), mouse eIF2D (1:1500, 12840-1-AP, Proteintech), p-eIF2α (1:1000, 119A11, Cell Signaling Technology), eIF2α (1:1000, L57A5, Cell Signaling Technology), α-tubulin (1:5000, YL1/2, Abcam), or β-actin (1:5000, A5441, Sigma). Following washing, the membrane was incubated for 1 h at room temperature with anti-mouse (1:5000, GE Healthcare), anti-rabbit (1:5000, GE Healthcare), or anti-rat horseradish peroxidase–conjugated secondary antibodies (1:1000, Cell Signaling Technology). The signal was detected using SuperSignal West Dura Extended Duration Substrate (ThermoFisher Scientific) and analyzed using ChemiDoc MP Imaging System and Image Lab software (version 6.0.1, BIO-RAD). Full scan western blot images are provided in the Source Data file.

**Luciferase assay.** The cells were lysed with 1× passive lysis buffer (Promega). Levels of nLuc and fLuc were assessed using Nano-Glo Dual-Luciferase Reporter assay system (Promega) and a Wallac 1420 VICTOR 3V luminometer (Perkin Elmer) according to the manufacturer's protocol. The 2030 Workstation software (version 4.0) was used.

**RT-PCR.** To analyze the expression levels of *C. elegans* mRNA, total RNA was extracted using TRIzol (ThermoFisher Scientific) according to the manufacturer's protocol and treated with TURBO DNase (ThermoFisher Scientific) for 30 min at 37 °C. RNA was then isolated using RNeasy Mini kit (Qiagen). cDNA was generated using SuperScript IV First-Strand Synthesis System (ThermoFisher Scientific). Quantitative RT-PCR was performed using Power SYBR Green PCR Master Mix (ThermoFisher Scientific) and primer sets (Supplementary Data 2) in a CFX96 Real-Time System (Bio-Rad). The CFX Manager software (version 3.1) was used.

To analyze mRNA expression levels in iPSC neurons, total RNA was extracted from siRNA-treated motor neurons using an RNeasy Mini Kit. The cDNA was generated using iSCRIPT Reverse Transcription Supermix (BIO-RAD). RT-qPCR analysis was performed using the iTaq Universal SYBR Green Supermix (BIO-RAD) on the CFX system. To calculate relative expression, the value of the average cycle of threshold (Ct) for housekeeping genes (GAPDH and ACTIN) was subtracted from the average Ct value of eIF2D. This ΔCt was then used to determine the relative gene expression described by the function $2^{-\Delta Ct}$ (ΔΔCt), represented as a value relative to the control sample.

**ELISA.** Two ELISA methods were used and DPRs were detected in a blinded fashion. (A) The worm pellets were suspended in cold co-IP buffer consisting of 50 mM Tris-HCl, pH 7.4, 300 mM NaCl, 5 mM EDTA, 1% Triton-X 100, 2% sodium dodecyl sulfate, 1 × Halt[TM] Protease, and Phosphatase Inhibitor Cocktail (ThermoFisher Scientific) and sonicated with 10 cycles of 0.5 s pulse/0.5 s pause. Protein lysates were cleared by centrifugation at 16,000 × *g* for 20 min at 4 °C. The protein concentration was determined using a BCA protein assay kit (Thermo-Fisher Scientific). Lysates were diluted to the same concentration using co-IP buffer, and the levels of poly-GP in lysates were measured with a Meso Scale Discovery-based immunoassay as described previously[71]. (B) *C. elegans*, HEK293 cells, and human patient iPSC-derived neurons were homogenized in lysis buffer containing 150 mM NaCl, 25 mM Tris, pH 7.5, 1 mM EDTA, 1 mM EGTA,1% Triton-X 100, 0.5% SDS, and 1 × Halt[TM] Protease Inhibitor Cocktail (Thermo-Fisher Scientific) and sonicated at a 20% pulse rate for 15–20 s on ice. Lysates were centrifuged at 16,000 × *g* for 15–20 min at 4 °C, and the supernatant was collected in a fresh tube. The protein concentration of the lysates was determined with the BCA protein assay (ThermoFisher Scientific). Lysates were diluted to the same concentration with lysis buffer and tested in duplicate wells using Meso Scale Discovery platform. Measurement of DPRs was carried out using a sandwich immunoassay as described earlier[72].

**Statistical analysis.** Statistical analysis was performed by an unpaired *t*-test, one-way ANOVA with Dunnett's multiple comparisons test or Tukey's multiple comparison test, and two-way ANOVA with the Šídák multiple comparison test using GraphPad Prism version 8.2.1. *P*-values for the worm lifespan assay were calculated by the Mantel–Cox log-rank test using GraphPad Prism 8.2.1. A *P*-value of <0.05 was considered significant. The data are presented as mean ± standard error of the mean (SEM).

**Reporting summary.** Further information on research design is available in the Nature Research Reporting Summary linked to this article.

## Data availability
The data supporting the findings of this study are available from the corresponding authors upon reasonable request. Source data are provided with this paper.

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

## Acknowledgements

We thank the *Caenorhabditis* Genetics Center (CGC), which is funded by the NIH Office of Research Infrastructure Programs (P40 OD010440), for providing strains. The antibodies in the MSD assay that detect poly-GA in human iPSC-derived motor neurons were used at Biogen, and were discovered by Neurimmune AG. This work was supported by a grant from the Lohengrin Foundation (R.P.R., P.K.), a basic science pilot grant from the Association for Frontotemporal Degeneration (AFTD) (R.P.R., P.K.), NIH NINDS/NIA grant R01NS104219 (E.K.), NIH grant R37NS057553 (F.-B.G.), NIH grant R01NS101986 (F.-B.G.), and the Target ALS Foundation (F.-B.G.). E.K. is a Les Turner ALS Research Center Investigator and a New York Stem Cell Foundation – Robertson Investigator.

## Author contributions

Study design: Y.S., R.P.R., and P.K. Literature research: Y.S., R.P.R., and P.K. Experimental studies: Y.S., J.A., G.K., A.C.F, G.G., P.I., E.C.W., Y.G., M.W.K., T.F.G., E.K., and P.K. Data analysis/interpretation: Y.S., J.A., A.E.X.B., P.K., R.P.R. A.C.F., E.K, T.F.G., F.-B.G., R.P.R., and P.K. Statistical analysis: Y.S. and A.E.X.B. Manuscript preparation—original draft: Y.S., R.P.R., and P.K. Manuscript editing: Y.S., G.G., A.C.F., E.K., T.F.G., R.P.R., and P.K.

## Competing interests

The authors declare no competing interests.
