## [Peer Review File · Nature Communications]

A *C. elegans* model of C9orf72-associated ALS/FTD uncovers a conserved role for eIF2D in RAN translationREVIEWER COMMENTS

Reviewer #1 (Remarks to the Author):

In this study, Sonobe et al. developed *C. elegans* models that express, either ubiquitously or exclusively in neurons, a transgene with G4C2 repeat expansion. These animals generate DPRs including polyGA and polyGP and display neurodegeneration, locomotor and lifespan defects. Translation initiation of polyGA is at a near-cognate start codon CUG upstream of the G4C2 repeats. Additionally, the authors showed that eukaryotic translation initiation factor 2D (eif2D/eIF2D) can regulate RAN translation. Genetic removal of eif2D increased lifespan in both *C. elegans* models. Knock-down of eIF2D inhibits the production of DPRs in a cell culture model. These findings provide additional tools to understand the mechanisms of RAN translation and may contribute to the development of future therapeutic strategies for C9 ALS/FTD. Overall, this is a potentially interesting story, but aspects of the story are not new and additional experiments that provide a more detailed mechanistic insight should be provided to strengthen the impact of this manuscript. Specific concerns and suggestions to improve the potential impact of the study are listed below.

Major concerns:

1. In C9 animals, the plasmids used for developing worm models contains 113 nt upstream and 99 nt of sequence downstream of the G4C2 repeats, followed by nanoluciferase (nLuc) reporter in the reading frame of poly-GA. In methods, the authors described "In Δ C9 ubi animals, the construct is identical but lacks the G4C2 repeats". Does the construct contain upstream and downstream flanking sequences surrounding the G4C2 repeats? The control animals, Δ C9 ubi, should contain the flanking sequences, but in Fig. 1A and 1B, it seems that this construct does not contain the flanking sequences.
2. A lot of C9orf72 models have been developed during last ten years, including *C. elegans* and mouse models. Multiple DPRs were detectable in many of these models. In C9 ubi and C9 neuro animals, only polyGA is clearly detected. Is the expression levels of RNA in these worms lower than in the other models? Can G4C2 RNA foci be detected in C9 ubi and C9 neuro animals? Do the G4C2 transcripts themselves have effects on phenotypes in UAG ubi and UAG neuro worms compared to Δ C9 ubi and Δ C9 neuro?
3. G4C2 RNAs should be ubiquitously expressed in C9 ubi, polyGA aggregates were observed in the head neurons (Fig. 1F). Can polyGA aggregates be detected in other neurons and non-neuronal cell types?
4. The authors showed that expression of polyGA was decreased in in eif2D-KO; C9 ubi and eif2D-KO; C9 neuro worms. A control animal with 113 nt upstream and 99 nt downstream sequences of G4C2 repeats should be used to test if translation of nLuc reporter can start at CUG codon. If so, can eif2D-KO reduce expression of nLuc reporter in a G4C2 repeat-independent manner?
5. Expression of C9 RNA was significantly increased in eif2D-KO; C9 neuro compared to C9 neuro. These data suggest that upregulation of G4C2 repeat-containing RNA in eif2D-KO may have effects on molecular and behavior phenotypes. Can G4C2 RNA foci be detected or increased in eif2D-KO; UAG neuro or eif2D-KO; UAG ubi compared to UAG neuro or UAG ubi? Are there differences in lifespan and locomotor phenotypes between eif2D-KO; UAG and UAG models with reduced DPRs. It is also important to determine if upregulation of C9 RNA by eif2D is repeat length dependent.
6. In the cell culture experiment, the expression constructs used for transfection are similar to the constructs for developing transgenic worms. PolyGA, polyGP and polyGR were detected in transfected HEK293 cells by using luciferase assay. Are there near-cognate initiation codons in upstream sequences of G4C2 repeats in polyGP and polyGR frames? If so, are these near-cognate start codons required for translation of these DPRs? Can polyGP and polyGR be detected in transfected cells using the protocols used for testing these proteins in worms? It is important to know if overexpression of eif2D regulates RAN translation. Does eif2D-KD or eif2D-overexpression also affect canonical translation initiation?
7. The authors showed that eif2D-KD reduces expression of DPRs, but more detailed mechanistic evidence should be provided to show how eif2D regulates translation of DPRs. Previous study

(Dmitriev et al., 2010) showed that eIF2D is involved in the process of GTP-independent tRNA delivery to P-site of 40 S ribosomes including non-Met (elongator) tRNAs. In addition, Starck et al. (2012) showed an elongator leucine-tRNA was used to initiate protein synthesis at CUG start codons. It's important to figure out if non-Met tRNAs are used for production of C9 DPRs and if the process of tRNA delivery is regulated by eif2D.

8. There is a growing body of evidence showing stress-activated phosphorylation of eIF2 α increases production of DPRs. Can eif2D modulate phosphorylation of eIF2 α in worm and cell culture models?

Minor points:

1. Y axis label is not clear in Fig. 1G and S1B
2. The size of protein ladders is missed in Fig. S1A.
3. In Fig. S1B, are there significant differences?
4. The authors should directly compare differences between C9 ubi and eif-2D9 (-) ; C9 ubi - is this significant?

Reviewer #2 (Remarks to the Author):

A hexanucleotide repeat expansion GGGGCC in the noncoding region of C9orf72 is the most common cause of amyotrophic lateral sclerosis (ALS) and frontotemporal dementia (FTD). Toxicities from the repeat RNAs and/or dipeptide repeats (DPRs), produced by repeat associated non-AUG (RAN) translation, contribute to the disease mechanism. This manuscript developed *C. elegans* models that express a transgene with 75 GGGGCC repeats, either ubiquitously or exclusively in neurons. Mutation of a non-canonical translation-initiating codon (CUG) upstream of the repeats blocked poly-GA production and ameliorated disease, suggesting poly-GA is pathogenic. The authors also showed that the eukaryotic translation initiation factor 2D (eif-2D/eIF2D) was shown to be necessary for RAN translation. Genetic removal of eif-2D increased lifespan in *C. elegans* models, and reduced the DPR levels of all three reading frames from a reporter transgene in the human HEK293 cell line.

Although this study generated a new GGGGCC repeat in vivo model in *C. elegans*, there are several weaknesses. First, there is only description of phenotypes but no mechanistic insights or no genetic screening for genetic modifiers of the toxicity. Meanwhile, there have been many publications using fly models to screen for modifiers and study the disease mechanisms. Second, the CUG codon for GA translation initiation has been shown in human cells previously by multiple groups, which is novel. Finally, although eIF-2D removal was shown to increase the lifespan of *C. elegans* probably by reducing the GA level (though it failed to show GP and GR were not changed), the mechanism was not explored and it was not investigated whether this can also bring beneficial effects in human patient neurons. Overall, the shortcomings of the current manuscript are significant and it cannot be recommended for publication in Nature Communications.

Other major concerns:

1. Figure 1G: need to show GP level on C9 UAG mutant. The manuscript claimed mutation of CUG blocked GA production and ameliorated disease, suggesting poly-GA is pathogenic. However, they did not measure the levels of poly-GP and poly-GR, and there was no evidence to show GP and GR were not affected. Therefore, it cannot support the conclusion that GA is the pathogenic species.
2. The authors identified eIF2D by searching the literature for factors known to initiate translation via the CUG codon. Again, they only measured the GA level but didn't test GP and GR levels. Therefore, the data cannot support GA is pathogenic. In fact, eIF2D knockdown in human 293 cells reduced the levels of all three DPRs (GA, GP and GR) from the repeat RAN translation reporter (Fig 5). This is actually contradictory to what the authors proposed. Furthermore, GP and GR are not in frame of CUG. If eIF2D mediates the CUG translation, why it affects the GP and GR?

3. There is no mechanism on how eIF2D modulate DPR levels. No evidence supporting this requires the CUG start codon. And no evidence supports that the regulation is at the translational level, instead of indirect effects, such as protein stability. In vitro and in vivo translation assays are required.
4. Previous work from Goodman, et al. *Acta Neuropathol Commun* (2019) screened all translation factors in a fly model expressing the GGGGCC repeats. They identified eIF4B and 4H, but didn't find any effects by eIF2D. How to consolidate these results? Does eIF2D actually influence eIF4B and 4H in *C.elegans*?
5. It is not known how deletion of eIF2D influences endogenous global translation. The authors only tested one gene with AUG start codon as control (GFP in *C.elegans* and Firefly luciferase in 293 cells). What about other endogenous genes, particularly ones with CUG translation initiation?
6. It is not known whether knock down of eIF2D in patient neurons can reduce the endogenous DPR levels and rescue the toxicity. As eIF2D showed different effects in *C.elegans* and fly models, it is uncertain whether it can really modulate the endogenous DPR production in patient cells. Furthermore, the function and endogenous targets of eIF2D in human neurons might be very different from *C.elegans*, and might have essential roles in human but not in *C.elegans*. The authors need to test the rescue effect in patient derived iPSC-neurons.

Minor points:

1. The whole manuscript did not examine the RNA foci formed by the repeats.
2. On page 11 line 243, it should be Fig. 4F-G.

Reviewer #3 (Remarks to the Author):

Summary of manuscript

The manuscript details the creation and initial analysis of a new G4C2-repeat model of ALS and FTD in *C. elegans*, with an additional focus on the impact of eIF2D loss of DPR expression and presumably consequent suppression of motor neurons defects in this model. Additional studies are undertaken in cell culture to extend results to mammalian systems. The development of a new model will be warmly welcomed by the field. There are some significant problems with the manuscript, which are described below. These include multiple missing controls, failure to include specific information/results, and incomplete description of methods, results and other materials.

Major Comments

1. The manuscript expresses the GA dipeptide in-frame with nanoluciferase, creating a fusion protein. But, essential controls are lacking; specifically, expression of GA without luc as a C-terminal fusion and expression of luc without GA. Scoring these controls in GA Westerns and in assays for locomotion, MN loss and synaptic protein loss is essential.
2. Clarity regarding where the transgenes are expressed is critical. The authors use 1094bp of the *snb-1* promoter, which was reported to drive ubiquitous expression by Stefanakis, et al. This is not the usual result for *snb-1* promoter constructs, which tend to drive expression predominantly in neurons. And, predominant neuronal expression is seen in this manuscript when GA::nLuc is detected in Figure 1F. While the reviewer does not doubt Stefanakis, transformation markers and relative concentrations of plasmids/DNA molecules can dramatically change expression patterns. At a minimum, the authors need to provide an image that shows the ubiquitous expression of nLuc driven by this promoter in their hands -- using precisely the same transformation markers and relative injection concentrations used for manuscript for other constructs. This is critical for readers to gauge how "ubiquitous" expression is - and the relative level of expression in the nervous system. Additionally, Figure 1F shows expression of GA in unidentified head neurons-- and does not show if GA is expressed in motor neurons, which are scored for survival in other figure panels. The manuscript must show GA expression in motor neurons or admit that they cannot detect it.

This also brings into question the manuscripts assertions regarding RAN translation in neurons; this speculation should be diminished or eliminated, given the high level of neuronal expression in transgenic animals with the snb-1 constructs.

3. New supplemental figures must be added that include an image of entire Western/immunoblot and show markers/size of bands - this is relevant for the GA, nanoLuc, detection

4. Details must be provided for the individual transgenic lines reported. Extrachromosomal lines were presumably scored separately and they should be reported separately in the supp materials, although it is certainly appropriate to merge these in the main text figures, if the lines have essentially the same impact on phenotype for each assay. This table should also provide sufficient detail to provide the reviewer/reader with the number of animals scored for each genotype/age. For example, Figure 2 suggests that motor neurons are lost, but the number of animals scored is unclear beyond ">10 animals scored". Eleven animals would not be enough for any genotype/age and the individual data points should be included here to permit assessment of variation.

5. Have eif-2A(gk358198) and eif-2D(gk904876) been reported in the literature previously? Were these alleles backcrossed prior to these studies (to remove mutations in other genes) and how many times? The impact of background mutations cannot be ruled out with 1) rescue studies, 2) additional alleles with non-complementation studies, or 3) well-controlled RNAi studies. Using one of these approaches is critical and absolutely required to conclude that gene function is required. Given the manuscript discussion focus on eIF2D, it would be ideal to focus on validating this gene.

6. Lifespan assays should be replicated without FUDR; the C. elegans field has numerous examples of FUDR altering lifespan for mutant genotypes. And, citing Kramer et al for lifespan methods is not sufficient; more detail is required. Were animals this manuscript actually reared and aged on RNAi-competent bacteria strains as in Kramer, or was the laboratory standard OP50 used here? How many biological replicates were used and how many animals were used in each of these trials? A data table is required for replicates of lifespan determinations, including median lifespan, number of animals included, number of animals censored, statistical determinations, etc. Individual experiments should be broken out allowing readers to determine which controls are matched with which experimental strains. This is standard in the field. Many other determinations should also be in a supplemental data table or the figure legend, including p-values (example - expression level analyses.) It is no longer sufficient to just label these with asterisks or N.S.

7. It is important for the manuscript to state if experimenters were blinded as to genotype for MN scoring and puncta analysis. Manuscript must also report results in much more detail, including how the GFP synaptic protein expression was quantified. Information required includes, but is not limited to, the number of neurons scored per animal and the number of animals scored for all determinations. Another age point for MN loss would be helpful, but not essential.

8. For the field in general, it is important to know if other types of neurons are affected, including glutamatergic and dopaminergic. Transgenes are available to test this. This would dramatically raise the impact of this model

9. The behavioral analysis presented in Figure 5 is extremely challenging for readers who are not familiar with this specific analysis system, which is not widely used in the field. It's unclear what the parameters described are measuring. This is especially important for the [eif-2D (-); C9 ubi] genotype; it is hard to understand if the defect described in these graphs actually causes impaired locomotion or just subtle changes. All the behavioral metrics presented in the panels A, B and C should be described in a supplementary figure and supported by videos showing the locomotion defect versus control animals. Note that the videos can be at an external site, as long as they have a permanent doi/location.

Minor Comments

A. The reviewer cannot find *unc-11c* in the Stefanakis, et al. reference cited. Please double check that this terminology is correct.

B. References are lacking in multiple places, including references for transgenes and alleles generated in other laboratories.

C. Instead of just stating that *eif-2A(gk358198)* and *eif-2D(gk904876)* are null alleles, the manuscript should at least report what the amino acid change is for each genes (e.g. W146*) to allow readers to understand why the manuscript suggests this is a complete loss of function allele.

D. The reviewer finds it frustrating that no assessments were performed in human cells to address a potential impact of *eif2A* on DPR generation (especially GR and GP). This would strengthen the manuscript as the cell culture system has higher expression of GR and GP, and also would provide a more sensitive context to study the impact of genetic background manipulation on DPRs. However, this lack does not impact the results presented.

E. In line 168 the manuscript describes a behavioral defect. But, these results are not presented until Figure 5, which is referenced in line 256 in the G4C2 suppressor analysis. This should be fixed.

F. At line 403 the manuscript states that "the present study uncovered a requirement for *eif-2D/eIF2D* in RAN translation". However other studies have suggested a role of *eif2D* in RAN translation (ref 19 of this paper) and these should be given more prominence in the introduction and discussion.

G. At line 109, it would be helpful if the manuscript was more precise regarding impact on specific dipeptides, with detail similar to that used at line 401.

Draft

Reviewer #1 (Remarks to the Author):

In this study, Sonobe et al. developed *C. elegans* models that express, either ubiquitously or exclusively in neurons, a transgene with G4C2 repeat expansion. These animals generate DPRs including polyGA and polyGP and display neurodegeneration, locomotor and lifespan defects. Translation initiation of polyGA is at a near-cognate start codon CUG upstream of the G4C2 repeats. Additionally, the authors showed that eukaryotic translation initiation factor 2D (eIF2D/eIF2D) can regulate RAN translation. Genetic removal of eIF2D increased lifespan in both *C. elegans* models. Knock-down of eIF2D inhibits the production of DPRs in a cell culture model. These findings provide additional tools to understand the mechanisms of RAN translation and may contribute to the development of future therapeutic strategies for C9 ALS/FTD. Overall, this is a potentially interesting story, but aspects of the story are not new and additional experiments that provide a more detailed mechanistic insight should be provided to strengthen the impact of this manuscript. Specific concerns and suggestions to improve the potential impact of the study are listed below.

Major concerns:

1. In C9 animals, the plasmids used for developing worm models contain 113 nt upstream and 99 nt of sequence downstream of the G4C2 repeats, followed by nanoluciferase (nLuc) reporter in the reading frame of poly-GA. In methods, the authors described “In Δ C9 ubi animals, the construct is identical but lacks the G4C2 repeats”. Does the construct contain upstream and downstream flanking sequences surrounding the G4C2 repeats? The control animals, Δ C9 ubi, should contain the flanking sequences, but in Fig. 1A and 1B, it seems that this construct does not contain the flanking sequences.

We regret this confusion. We have now edited the manuscript to make it clear that Δ C9^{ubi} animals lack the G4C2 repeats as well as the 113nt upstream and 99nt downstream of the repeats. As expected, Δ C9^{ubi} animals do not produce DPRs and display normal lifespan and locomotion (**Fig. 1-2, new Supplementary Fig. 8-10**). Prompted by the reviewer’s comment, we also generated a new control construct, which contains the upstream 113 nt and downstream 99 nt sequences, but lacks the G4C2 repeat sequences. We microinjected the construct into *C. elegans*, and these new transgenic animals are referred to as Δ G4C2 repeats^{ubi} (**new Supplementary Fig. 2, 6**). Similar to Δ C9^{ubi}, the Δ G4C2 repeats^{ubi} animals do not produce poly-GA and display normal movement. We included these findings in first section of Results entitled “Generation of a new *C. elegans* model...”.

2a. A lot of C9orf72 models have been developed during last ten years, including *C. elegans* and mouse models. Multiple DPRs were detectable in many of these models. In C9 ubi and C9 neuro animals, only polyGA is clearly detected.

In the revised manuscript, we employed additional immunoassays that showed robust expression of poly-GP and modest production of poly-GR in C9^{ubi} animals (**Fig. 1, new Supplementary Fig. 15**). These new results suggest that all three DPRs from the sense transcript (poly-GA, poly-GP, poly-GR) are produced in our C9^{ubi} animals, albeit at different levels. We revised the manuscript accordingly in Results (page 7, lines 170-171).

2b. Is the expression level of RNA in these worms lower than in the other models?

In order to answer this question, we would need to obtain from other labs *C. elegans* and mouse models expressing the G4C2 repeats, and then measure mRNA levels with qRT-PCR (using the same primer set), as well as DPR levels. Although we reached out to other labs, we were unable to obtain *C. elegans* strains that carry a G4C2 construct because these strains were no longer available or lost. Another issue is that current animal models of *C9orf72* vary greatly in construct design from prior published studies (e.g., variable G4C2 repeat length, some models include intronic sequence, others don't, etc.). Hence, a comparison of G4C2 repeat RNA levels between our worms and other models will be challenging. Therefore, we respectfully consider this outside the main scope and focus of the present study.

2c. Can G4C2 RNA foci be detected in C9 ubi and C9 neuro animals?

We carried out RNA fluorescent in situ hybridization (FISH) against the G4C2 repeats and detected RNA foci in C9^{ubi} and C9^{neuro} animals. In both cases, RNA foci were present in head and tail neurons, as well as ventral nerve cord motor neurons (**Fig. 1C, new Supplementary Fig. 3, 13**). Due to their ubiquitously active promoter, the C9^{ubi} transgenic animals also displayed RNA foci in non-neuronal tissues, such as intestine and hypodermis. This was not the case for the C9^{neuro} animals. These results are now provided in page 7 and 12 (1st paragraph).

2d. Do the G4C2 transcripts themselves have effects on phenotypes in UAG ubi and UAG neuro worms compared to Δ C9 ubi and Δ C9 neuro?

We have now conducted an in-depth analysis of the UAG^{ubi} animals:

With RNA FISH, we detected RNA foci in both neurons and non-neuronal cells of UAG^{ubi} animals (**Fig. 1c, new Supplementary Fig. 3, 13**). As expected, no RNA foci were detected in Δ C9^{ubi} animals. With Western blotting and ELISA assays, we assayed the DPR levels in UAG^{ubi} animals. Unlike poly-GP and poly-GR, we found that poly-GA is significantly down-regulated in UAG^{ubi} animals compared to C9^{ubi} worms (**Fig. 1d-h**). Hence, although G4C2 transcripts are detected in UAG^{ubi} animals, the poly-GA levels are significantly reduced.

Survival and locomotion data on UAG^{ubi} animals were provided in our initial submission (**Fig. 2a-b**). Survival is similar in UAG^{ubi} worms when compared to Δ C9^{ubi}. Similarly, an in-depth analysis of locomotion revealed no differences between Δ C9^{ubi} and UAG^{ubi} (**new Supplementary Fig. 8-9**). Because G4C2 transcripts are detected with RT-PCR (**Fig. 1b**) in UAG^{ubi} at levels comparable to C9^{ubi} animals, these findings suggest that the G4C2 transcripts themselves have no detectable effect on the survival and locomotion phenotypes we observe in C9^{ubi} animals (**Fig. 2a-b**).

We made similar observations in UAG^{neuro} animals. The G4C2 transcripts are detected with RT-PCR (**Fig. 3b**) in UAG^{neuro} at levels comparable to C9^{neuro} animals, however, the survival of UAG^{neuro} worms is similar to Δ C9^{neuro} (**Fig. 3f**). All these results are included in the revised manuscript.

3. G4C2 RNAs should be ubiquitously expressed in C9 *ubi*, polyGA aggregates were observed in the head neurons (Fig. 1F). Can polyGA aggregates be detected in other neurons and non-neuronal cell types?

We thank the reviewer for this comment, which prompted us to perform additional key experiments. **First**, RNA FISH showed that G4C2 RNAs are indeed expressed in neuronal and non-neuronal cells in C9 ^{*ubi*} (please see **response 2c** above). **Second**, antibody staining for poly-GA revealed expression in head neurons, as well as ventral nerve cord motor neurons and non-neuronal cells (**Fig. 1i**). **Third**, we generated new *C. elegans* animals, in which we can observe poly-GA accumulation with a GFP reporter. These C9-GFP ^{*ubi*} animals carry the same construct as C9 ^{*ubi*} animals in Figure 1, but the nLuc sequence is replaced with GFP. We observed aggregated GFP puncta (a proxy for poly-GA expression) in head and tail neurons, as well as ventral nerve cord motor neurons and other tissues (e.g., muscle, intestine) (new **Supplementary Fig. 5a-b**). We noticed that GFP accumulation progressively increased with age in these cells of C9-GFP ^{*ubi*} animals. **Fourth**, we generated an additional control: animals that carry the exact same C9 ^{*ubi*} construct, but without the nLuc sequence. Again, we observed poly-GA accumulation in neuronal and non-neuronal cells (new **Supplementary Fig. 4b**). **Lastly**, we generated transgenic reporter animals that express GFP under the control of the same promoter (*snb-1*) and 3'UTR (*unc-54*) sequences used in C9 ^{*ubi*} animals. These animals confirmed that the *snb-1* promoter is indeed active in most, if not all, *C. elegans* cells (new **Supplementary Fig. 4b**), as previously reported (Stefanakis et al., *Neuron*, 2015, PMID: 26291158). We have updated the text in Results to include all these findings.

4. The authors showed that expression of polyGA was decreased in *eif2D*-KO; C9 *ubi* and *eif2D*-KO; C9 neuro worms. A control animal with 113 nt upstream and 99 nt downstream sequences of G4C2 repeats should be used to test if translation of nLuc reporter can start at CUG codon. If so, can *eif2D*-KO reduce expression of nLuc reporter in a G4C2 repeat-independent manner?

We have now made two new transgenic *C. elegans* strains: i). As noted in the response to comment 1 above, the Δ G4C2 repeats ^{*ubi*} animals carry a transgene that has no G4C2 repeats, but is otherwise identical to the transgene of C9 ^{*ubi*} animals in Figure 1A (same promoter and 3'UTR sequences, intronic sequences [113 nt upstream and 99 nt downstream], with nLuc in the reading frame of the upstream CUG). A luciferase assay demonstrated that translation of nLuc reporter can start at CUG codon in these Δ G4C2 repeats ^{*ubi*} animals (page 8, 2nd paragraph, **new Supplementary Fig. 6**).

Moreover, the level of nLuc is decreased by ~84% in *eif-2d* (-); Δ G4C2 repeats ^{*ubi*} animals when compared to Δ G4C2 repeats ^{*ubi*} animals (**new Supplementary Fig. 6**). Although the mRNA levels of both *C9orf72* (intronic) and *nLuc* were also decreased by ~43% in *eif-2d* (-); Δ G4C2 repeats ^{*ubi*} animals compared to Δ G4C2 repeats ^{*ubi*} animals, these findings support the idea that *eif-2D* can control nLuc translation at the CUG codon in a G4C2 repeat-independent manner.

We also generated new transgenic *C. elegans* animals that carry a construct with 5 G4C2 repeats, referred to as C9 5xG4C2 ^{*ubi*} (**new Supplementary Fig. 23**). The nLuc reporter is in the poly-GA frame. Importantly, the RNA levels of intronic *C9orf72* and *nLuc* were similar between C9 5xG4C2 ^{*ubi*} and *eif-2d* (-); C9 5xG4C2 ^{*ubi*} animals; however, a

luciferase assay showed that the level of *nLuc* (indicative of poly-GA expression) was significantly decreased in the *eif-2d* (-); *C9 5xG4C2^{ubi}* animals (**new Supplementary Fig. 23**). These *C9 5xG4C2^{ubi}* results together with the data on *C9^{ubi}* animals that carry 75 *G4C2* repeats (**Fig. 4-5**) strongly suggest that *eif-2D* controls poly-GA translation independent of *G4C2* repeat length. These results are now referred to in page 14 (2nd paragraph).

5a. Expression of *C9* RNA was significantly increased in *eif2D-KO*; *C9* neuro compared to *C9* neuro. These data suggest that upregulation of *G4C2* repeat-containing RNA in *eif2D-KO* may have effects on molecular and behavior phenotypes. Can *G4C2* RNA foci be detected or increased in *eif2D-KO*; *UAG* neuro or *eif2D-KO*; *UAG ubi* compared to *UAG* neuro or *UAG ubi*?

To address this point, we carried out RNA FISH to detect *G4C2* mRNA foci in *UAG^{ubi}* and *eif2D-KO*; *UAG^{ubi}* animals. We indeed detected RNA foci of *G4C2* mRNA in neurons and non-neuronal cells of *UAG^{ubi}* and *eif2D-KO*; *UAG^{ubi}* animals (**new Supplementary Fig. 13**). Although we cannot exclude the possibility that the RNA foci in *eif2D-KO*; *UAG^{ubi}* animals can cause molecular phenotypes, our new experiments did not detect any deleterious behavioral effects on locomotion and lifespan of these worms.

In contrast, the *C9^{ubi}* animals show RNA foci, produce DPRs, and display defects in locomotion and lifespan, as well as motor neuron cell loss (**Figures 1, 2, 5**). Because the *eif2D-KO*; *C9^{ubi}* animals display normal locomotion (**new Supplementary Fig. 19**) and lifespan (**new Supplementary Fig. 17, 21**) despite producing RNA foci (**new Supplementary Fig. 13**), we attribute this amelioration of defects to the reduced DPR production we observed in these animals (page 13, 1st paragraph).

5b. Are there differences in lifespan and locomotor phenotypes between *eif2D-KO*; *UAG* and *UAG* models with reduced DPRs.

We generated *eif2D-KO*; *UAG^{ubi}* animals and conducted a lifespan assay together with $\Delta C9^{ubi}$, *UAG^{ubi}*, *C9^{ubi}*, *eif2D-KO*; *C9^{ubi}* animals (**new Supplementary Fig. 17**). The assay was repeated twice. We found no significant difference in the lifespan of *eif2D-KO*; *UAG^{ubi}* and *UAG^{ubi}* worms ($P = 0.0640$). Similar results were obtained when we assessed the lifespan of $\Delta C9^{neuro}$, *UAG^{neuro}*, *C9^{neuro}*, *eif2D-KO*; *C9^{neuro}*, *eif2D-KO*; *UAG^{neuro}* animals (**new Supplementary Fig. 21**). We also carried out automated worm tracking locomotion analysis and observed no significant locomotion defects in *eif2D-KO*; *UAG^{ubi}* animals compared to $\Delta C9^{ubi}$ and *UAG^{ubi}* controls (**new Supplementary Fig. 19**).

5c. It is also important to determine if up-regulation of *C9* RNA by *eif2D* is repeat length dependent.

The increase in *C9orf72* mRNA was only observed in *eif-2d* (-); *C9^{neuro}* animals (**Figure 4f**). RNA FISH experiments also showed *G4C2* mRNA foci are detected both in *C9^{neuro}* and *eif-2d* (-); *C9^{neuro}* animals (**new Supplementary Fig. 13**).

We do not think the *C9orf72* mRNA in *eif-2d* (-); *C9^{neuro}* animals is related to the repeat length because of the following data:

- (a) *C9orf72* and nLuc mRNA levels are not significantly upregulated in *eif-2d* (-); *C9^{ubi}* compared to *C9^{ubi}* animals (**Figure 4a-b**).
- (b) With RT-PCR, we found no significant up-regulation in the levels of *C9orf72* intronic mRNA in *eif-2d* (-) animals carrying transgenes with no repeats (Δ G4C2 repeats *ubi*) 5 repeats *C9 5xG4C2^{ubi}*, or 75 repeats (UAG *ubi*, UAG *neuro*) (**new Supplementary Fig. 6, 14, 18, 23**).

6a. In the cell culture experiment, the expression constructs used for transfection are similar to the constructs for developing transgenic worms. PolyGA, polyGP and polyGR were detected in transfected HEK293 cells by using luciferase assay. Are there near-cognate initiation codons in upstream sequences of G4C2 repeats in polyGP and polyGR frames? If so, are these near-cognate start codons required for translation of these DPRs?

In the **GP** reading frame, we found 3 near-cognate initiation codons in the 113 nt-long intronic sequence we used in our constructs: a CUG codon (83 nt) and two AGG codons 53 nt and 56 nt upstream of the G4C2 repeat. However, there is an UAG stop codon 2 nt upstream of the G4C2 repeat in the GP reading frame, preventing expression of a poly-GP protein from any of these three upstream near-cognate initiation codons. The presence of this UAG suggests poly-GP translation is initiated within the G4C2 repeat in our transgenic worms and cell culture system.

In the **GR** reading frame, Boivin et al. (EMBO J 39: e100574 2020, PMID: 31930538) demonstrated that an AGG located 1 nt upstream is used for poly-GR translation initiation. Thus, translation initiation of poly-GR in our transgenic worms and cell culture system may occur at this AGG.

In the **GA** reading frame, we and other labs have shown that a CUG located 24 nt upstream of the repeat is required for poly-GA translation (Neurobiol Dis 116: 155-165 2018, PMID: 29792928; Nat Commun 8: 2005 2017; PMID: 29222490; Nat Commun 9 152 2018, PMID: 29323119; EMBO J 39, e100574 2020. PMID: 31930538). Moreover, when we transfected HEK293 cells with the bicistronic construct shown in **Figure 6e**, but with a mutation in this CUG codon (CUG > UAG), a significant decrease in poly-GA occurred with no effect on poly-GP and poly-GR (**new Supplementary Fig. 5d**). We also evaluated in *C. elegans* the effect of the CUG (24 nt upstream of the repeat) to UAG mutation on DPR production (**Fig. 1d-h**). In these worms (*UAG^{ubi}*), we found a decrease in poly-GA, but no significant effects on poly-GP and poly-GR production.

We acknowledge in the Discussion (pages 19-20) the possibility of a frameshift used in DPR translation, as found by other investigators using a different construct (Nat Commun 9, 152 2018, PMID: 29323119); however, our results in HEK293 and *C. elegans* strongly suggest that the upstream CUG is used for translation initiation of poly-GA.

6b. Can polyGP and polyGR be detected in transfected cells using the protocols used for testing these proteins in worms?

We have now carried out ELISA assays on transgenic worms and transfected HEK293 cells. We were able to detect poly-GP and poly-GR in both worms (**Fig. 1g-h**) and HEK293 cells

(**Fig. 6j-k**). We have now provided these data in the Results section entitled “Generation of a new *C. elegans* model...” and “Knockdown of *eIF2D* suppresses...”.

6c. It is important to know if overexpression of *eif2D* regulates RAN translation.

To address this, we generated a construct with the CMV promoter driving expression of human *eIF2D* tagged with the V5 epitope. In a control construct, we swapped *eIF2D* with GFP. Each of these constructs was transfected into HEK293 cells along with the C9 bicistronic constructs (nLuc in frame of GA, GP, or GR in the second cistron) shown in **Figure 6e**. The luciferase assay showed that overexpression of *eIF2D* significantly upregulated the expression levels of poly-GA and poly-GR (**new Supplementary Fig. 26**), suggesting *eIF2D* overexpression can increase RAN translation. We now describe these findings in page 16 (1st paragraph).

6d. Does *eif2D*-KD or *eif2D*-overexpression also affect canonical translation initiation?

In *C. elegans*, we found that genetic removal of *eif-2d* does not affect expression of an AUG-dependent transgenic GFP marker (**new Supplementary Fig. 22**). We also investigated worms carrying endogenous reporter alleles for two transcription factors (UNC-3, CFI-1) known to be expressed in *C. elegans* ventral nerve cord motor neurons (Patel et al, eLife, 2017, PMID: 28422646; Li et al., eLife, 2020, PMID: 33001031). These alleles generate protein fusions, i.e., UNC-3 is fused to GFP and CFI-1 is fused to mNeonGreen, allowing the visualization of motor neurons expressing these reporter alleles. We found no difference in the number of motor neurons expressing UNC-3::GFP or CFI-1::mNeonGreen between wild-type and *eif-2d* mutant worms (**new Supplementary Fig. 22**). These findings together with the normal lifespan and locomotion of *eif-2d* mutant worms suggest that *eIF2D* does not broadly affect canonical AUG-dependent translation. However, we cannot exclude the possibility that *eIF2D* may be required for the translation of other AUG-initiated transcripts. We now present these results and acknowledge this possibility in page 13 (last paragraph).

In HEK293 cells, we used an *eIF2D* shRNA that knocked down *eIF2D* protein levels by ~50% (**Figure 6f-g**), and importantly led to a reduction in DPR production (**Figure 6h-i**) with unaltered protein levels of alpha-tubulin and fLuc (**Figure 6f, I**). These results suggest *eIF2D* knockdown affects DPR production but not canonical AUG-dependent translation.

7. The authors showed that *eif2D*-KD reduces expression of DPRs, but more detailed mechanistic evidence should be provided to show how *eif2D* regulates translation of DPRs. Previous study (Dmitriev et al., 2010) showed that *eIF2D* is involved in the process of GTP-independent tRNA delivery to P-site of 40 S ribosomes including non-Met (elongator) tRNAs. In addition, Starck et al. (2012) showed an elongator leucine-tRNA was used to initiate protein synthesis at CUG start codons. It's important to figure out if non-Met tRNAs are used for production of C9 DPRs and if the process of tRNA delivery is regulated by *eif2D*. We thank the reviewer for this point. In the initial submission, we showed that *eif-2D* genetic removal in the worm (**Fig. 4**) and *eIF2D* knockdown in HEK293 cells reduces poly-GA production (**Fig. 6**). We now provide new data in NSC-34 motor neuron-like cells, in which

eIF2D knockdown reduced poly-GA expression (**Fig. 6a-d**).

As mentioned in Response **6a** (above), poly-GA translation is initiated from an upstream CUG (-24nt from G4C2) and poly-GR translation is initiated from an AGG (-1nt from G4C2). A recent study showed that **Met-tRNA** is used for initiation of poly-GA translation at the CUG (-24nt), as well as for initiation of poly-GR translation at the AGG (-1nt) (Boivin et al. EMBO J 39: e100574, 2020 PMID: 31930538). The initiation codon for poly-GP remains unknown.

Dmitriev et al. 2010 (PMID: 20566627) previously showed that both Met-tRNA and non-Met tRNA are delivered by eIF2D to the P site of the 40 S ribosomal subunit during translation initiation at the hepatitis C virus (HCV) internal ribosome entry site (IRES) in HEK293 cells. Cryo-electron microscopy results showed that the SUI1 domain of eIF2D contacts the codon-anticodon duplex on the HCV IRES, and also contacts Met-tRNA, presumably stabilizing its orientation (Cell 159:597-607 2014 PMID: 25417110; Proc Natl Acad Sci USA 115 E4159-E4168 2018, PMID: 29666249). The SUI1 domain is also found in the translation initiation factor eIF1, where it plays a role in initiator codon recognition (PMID:18593708; PMID:12435632; PMID:1729602; PMID:16362046). It has been proposed that eIF2D may act by changing the conformation of the 40S ribosomal subunit in a manner similar to eIF1, but unlike eIF1, eIF2D stabilizes tRNA binding within the P-site (Dmitriev et al. 2010, PMID: 20566627).

Based on these previous studies, we hypothesized that the SUI1 domain of eIF2D is required for DPR translation. To test this and provide mechanistic insights, we employed CRISPR/Cas9 genome editing in *C. elegans* and selectively replaced the SUI1 domain with a 3xFLAG epitope at the endogenous *eif-2d* locus. We observed *in vivo* a significant down-regulation of poly-GA expression in homozygous animals that lack the SUI domain (termed *eif-2D* Δ^{SUI1}), albeit they did express the expected truncated eIF2D protein at normal levels. Furthermore, we used animals carrying a point mutation within the SUI1 domain that changes glutamine at amino acid position 512 to a STOP codon. Again, we observed a reduction in poly-GA expression similar to that seen in *eif-2d* Δ^{SUI1} animals (new **Figure 5**). These genetic experiments strongly suggest the SUI1 domain has a key role in eIF2D's action in RAN translation of DPRs. These results are now provided in a new section of Results entitled "The SUI1 domain of eif-2D/eIF2D is required for poly-GA translation in *C. elegans*".

8. There is a growing body of evidence showing stress-activated phosphorylation of eIF2 α increases production of DPRs. Can eif2D modulate phosphorylation of eIF2 α in worm and cell culture models?

Our *in vivo* experiments with the *C. elegans* model showed that the level of p-eIF2 α is similar between wild-type (N2 strain), *eif-2D* (-), $\Delta C9^{ubi}$, $C9^{ubi}$, and *eif-2D* (-); $C9^{ubi}$ animals (new **Supplementary Fig. 27**), suggesting *eif-2D* does not modulate phosphorylation of eIF2 α in the worm.

It is known that sodium arsenite (a known ER stressor) increased DPR production in HEK293 cells transfected with a G4C2 construct that was similar to $C9^{ubi}$ (Green et al. Nat Commun 2018, PMID: 29222490; Cheng et al. Nat Commun 2018, PMID: 29302060). We have now

carried out *in vitro* studies showing that eIF2D knockdown does not affect the level of p-eIF2 α in HEK293 cells following administration of sodium arsenite (new **Supplementary Fig. 28**).

Altogether, these findings suggest *eif-2D*/eIF2D controls DPR translation without altering the levels of p-eIF2 α . We present these findings in the last section of the Results.

Minor points:

1. Y axis label is not clear in Fig. 1G and S1B

This comment referred to ELISA measurements. We have now changed the Y axis label from "Fold change" to "Arbitrary Units" (a.u) in all ELISA assays

2. The size of protein ladders is missed in Fig. S1A.

For all Western blots, we now provide an image of the entire gel with the size of protein ladders in Source Data File.

3. In Fig. S1B, are there significant differences?

The original Fig S1B showed that poly-GR is not detectable in worms carrying 75 G4C2 repeats. We now have used a more sensitive ELISA assay and were able to detect poly-GR, albeit at low levels in C9^{ubi} animals. New figures showing that poly-GP and poly-GR are detected in C9^{ubi} worms (**Fig. 1g-h**) and HEK293 cells (**Fig. 6j-k**) are now included.

4. The authors should directly compare differences between C9 ubi and eif-2D (-); C9 ubi - is this significant?

In the revised figures, we directly compare C9^{ubi} and *eif-2D* (-); C9^{ubi} in all instances.

Reviewer #2 (Remarks to the Author):

A hexanucleotide repeat expansion GGGGCC in the noncoding region of C9orf72 is the most common cause of amyotrophic lateral sclerosis (ALS) and frontotemporal dementia (FTD). Toxicities from the repeat RNAs and/or dipeptide repeats (DPRs), produced by repeat associated non-AUG (RAN) translation, contribute to the disease mechanism. This manuscript developed *C. elegans* models that express a transgene with 75 GGGGCC repeats, either ubiquitously or exclusively in neurons. Mutation of a non-canonical translation-initiating codon (CUG) upstream of the repeats blocked poly-GA production and ameliorated disease, suggesting poly-GA is pathogenic. The authors also showed that the eukaryotic translation initiation factor 2D (eif-2D/eIF2D) was shown to be necessary for RAN translation. Genetic removal of eif-2D increased lifespan in *C. elegans* models, and reduced the DPR levels of all three reading frames from a reporter transgene in the human HEK293 cell line.

Although this study generated a new GGGGCC repeat in vivo model in *C. elegans*, there are several weaknesses. First, there is only description of phenotypes but no mechanistic insights or no genetic screening for genetic modifiers of the toxicity. Meanwhile, there have been many publications using fly models to screen for modifiers and study the disease mechanisms.

We thank the reviewer for this comment, as it gave us the opportunity to improve the manuscript by designing new mechanistic experiments and to more clearly present the novelty of our findings. With respect to the question about mechanism, please see our answer to **Reviewer 1, query 7**. As the reviewer mentioned, we generated a new GGGGCC repeat model in *C. elegans*, which we have now characterized in depth at the molecular, cellular, and behavioral level, resulting in **6 main Figures** and **30 Supplementary Figures**. This model enabled us to discover a new role for *eif-2d*/eIF2D in RAN translation of DPRs. Intriguingly, lack of *eif-2d* gene activity in worms reversed the lifespan and locomotion defects of *C9^{ubi}* and/or *C9^{neuro}* animals (**Figures 4-5**). The present study is focused on establishing a new *C. elegans* model and identifying *eif-2d*/eIF2D as a new player in RAN translation. As suggested by the reviewer, we plan to use this *C. elegans* model in a genetic screen for modifiers, but respectfully consider such an endeavor outside the scope of the present study.

Second, the CUG codon for GA translation initiation has been shown in human cells previously by multiple groups, which is novel... the mechanism was not explored... and it was not investigated whether this can also bring beneficial effects in human patient neurons. We agree our finding of translation initiation of poly-GA from an upstream CUG has been found in other experimental systems (Neurobiol Dis 116: 155-165 2018, PMID: 29792928; Nat Commun 8: 2005 2017, PMID: 29222490; Nat Commun 9 152 2018, PMID: 29323119; EMBO J 39, e100574 2020, PMID: 31930538). Extending these previous observations, our study evaluates *in vivo* the functional significance of mutating this CUG. Nematodes carrying a CUG > UAG mutation display G4C2 RNA foci (**Fig. 1c**, new **Supplementary Fig. 3, 13**), but their ability to produce poly-GA is significantly compromised (**Fig. 1d-e**). Moreover, these animals (*UAG^{ubi}*) display a normal lifespan and improved locomotion compared to *C9^{ubi}* animals (**Fig. 2a-b**, new **Supplementary Fig. 7-9**). Altogether,

these findings suggest poly-GA RAN translation of the 75 G4C2 repeats from this CUG plays a key role in the worm's survival and locomotion. We include these new data in pages 8-9.

Furthermore, we show in *C. elegans* that *eif-2D* genetic removal significantly decreased poly-GA production and improved the lifespan and locomotion defects of *C9^{ubi}* animals (**Fig. 4-5**, new **Supplementary Fig. 8-9, 16-17**). Importantly, we found in two different cell lines (human HEK293, mouse NSC-34) that eIF2D knockdown affected poly-GA production (**Fig. 6**), suggesting a conserved role for eIF2D.

Overall, the shortcomings of the current manuscript are significant and it cannot be recommended for publication in Nature Communications.

In the revised manuscript, we have included a substantial amount of new data that address the issues raised by the reviewers. Compared to the original submission, every main Figure contains additional data, and we also provide 30 new supplementary figures. We feel these new findings have significantly improved the manuscript and strengthened our conclusions. We thank the reviewers for their constructive comments.

Other major concerns:

1. Figure 1G: need to show GP level on *C9* UAG mutant. The manuscript claimed mutation of CUG blocked GA production and ameliorated disease, suggesting poly-GA is pathogenic. However, they did not measure the levels of poly-GP and poly-GR, and there was no evidence to show GP and GR were not affected. Therefore, it cannot support the conclusion that GA is the pathogenic species.

In the revised manuscript, we include new immunoassay results showing robust expression of poly-GP and modest production of poly-GR in *C9^{ubi}* animals (**Fig. 1g-h**). These findings suggest that all three DPRs from the sense transcript (poly-GA, poly-GP, poly-GR) are produced in our *C9^{ubi}* animals. We also found that production of poly-GA (but not poly-GP or poly-GR) is affected in the UAG *ubi* animals, suggesting poly-GA is the pathogenic species.

Of note, we detected RNA foci in *C9^{ubi}* animals (**Fig. 1c**, new **Supplementary Fig. 3, 13**). Hence, we cannot exclude the possibility that these foci could also contribute to the observed toxicity in *C9^{ubi}* animals; however, we found that the UAG *ubi* animals that carry the CUG > UAG mutation produce RNA foci (**Fig. 1c**, new **Supplementary Fig. 3, 13**), poly-GP, and poly-GR, but lack poly-GA (**Fig. 1d-h**). Because these animals show a normal lifespan and no locomotion defects (**Fig. 2a-b**, new **Supplementary Fig. 7-9**), these findings support the idea that poly-GA is a pathogenic DPR in our worm model (pages 8-9).

2a. The authors identified eIF2D by searching the literature for factors known to initiate translation via the CUG codon. Again, they only measured the GA level but didn't test GP and GR levels. Therefore, the data cannot support GA is pathogenic. In fact, eIF2D knockdown in human 293 cells reduced the levels of all three DPRs (GA, GP and GR) from the repeat RAN translation reporter (Fig 5). This is actually contradictory to what the authors proposed.

We agree with the reviewer and regret any confusion. In the revised manuscript, we changed the text to clarify the role of eIF2D based on new experiments detailed below.

We have now measured GA, GP and GR in *C9^{ubi}* animals, and witnessed robust

expression of GA and GP, but low expression of GR in these worms (**Fig. 1d-h**). Importantly, we found that GA is dramatically reduced in *eif-2* (-); *C9^{ubi}* animals (**Fig. 4-5**), whereas GP is only modestly decreased (new **Supplementary Fig. 15**). We did not observe any GR decrease in *eif-2D* (-); *C9^{ubi}* animals (new **Supplementary Fig. 15**). We also observed no significant effects on the *C9orf72* mRNA levels in *eif-2D* (-); *C9^{ubi}* animals (**Fig. 4-5**). Hence, the *eif-2D* (-); *C9^{ubi}* animals produce *C9orf72* mRNA and a small amount of poly-GR, but display a strong decrease in poly-GA and a modest reduction in poly-GP, which likely explain the amelioration of lifespan and locomotion defects of these animals when compared to *C9^{ubi}*.

The relatively small effect on GP with *eif-2D* knockout suggests that, in addition to eIF2D, other factors may be involved in GP translation in *C. elegans*. A promising candidate is DENR, another initiation factor shown in a recent fly study to act redundantly with eIF2D during development and integrated stress response (Vasudevan et al., Nat Comms, 2020, PMID: 32938929). Please, see the new Discussion section entitled “Multiple factors are involved in DPR translation from G4C2 nucleotide repeats”.

The prominent effect on GA with *eif-2* knockout suggests eIF2D is essential for poly-GA translation in *C. elegans*. To extend our *C. elegans* findings to mammalian cells, we transfected mouse NSC-34 motor neuron-like cells with a **monocistronic** 75xG4C2 construct (**Fig. 6a-d**), which is very similar to the **monocistronic** construct used to generate the *C9^{ubi}* animals (**Fig. 1**). Knockdown of eIF2D in NSC-34 cells also resulted in reduced poly-GA expression, as presented in Results section “Knock-down of *eIF2D* suppresses DPR...”.

Our eIF2D data with monocistronic 75xG4C2 constructs in *C. elegans* (**Fig. 4-5**) and NSC-34 cells (**Fig. 6a-d**) are complemented with our *in vitro* data in HEK293 cells transfected with a **bicistronic** construct carrying 75 G4C2 repeats (**Figure 6e-l**). In these cells, we can robustly detect all three DPRs (GA, GP, GR) (**Fig. 6f-k**). Similar to our findings in *C. elegans eif-2D* mutants, eIF2D knockdown decreased poly-GA and poly-GP (**Figure 6h-i**), whereas eIF2D overexpression increased DPR production (**Supplementary Fig. 26**). Importantly, the protein levels of fLuc, which is expressed by the first cistron via canonical AUG-dependent translation, were similar in cells transfected with either control or eIF2D shRNAs (**Fig. 6l**), suggesting a specific role for eIF2D in non-canonical translation in human cells. Interestingly, we also observed an effect on poly-GR upon eIF2D knockdown in HEK293 cells transfected with the bicistronic construct (**Fig. 6i**). We surmise the way DPR translation is initiated in this bicistronic construct is likely quite different from the monocistronic constructs used in the worm and NSC-34 cells, and from the endogenous human C9ORF72 locus. Consistent with this notion, robust poly-GR levels are present in these HEK293 cells (**Fig. 6k**), but very low levels of poly-GR are present in our worm model (**Fig. 1h**) despite using the same detection method (ELISA). Taken together, our *in vivo* (*C. elegans*) and *in vitro* (NSC-34, HEK293) data uncovered a conserved role for eIF2D in controlling RAN translation of at least one DPR (poly-GA). Please, see Results section entitled “Knock-down of *eIF2D* suppresses DPR...”.

2b. Furthermore, GP and GR are not in frame of CUG. If eIF2D mediates the CUG translation, why it affects the GP and GR?

Please, see our responses to Reviewer 1, queries **6a** and **7**. In the GA reading frame, we and

other labs have shown that a CUG located 24 nt upstream of the repeat is required for poly-GA translation (Neurobiol Dis 116: 155-165 2018, PMID: 29792928; Nat Commun 8: 2005 2017, PMID: 29222490; Nat Commun 9 152 2018, PMID: 29323119; EMBO J 39, e100574 2020 PMID: 31930538). Moreover, when we transfected HEK293 cells with a 75xG4C2 construct that has a mutation in this CUG codon (CUG > UAG), a selective effect on poly-GA was observed but with no significant decrease in poly-GP and poly-GR (new **Supplementary Fig. 5d**). However, we cannot eliminate the possibility of a frameshift used in DPR translation as found by other investigators using a different G4C2 construct (Nat Commun 9, 152 2018, PMID: 29323119).

We also evaluated in *C. elegans* the effect of the CUG (24 nt upstream of the repeat) to UAG mutation on DPR production (new Suppl. Fig. 17). In these worms (*UAG^{ubi}*), we found a decrease in poly-GA, but no significant effect on poly-GP and poly-GR production. Altogether, our results strongly suggest that the upstream CUG is used for translation initiation of poly-GA.

Our new findings in *C. elegans* and mammalian cells (NSC-34, HEK293) suggest that the role of eIF2D in poly-GA translation is evolutionarily conserved. Hence, we have now focused the paper on this important finding.

A recent study showed that Met-tRNA is used for initiation of poly-GA translation at the CUG (-24nt) (Boivin et al. EMBO J 39: e100574, 2020 PMID: 31930538). Moreover, Dmitriev et al. 2010 (PMID: 20566627) showed that Met-tRNA can be delivered by eIF2D to the P site of the 40 S ribosomal subunit during translation initiation. In Discussion (pages 21-22), we suggest a model in which eIF-2D acts at the CUG codon to deliver the Met-tRNA (necessary for poly-GA initiation) to the P site of the 40 S ribosomal subunit. We also acknowledge that eIF2D has additional functions besides translation initiation. For example, eIF2D was implicated in recycling 40S ribosomal subunits, which enables ribosomes and mRNAs to participate in multiple rounds of translation, leading to efficient protein expression.

As correctly pointed out by the reviewer, GP and GR are not in frame with the CUG initiation codon and our new experiments with *eif-2D* mutant worms showed either a small effect (GP) or no effect (GR). Hence, we no longer claim that eIF2D controls all three DPRs (GA, GP, GR) of the sense transcript, and have modified the text accordingly. Our new data suggest the importance of eIF2D in translation of poly-GA, but also suggest that a combination of translation initiation factors are necessary for translation of the DPRs (please see Discussion – pages 22-23).

3. There is no mechanism on how eIF2D modulate DPR levels. No evidence supporting this requires the CUG start codon. And no evidence supports that the regulation is at the translational level, instead of indirect effects, such as protein stability. In vitro and in vivo translation assays are required.

The effects we observe in *C. elegans* when we mutate the CUG codon (*UAG^{ubi}* animals), or *eif-2D* itself (*eif-2D* (-); *C9^{ubi}* animals) are similar. In brief, poly-GA is markedly affected (**Fig. 1d-e, 4c-d**), and lifespan/locomotion phenotypes are ameliorated compared to *C9^{ubi}* animals (**Fig 2a-b, new Supplementary Fig. 7-9**). Regarding a possible mechanism for eIF2D action, please see our response to Reviewer 1, query 7. In brief, we now provide *in vivo* evidence in

C. elegans that the SUI1 domain (critical for initiator codon recognition) of *eif-2D* is required for DPR translation (**new Fig. 5**). Of note, the SUI1 domain is thought to stabilize the codon:anticodon duplex by stabilizing the orientation of Met-tRNA during translation (Cell 159:597-607 2014 PMID: 25417110, Proc Natl Acad Sci USA 115 E4159-E4168 2018 PMID: 29666249).

In our response to Reviewer 1, query 4, we present new results that support the idea that *eif-2D* can control nLuc translation initiated at the CUG codon in a G4C2 repeat-independent manner. These experiments also suggest that *eif-2D* controls poly-GA translation independent of G4C2 repeat length.

We have conducted new experiments (summarized here) at the mRNA and protein levels supporting the notion of eIF2D acting at the translational level: **a)** In our *C9^{ubi}* animals, *eif-2D* gene knockout decreased DPR (e.g., poly-GA) translation but did not affect *C9* mRNA levels nor formation of RNA foci, suggesting that the decrease in DPR levels occurs at the translational level (**Fig. 4-5, new Supplementary Figure 13**). **b)** To address a concern raised by reviewer 3 (please see query 5c), we supplemented *eif-2D* in *eif-2D* (-); *C9^{ubi}* animals. Compared to *C9^{ubi}*, this increased the levels of poly-GA (partial rescue), but had no effect on the *C9orf72* and *nLuc* mRNAs levels (**new Supplementary Figure 14**). **c)** We generated new transgenic *C. elegans* animals that carry a construct with 5 G4C2 repeats, referred to as *C9 5xG4C2^{ubi}* (**new Supplementary Figure 23**). The nLuc reporter is in the poly-GA frame. Importantly, the mRNA levels of intronic *C9orf72* and *nLuc* were similar between *C9 5xG4C2^{ubi}* and *eif-2d* (-); *C9 5xG4C2^{ubi}* animals; however, a luciferase assay showed that the level of nLuc (indicative of poly-GA expression) was significantly decreased in the *eif-2d* (-); *C9 5xG4C2^{ubi}* animals.

To evaluate possible indirect effects mediated by eIF2D, we **first** interfered with the ability of the cell to degrade proteins via the proteasomal pathway. We co-transfected HEK293 cells with the G4C2 construct shown in Figure 6e and an shRNA against eIF2D. Upon administration of the proteasome inhibitor MG132, the downregulation of eIF2D was still able to reduce poly-GA translation (**new Supplementary Figure 29**), suggesting protein degradation via the proteasomal pathway is not associated with the observed reduction in DPR expression upon eIF2D knockdown. **Second**, please see our response to reviewer 1 (query 8). Therein, we provide evidence that eIF2D does not modulate phosphorylation of eIF2 α , a stress-activated protein involved in DPR production.

We have significantly modified the text to include all the above results, and in Discussion (page 22) we also acknowledge the possibility that the observed effects of eIF2D in DPR production could occur through yet-to-be identified functions.

4. Previous work from Goodman, et al. Acta Neuropathol Commun (2019) screened all translation factors in a fly model expressing the GGGGCC repeats. They identified eIF4B and 4H, but didn't find any effects by eIF2D. How to consolidate these results?

The study by Goodman et al. generated a fly model with G4C2 repeats and performed an RNAi screen for canonical translation initiation factors responsible for poly-GR expression (cited in Introduction, page 4, 1st paragraph). The study identified eIF4B and eIF4H, two accessory proteins that increase the helicase activity of eIF4A and are known to interact

directly with G4C2 mRNA. The most parsimonious explanation for not recovering eIF2D from this RNAi screen is that the RNAi clone used may have not downregulated eIF2D at levels sufficient to observe an effect on poly-GR. In the revised manuscript, we include new experiments using 3 strong loss-of-function *eif-2d* alleles (two cause a premature STOP, and one selectively eliminates the SUI domain within endogenous *eif-2D*). In all cases, we observed a dramatic reduction in poly-GA expression. We also note that *C. elegans* animals lacking *eif-2D* gene activity displayed a small reduction in poly-GP, but no effect was observed on poly-GR (new **Supplementary Figure 15**). Based on these results, we suggest in Discussion that a number of different factors may be involved in translation of the DPRs (pages 22-23).

Relevant to the reviewer's question is a recent publication in *Drosophila* (cited in Discussion – page 23, 1st paragraph) showing that eIF2D functions redundantly with DENR during development and the integrated stress response (Nat Commun 11: 4677 2020, PMID: 32938929), suggesting that the function of eIF2D can be compensated by DENR depending on the levels of eIF2D knockdown. Lastly, the G4C2 construct used in the Goodman study was different from the ones used in our study since it had no downstream intronic sequence and a different number of G4C2 repeats. Importantly, eIF4B and eIF4H enhance helicase activity of eIF4A, which enables recruitment of the ribosome to the mRNA. This mechanism of action appears different from that of eIF2D, which is associated with the stability of the codon-anticodon duplex and translation initiation. Please see response to **Reviewer 1**, query 7.

Does eIF2D actually influence eIF4B and 4H in *C. elegans*?

Due to the lack of antibodies specific to HRPF-2 (*C. elegans* ortholog of eIF4B) and DDR-2 (*C. elegans* ortholog of eIF4H), we looked at mRNA levels using RT-PCR. Compared to wild-type animals, we found no effect on *hrpf-2*/eIF4B or *ddr-2*/eIF4H mRNA levels in animals carrying a loss-of-function *eif-2d* (*gk904876*) allele (new **Supplementary Figure 30**). Conversely, we found that the protein levels of eIF-2D are unaffected in animals carrying a deletion (putative null) allele for *hrpf-2*/eIF4B (*gk5105*). Confirmed loss-of-function alleles for *ddr-2*/eIF4H are not available in *C. elegans*. These results are noted in the Discussion (page 23, 1st paragraph).

5. It is not known how deletion of eIF2D influences endogenous global translation. The authors only tested one gene with AUG start codon as control (GFP in *C. elegans* and Firefly luciferase in 293 cells). What about other endogenous genes, particularly ones with CUG translation initiation?

Please see our response to Reviewer 1, query **6d**. The response refers to our new experiments showing the expression of several genes that use AUG as a start codon is not affected by *eif-2D*/eIF2D. We note that we were unable to identify in the *C. elegans* literature endogenous genes that use CUG for translation initiation.

Moreover, the effects we observe in *C. elegans* when we mutate the CUG codon (*UAG^{ubi}* animals), or *eif-2D* itself (*eif-2D* (-); *C9^{ubi}* animals) are similar. In brief, poly-GA is

markedly affected, and lifespan/locomotion phenotypes are ameliorated compared to $C9^{ubi}$ animals (**Fig. 1d-e, 2a-b, 4c-d, New Supplementary Fig. 7-9**). Altogether, our results support the idea that eIF2D is important for CUG-initiated RAN translation of poly-GA.

6. It is not known whether knock down of eIF2D in patient neurons can reduce the endogenous DPR levels and rescue the toxicity. As eIF2D showed different effects in *C. elegans* and fly models, it is uncertain whether it can really modulate the endogenous DPR production in patient cells. Furthermore, the function and endogenous targets of eIF2D in human neurons might be very different from *C. elegans*, and might have essential roles in human but not in *C. elegans*. The authors need to test the rescue effect in patient derived iPS-neurons.

Thank you for this comment. Regarding the different eIF2D effects in worm and fly G4C2 models, please see our response above (Reviewer 2, query 4).

Summary of the role of *eif-2D* in *C. elegans*: We have now measured GA, GP and GR in $C9^{ubi}$ animals, and witnessed robust expression of GA and GP, but low expression of GR in these worms (**Fig. 1d-h**). We found that GA is dramatically reduced in *eif-2D* (-); $C9^{ubi}$ animals (**Fig. 4**), whereas GP is only modestly decreased (new **Suppl. Fig. 15**). We did not observe any GR decrease in *eif-2D* (-); $C9^{ubi}$ animals (new **Suppl. Fig. 15**). Importantly, similar *C9orf72* mRNA levels are detected in $C9^{ubi}$ and *eif-2D* (-); $C9^{ubi}$ animals (**Fig. 4a**). Hence, there is no effect on poly-GR and *C9orf72* mRNA levels *eif-2D* (-); $C9^{ubi}$ animals, although a strong decrease in poly-GA and a modest reduction in poly-GP are observed, likely explaining the amelioration of lifespan (**Fig. 4e**) and locomotion defects (new **Suppl. Fig. 8-9**) of these animals when compared to $C9^{ubi}$.

The dramatic reduction of poly-GA in *eif-2D* mutants suggests *C. elegans* eIF-2D plays a crucial role in poly-GA translation (**Fig. 4-5**). To extend these findings to mammalian cells, we have now transfected mouse NSC-34 motor neuron-like cells with a monocistronic construct carrying 75 G4C2 repeats and flanking *C9orf72* intronic sequences (new **Fig. 6a**), similar to the one used to generate the $C9^{ubi}$ animals (**Fig. 1a**). shRNA-mediated knockdown of eIF2D in NSC-34 cells resulted in a strong reduction of poly-GA (new **Fig. 6b-d**), similar to the effects observed with *C. elegans eif-2D* mutants.

Next, we employed an *in vitro* platform to differentiate iPSC-derived motor neurons from a healthy individual and two ALS patients with G4C2 repeat expansions. We were unable to detect poly-GA expression above background levels in the ALS patient-derived motor neurons (despite using a highly sensitive MSD assay), preventing us from testing the role of eIF2D in poly-GA translation (**Supplementary Fig. 25**). Nevertheless, we did detect poly-GR and poly-GP in these same samples, but – similar to our *C. elegans* findings – siRNA-mediated knockdown of eIF2D had no effect on poly-GR (**Supplementary Fig. 25**). Lastly, we also observed no difference in poly-GP expression after eIF2D knockdown (**Supplementary Fig. 25**), which together with the small effect on poly-GP observed in *C. elegans eif-2D* mutants suggests multiple translation initiation factors control poly-GP expression.

Our data with monocistronic G4C2 constructs in *C. elegans* (**Fig. 4-5**) and mouse NSC-34 cells (**Fig. 6a-d**) are complemented with *in vitro* data in human HEK293 cells

transfected with a bicistronic construct also carrying 75 G4C2 repeats (**Figure 6e**). Similar to our findings in *C. elegans eif-2D* mutants, eIF2D knockdown decreased poly-GA and poly-GP (**Figure 6h-i**). The HEK293 results are also mentioned in the response to query 2a of Reviewer #2.

Taken together, our *in vivo* (*C. elegans*) and *in vitro* (NSC-34, HEK293) data uncovered a conserved role for eIF2D in controlling RAN translation of at least one DPR (poly-GA).

All the aforementioned data are now included in the Results section entitled “Knockdown of *eIF2D* suppresses poly-GA production in human cells” and discussed in the Discussion section entitled “Multiple factors are involved in DPR translation from G4C2 nucleotide repeats”.

Minor points:

1. The whole manuscript did not examine the RNA foci formed by the repeats.

We have now examined RNA foci in $\Delta C9^{ubi}$, $C9^{ubi}$, UAG^{ubi} , $eif-2D(-)$; UAG^{ubi} , $C9^{neuro}$, and $eif-2D(-)$; $C9^{neuro}$ animals. Please, see our responses to Reviewer 1, queries **2c**, **2d**, **5a**.

2. On page 11 line 243, it should be Fig. 4F-G.

Fixed. Thank you.

Reviewer #3 (Remarks to the Author):

Summary of manuscript: The manuscript details the creation and initial analysis of a new G4C2-repeat model of ALS and FTD in *C. elegans*, with an additional focus on the impact of eif2D loss of DPR expression and presumably consequent suppression of motor neurons defects in this model. Additional studies are undertaken in cell culture to extend results to mammalian systems. The development of a new model will be warmly welcomed by the field. There are some significant problems with the manuscript, which are described below. These include multiple missing controls, failure to include specific information/results, and incomplete description of methods, results and other materials.

We thank the reviewer for pointing out that our *C. elegans* model will be an important contribution to the ALS/FTD field. As detailed below, we have conducted new experiments and included additional controls and the requested information.

Major Comments

1. The manuscript expresses the GA dipeptide in-frame with nanoluciferase, creating a fusion protein. But, essential controls are lacking; specifically, expression of GA without luc as a C-terminal fusion and expression of luc without GA. Scoring these controls in GA Westerns and in assays for locomotion, MN loss and synaptic protein loss is essential.

We have now generated two additional control strains. **First**, we generated a construct that contains the upstream 113 nt and downstream 99 nt sequences, as well as nLuc, but lacks the G4C2 repeat sequences (new **Supplementary Fig. 2**). We microinjected the construct into *C. elegans*, and these new transgenic animals are referred to as Δ G4C2 repeats^{ubi}.

Similar to the Δ C9^{ubi} control strain (which lacks the G4C2 repeats and the surrounding intronic sequences but express nLuc - Figure 1a), the Δ G4C2 repeats^{ubi} animals do not produce poly-GA and display normal movement (new **Supplementary Fig. 2**). **Second**, we generated a construct that contains the upstream 113 nt and downstream 99 nt sequences, as well as the G4C2 repeat sequences, but lacks nLuc (new **Supplementary Fig. 2, 4b**).

Transgenic animals carrying this construct are named “C9 Δ nLuc^{ubi}”. These animals express *C9orf72* mRNA and can produce poly-GA (new **Supplementary Fig. 2, 4b**). We have now included these new controls in Results section entitled “Generation of a new *C. elegans* model for *C9orf72* mediated ALS/FTD”).

2a. Clarity regarding where the transgenes are expressed is critical. The authors use 1094bp of the *snb-1* promoter, which was reported to drive ubiquitous expression by Stefanakis, et al. This is not the usual result for *snb-1* promoter constructs, which tend to drive expression predominantly in neurons. And, predominant neuronal expression is seen in this manuscript when GA::nLuc is detected in Figure 1F.

We performed several new experiments to address this important comment. **First**, RNA FISH showed that G4C2 RNAs are expressed in neuronal and some non-neuronal cells in C9^{ubi} animals, which drive expression of 75 G4C2 repeats using the *snb-1* promoter (1094bp) (new **Fig. 1c**, new **Supplementary Fig. 3**). **Second**, antibody staining for poly-GA in C9^{ubi} animals showed poly-GA aggregation in head neurons, as well as ventral nerve cord motor neurons

and some non-neuronal cells (**Fig. 1i**). **Third**, we generated transgenic reporter animals that express GFP under the control of the same *snb-1* promoter (1094bp) and 3'UTR (*unc-54*) sequences used in *C9^{ubi}* animals. These animals (*Psnb-1::GFP*) confirmed that the *snb-1* promoter is indeed active in most, if not all, *C. elegans* cells (new **Supplementary Fig. 1**), as previously reported by Stefanakis et al. (*Neuron*, 2015, PMID: 26291158). **Fourth**, we generated new *C. elegans* animals, in which a GFP reporter is in the poly-GA open reading frame. These *C9-GFP^{ubi}* animals carry the exact same construct as *C9^{ubi}* animals in Figure 1, but the nLuc sequence is replaced with GFP (new **Supplementary Fig. 5a-b**). We observed aggregated GFP puncta (a proxy for poly-GA accumulation) in head and tail neurons, as well as ventral nerve cord motor neurons and other tissues (e.g., muscle, intestine) (new **Supplementary Fig. 5a-b**). Poly-GA-GFP accumulation was progressive with age in these cells of *C9-GFP^{ubi}* animals. We now present all these new data in the first Results section entitled "Generation of a new *C. elegans* model...".

2b. While the reviewer does not doubt Stefanakis, transformation markers and relative concentrations of plasmids/DNA molecules can dramatically change expression patterns. At a minimum, the authors need to provide an image that shows the ubiquitous expression of nLuc driven by this promoter in their hands -- using precisely the same transformation markers and relative injection concentrations used for manuscript for other constructs. This is critical for readers to gauge how "ubiquitous" expression is - and the relative level of expression in the nervous system.

Please see our response to **query 2a**. We further note that to generate the *C9-GFP^{ubi}* animals, we used injection concentrations identical to the ones used for *C9^{ubi}* animals, as detailed below and in the revised Materials and Methods.

Plasmid DNA concentration in injection mix for *C9^{ubi}* animals:

C9^{ubi} plasmid 100 ng/μl, *myo-2:gfp* plasmid 3 ng/μl, pBS plasmid 20 ng/μl

Plasmid DNA concentration in injection mix for *C9-GFP^{ubi}* animals:

C9-GFP^{ubi} plasmid 100 ng/μl, *myo-2:mCherry* plasmid 3 ng/μl, pBS plasmid 20 ng/μl

Lastly, reliable antibodies that recognize nLuc using immunohistochemistry are unfortunately not available.

2c. Additionally, Figure 1F shows expression of GA in unidentified head neurons-- and does not show if GA is expressed in motor neurons, which are scored for survival in other figure panels. The manuscript must show GA expression in motor neurons or admit that they cannot detect it.

Please see our response to **query 2a**. We now provide images showing poly-GA aggregation in head neurons, as well as ventral nerve cord motor neurons of *C9^{ubi}* animals (**Fig. 1i**).

Similar poly-GA aggregation in motor neurons was also observed in the *C9-GFP^{ubi}* animals that carry the exact same construct as *C9^{ubi}* animals in Figure 1a, but the nLuc sequence is replaced with GFP (new **Suppl. Fig. 5b**).

Importantly, we also observed an age-dependent poly-GA accumulation (new **Suppl. Fig. 5a-b**). Younger animals show less poly-GA accumulation in neurons than older animals, consistent with observations in a recent *C9orf72* mouse model describing age-dependent DPR accumulation (Choi et al. *Nat Neuroscience*, PMID: 31086314). We now note in the text this age-dependent poly-GA accumulation (page 7, 2nd paragraph).

2d. This also brings into question the manuscripts assertions regarding RAN translation in neurons; this speculation should be diminished or eliminated, given the high level of neuronal expression in transgenic animals with the *snb-1* constructs.

We completely agree and have eliminated such assertions in the revised manuscript.

3. New supplemental figures must be added that include an image of entire Western/immunoblot and show markers/size of bands - this is relevant for the GA, nanoLuc, detection.

We now provide new supplemental images that show the entire Western blots in the Source Data file. This file also contains the raw data that provided the reported means/averages in the bar charts.

4a. Details must be provided for the individual transgenic lines reported. Extrachromosomal lines were presumably scored separately, and they should be reported separately in the supp materials, although it is certainly appropriate to merge these in the main text figures, if the lines have essentially the same impact on phenotype for each assay. This table should also provide sufficient detail to provide the reviewer/reader with the number of animals scored for each genotype/age.

We regret a lack in clarity. We now provide **Supplementary Table 5**, which includes important details on all transgenic lines generated for this study. In brief, at least 2-3 transgenic lines (extrachromosomal arrays) were generated for each construct and were analyzed in the context of molecular and lifespan assays, as shown in Figures 1 and 3. For example, **Figure 1d** shows poly-GA detection on three different extrachromosomal lines for each genotype ($\Delta C9^{ubi}$, $C9^{ubi}$, UAG^{ubi}). In our lifespan assays, multiple lines were tested and the experiments were repeated 3 times. Results for each transgenic line and each replicate are now shown in **new Supplementary Figures 7, 12, 16, 17, 20, 21**. In the main figures (**Figures 2a, 3f, 4e, 4j**), the lifespan assay results are merged for the different transgenic lines because they had a similar effect on the phenotype. We noted this in the respective figure legends.

We also generated integrated versions of the $\Delta C9^{ubi}$, $C9^{ubi}$, and UAG^{ubi} constructs (**Supplementary Table 5**). We obtained one integrated line for each and outcrossed it 5x to a wild-type (N2) background. Details are provided in Materials and Methods. We used these animals carrying the integrated transgenes for RNA FISH experiments (**Fig. 1c**), ELISA (**Fig. 1g-h**) quantification of the number of cholinergic motor neurons (**Fig. 2e**) and a swimming (thrashing) assay on (**Fig. 2b**).

4b. For example, Figure 2 suggests that motor neurons are lost, but the number of animals scored is unclear beyond “>10 animals scored”. Eleven animals would not be enough for any genotype/age and the individual data points should be included here to permit assessment of variation.

We now provide this information for all experiments in the Source Data file. For **Figure 2e**, we have conducted additional analysis and included the individual data points. For motor neuron cell body counts, we scored 25 animals per genotype per stage (L4, day 2, day 5).

5a. Have *eif-2A(gk358198)* and *eif-2D(gk904876)* been reported in the literature previously? Our study is the first one to use these mutant alleles. As suggested by the reviewer, we have conducted additional experiments to verify that *eif-2D* function is required for DPR (poly-GA) translation. Please see response to **query 5c**.

5b. Were these alleles backcrossed prior to these studies (to remove mutations in other genes) and how many times?

Animals carrying the *eif-2A (gk358198)* and *eif-2D (gk904876)* alleles were backcrossed three times to the wild-type (N2) strain. This info is now included in Materials and Methods.

5c. The impact of background mutations cannot be ruled out with 1) rescue studies, 2) additional alleles with non-complementation studies, or 3) well-controlled RNAi studies. Using one of these approaches is critical and absolutely required to conclude that gene function is required. Given the manuscript discussion focus on eIF2D, it would be ideal to focus on validating this gene.

We completely agree and thank the reviewer for this excellent point. In the original submission, we showed that poly-GA translation is dramatically decreased in *eif-2D (gk904876)*; *C9^{ubi}* animals compared to *C9^{ubi}* controls (**Fig. 4c-d**). The *gk904876* allele carries a point (nonsense) mutation in the second exon, generating a premature STOP codon early in the protein - at the 25th amino acid (R25Opal) out of the 549aa that make up the wild-type eIF2D protein. Prompted by the reviewer’s comment, we used two additional mutant alleles for *eif-2D*. **First**, the *gk561128* allele carries a point (nonsense) mutation in the fifth exon, generating a premature STOP codon at the 512th amino acid position (Q512Ochre). This mutation disrupts the SUI1 domain, which plays a crucial role in initiator codon recognition (PMID:18593708; PMID:12435632; PMID:1729602; PMID:16362046). Similar to our findings with the *gk904876* allele, poly-GA translation is dramatically decreased in *eif-2D (gk561128)*; *C9^{ubi}* animals compared to *C9^{ubi}* controls (new **Figure 5**). To help the reader, we now show the precise molecular lesion for each *eif-2D* allele in **Fig. 5a**.

Second, we employed CRISPR/Cas9 genome editing in *C. elegans* and selectively replaced the SUI1 domain with a 3xFLAG epitope at the endogenous *eif-2D* locus (new **Fig. 5a-d**). Again, we observed a dramatic down-regulation of poly-GA expression in homozygous animals that lack the SUI1 domain (termed *eif-2D (syb3432[eif-2d Δ^{SUI1}])*), albeit they did express the expected truncated eIF2D protein. These two additional mutant alleles confirm our initial observations with the *eif-2D (gk904876)* allele and provide mechanistic insights by implicating the SUI1 domain of eIF2D in DPR translation.

Lastly, we performed a rescue experiment by supplementing eIF2D (using the *snb-1* promoter) into *eif-2D* (*gk904876*); *C9^{ubi}* animals (new **Supplementary Fig. 14**). Animals carrying the rescue construct (*FLAG-eIF2D^{ubi}*) displayed increased levels of poly-GA translation compared to *eif-2D* (*gk904876*); *C9^{ubi}* animals. We observed partial rescue though, likely due to mosaicism of the extrachromosomal arrays used for this experiment (transgenes *C9^{ubi}* and *FLAG-eIF2D^{ubi}* were both used as extrachromosomal arrays. These new findings are now included in Results (page 11-14).

6a. Lifespan assays should be replicated without FUDR; the *C. elegans* field has numerous examples of FUDR altering lifespan for mutant genotypes. And, citing Kramer et al for lifespan methods is not sufficient; more detail is required.

We have now carried out the lifespan assays without FUDR, and the new results (new **Supplementary Figures 17, 21**) are in agreement with the ones reported in the original submission (**Fig. 2a, 3f, 4e, 4j**). Importantly, the assays were repeated on two sets of animals that either drove expression of the G4C2 repeats broadly (*C9^{ubi}*), or exclusively in the nervous system (*C9^{neuro}*).

Genotypes for **Set 1**: $\Delta C9^{ubi}$, *C9^{ubi}*, *UAG^{ubi}*, *eif-2D* (-); *C9^{ubi}*, *eif-2D* (-); *UAG^{ubi}*. The assays for Set 1 were repeated twice (new **Supplementary Figure 17**).

Genotypes for **Set 2**: $\Delta C9^{neuro}$, *C9^{neuro}*, *UAG^{neuro}*, *eif-2D* (-); *C9^{neuro}*, *eif-2D* (-); *UAG^{neuro}*, *eif-2D* (-). The assays for Set 2 were repeated three times (new **Supplementary Figure 21**).

The new results are in agreement with the ones reported in the original submission, showing that lack of *eif-2D* restores the lifespan defects of *C9^{ubi}* and *C9^{neuro}* animals (**Fig. 4e, 4j**, new **Suppl. Fig. 16, 20**). We now provide more details on these assays in Materials and Methods, and the results are presented in the Results section entitled “Genetic removal of the eukaryotic translation initiation ...”.

6b. Were animals this manuscript actually reared and aged on RNAi-competent bacteria strains as in Kramer, or was the laboratory standard OP50 used here?

Standard OP50 bacteria were used. We now describe with more details the lifespan assays in Materials and Methods.

6c. How many biological replicates were used and how many animals were used in each of these trials? A data table is required for replicates of lifespan determinations, including median lifespan, number of animals included, number of animals censored, statistical determinations, etc. Individual experiments should be broken out allowing readers to determine which controls are matched with which experimental strains. This is standard in the field. Many other determinations should also be in a supplemental data table or the figure legend, including p-values (example - expression level analyses.) It is no longer sufficient to just label these with asterisks or N.S.

Thank you for this comment. To address it, we generated 6 new figures (new **Supplementary Fig. 7, 12, 16, 17, 20, 21**). In each of these, we provide the total number of animals per transgenic line used in the lifespan assays shown in main Figures **2a, 3f, 4e, and 4j**. We also include median lifespan, number of animals censored, and statistical determinations. Within each of these Supplementary Figures, the survival data are now shown per replicate, as well as combined. Each lifespan assay included >20 worms for each genotype and was repeated two or three times (new **Supplementary Fig. 7, 12, 16, 17, 20, 21**). We note that worms from all genotypes were assessed simultaneously in each replicate. As requested, we now use exact p values, not asterisks or N.S, in every graph of the manuscript, including RT PCR and WB quantifications. The raw data for every graph are included in the Source Data File.

7a. It is important for the manuscript to state if experimenters were blinded as to genotype for MN scoring and puncta analysis.

The motor neuron cell body and puncta analyses, as well as the DPR detection assays were conducted in a blinded fashion. We now mention this in Materials and Methods.

7b. Manuscript must also report results in much more detail, including how the GFP synaptic protein expression was quantified. Information required includes, but is not limited to, the number of neurons scored per animal and the number of animals scored for all determinations. Another age point for MN loss would be helpful, but not essential.

We agree this is very important and have now significantly expanded the Materials and Methods section to include more details on all assays and analyses, including quantification of MN loss. Please, see response to **query 4b**. All figure legends mention the exact number of animals scored.

Regarding the GFP synaptic protein quantifications, we were unable to increase the number of animals because of the following issue: The $C9^{ubi}$ animals display locomotion and lifespan defects, and also have a small brood size. When crossed with the animals carrying the *otIs437 [Punc-3::eGFP::rab-3cDNA]* transgene used for synaptic analysis, all these defects were exacerbated to the point we were unable to maintain the strain in subsequent generations and increase the number of animals for the synaptic analysis. Because this is likely to occur due to background mutations carried over from the *otIs437* genetic background, we decided to remove the synaptic analysis data altogether from the manuscript. This change however does not affect our conclusion on MN cell loss. The $C9^{ubi}; vsIs48 [unc-17::GFP]$ animals superficially do not behave any different from the $C9^{ubi}$ animals, hence we were able to increase the number of animals to 25 and observed a statistically significant effect on MN cell loss (**Fig. 2e**).

8. For the field in general, it is important to know if other types of neurons are affected, including glutamatergic and dopaminergic. Transgenes are available to test this. This would dramatically raise the impact of this model.

To address this crucial point, we crossed the $\Delta C9^{ubi}$, $C9^{ubi}$, and UAG^{ubi} animals with the dopaminergic reporter strain *otIs181 [dat-1::mCherry + ttx-3::mCherry]*. We quantified the

total number of dopaminergic head neurons at larval (L4) and adult (day 2, day 5) stages. This longitudinal analysis did not reveal any significant differences in the number of dopaminergic head neurons when $C9^{ubi}$ animals were compared to controls ($\Delta C9^{ubi}$, UAG^{ubi}) (new **Suppl. Fig. 11**). Despite multiple attempts, we were not able to cross a red glutamatergic marker strain (*otIs292[eat-4::mCherry + rol-6(su1006)]*) with $C9^{ubi}$ animals, likely due to genetic linkage of the two integrated transgenes. We now mention the results with the dopaminergic marker in Results section entitled “ $C9^{ubi}$ animals display progressive locomotion defects and motor neuron degeneration”.

9. The behavioral analysis presented in Figure 5 is extremely challenging for readers who are not familiar with this specific analysis system, which is not widely used in the field. It's unclear what the parameters described are measuring. This is especially important for the [*eif-2D* (-); $C9^{ubi}$] genotype; it is hard to understand if the defect described in these graphs actually causes impaired locomotion or just subtle changes. All the behavioral metrics presented in the panels A, B and C should be described in a supplementary figure and supported by videos showing the locomotion defect versus control animals. Note that the videos can be at an external site, as long as they have a permanent doi/location.

We agree and have now moved these data into new **Supplementary Figures 8, 9, 10, and 19**. These data showed a modest effect in locomotion of $C9^{ubi}$ animals due to: (a) mosaicism - the automated tracking analysis was performed on animals carrying the $C9^{ubi}$ transgene as an extrachromosomal array (the integrated $C9^{ubi}$ transgene is unfortunately genetically linked to the *eif-2D* locus), and (b) for technical reasons the automated tracking was performed on young adult animals at day 2, a time point preceding MN cell loss (**Fig. 2e**). Motivated by the reviewer's comment, we conducted - in a longitudinal fashion (L4, day 2, day 5) - a thrashing experiment, which measures the rate of body flexion while the animals are swimming. This enabled us to detect an age-dependent defect in the rate of body bends of $C9^{ubi}$ animals (**Fig. 2b**). Importantly, we used three control strains for this thrashing experiment ($\Delta C9^{ubi}$, UAG^{ubi} in **Fig. 2b** and $\Delta G4C2$ repeats^{*ubi*} animals in new **Supplementary Fig. 2e**). Lastly, we now provide videos showing the locomotion defects of $C9^{ubi}$ animals compared to control $\Delta C9^{ubi}$ and UAG^{ubi} strains (new Suppl. Videos 1-3). These new results are presented in Results section entitled “ $C9^{ubi}$ animals display progressive locomotion defects...”.

Minor Comments.

A. The reviewer cannot find *unc-11c* in the Stefanakis, et al. reference cited. Please double check that this terminology is correct.

We have corrected this. The correct terminology is *unc-11 prom8*.

B. References are lacking in multiple places, including references for transgenes and alleles generated in other laboratories.

In the revised manuscript, we included additional references in Results, Discussion and Materials and Methods, as well as to alleles and transgenes.

C. Instead of just stating that *eif-2A* (*gk358198*) and *eif-2D* (*gk904876*) are null alleles, the manuscript should at least report what the amino acid change is for each genes (e.g. W146*) to allow readers to understand why the manuscript suggests this is a complete loss of function allele.

Thank you for this comment. Regarding the three different alleles of *eif-2D*, please see our response to **Reviewer 3, query 5c**. The *eif-2A* (*gk358198*) allele carries a point (nonsense) mutation in the fifth exon, generating a premature STOP at the 136th amino acid position (W136Amber). The wild-type eIF2A protein contains 177 amino acids. In the revised manuscript, we describe in detail the *eif-2A* and *eif-2D* alleles in text (page 11) and **Fig. 5a**.

D. The reviewer finds it frustrating that no assessments were performed in human cells to address a potential impact of eIF2A on DPR generation (especially GR and GP). This would strengthen the manuscript as the cell culture system has higher expression of GR and GP, and also would provide a more sensitive context to study the impact of genetic background manipulation on DPRs. However, this lack does not impact the results presented.

In the original submission, we discussed and cited our previous study on the effect of eIF2A on DPR translation in cultured cells as well as chick embryo spinal cords (Sonobe et al., *Neurobiology of Disease*, PMID: 29792928). In page 23 (Discussion), we note:

*“Our previous study demonstrated that knock-down of eIF2A in HEK293 cells and neural cells in the chick embryo led to a modest (30%) decrease in poly-GA expression from G4C2 expanded repeats²⁰. In the present study, however, eif-2A/eIF2A deficiency had no effect on poly-GA expression in C. elegans, while eIF2D knock-down in NSC-34 and HEK293 cells and eif-2D knock-out in C. elegans had a prominent effect on poly-GA expression, suggesting eIF2D plays a more important role in RAN translation than eIF2A. It is possible, however, that eIF2A also plays a significant role in RAN translation, but only upon cellular stress since it can translate a subset of mRNAs when eIF2 α is phosphorylated^{17, 18, 50}. We note that eIF2D does not control the levels of phosphorylated eIF2 α (**Supplementary fig. 27-28**).”*

E. In line 168 the manuscript describes a behavioral defect. But, these results are not presented until Figure 5, which is referenced in line 256 in the G4C2 suppressor analysis. This should be fixed.

We have now fixed this issue.

F. At line 403 the manuscript states that “the present study uncovered a requirement for eif-2D/eIF2D in RAN translation”. However other studies have suggested a role of eif2D in RAN translation (ref 19 of this paper) and these should be given more prominence in the introduction and discussion.

We reread reference 19 (Sendoel, A. et al. Translation from unconventional 5' start sites drives tumour initiation. *Nature* **541**, 494-499, 2017). The study focuses on eIF2A, not eIF2D. As suggested by the reviewer though, we now include additional eIF2D references in the revised manuscript (pages 21-23).

G. At line 109, it would be helpful if the manuscript was more precise regarding impact on specific dipeptides, with detail similar to that used at line 401.

We have now fixed this. Thanks.

REVIEWER COMMENTS

Reviewer #1 (Remarks to the Author):

This is a much-improved manuscript with substantial new data, which greatly strengthen the manuscript and which now provide mechanistic insight into the effects of eIF-2D on RAN translation. I do still have concerns with the way some of the experiments are described but these can be addressed by carefully editing the text.

Line 149. The authors should also list C9 Δ nLuc as being positive for mRNA. This is shown in Supplemental Fig 2B but was not mentioned in the text.

Line 179-181: "Interestingly, translation (measured by the nLuc reporter) was initiated from this same CUG in Δ G4C2 repeats ubi animals, even though there were no G4C2 repeats (Supplementary Fig. 6)." This statement is not supported by the data since no experiment was done to show that the CUG is the initiation site for nLuc – all of the flanking sequence was deleted so it is not clear that translation initiation really in fact starts with the same CUG or at another site within the deleted sequence. This can be addressed by revising the text.

The authors should be careful to say that they are measuring steady-state levels of DPRs not DPR production. This is important because no assays have been done to assess protein production vs protein degradation. This is again a change that can be addressed by editing the text but it is a critical edit that needs to be made throughout the manuscript. I highlight a few examples below but there are many more instances as well and this needs to be changed throughout the manuscript.

Lines 235-237 "Our findings suggest that DPR (poly-GA) production from the G4C2 repeats exclusively in neurons is sufficient to trigger disease associated phenotypes, establishing C9 neuro animals as a model for C9orf72-mediated ALS/FTD."

This is not necessarily true that polyGA production is responsible for this change and should be rephrased to say that decreased levels of GA proteins are sufficient to improve the phenotypes.

260-262 "Of note, we found a prominent decrease in poly-GA expression in eif-2D (gk904876); C9 ubi compared to C9 ubi animals, but no effect in eif-2A (gk358198); C9 ubi animals (Fig. 4c-d), implicating eif-2D in DPR translation."

It is important when discussing the results of this figure that that authors state that they see a decrease in steady state levels and to specify that this is for the GA protein and not the more general and incorrect statement of "DPR production".

270-271 "Since eif-2D is necessary for translation of at least two of the DPRs, loss of eif-2D gene activity may improve survival in C9 ubi animals."

Needs to be reworded as the authors have not performed pulse-chase or other experiments to distinguish if eif-2D effects which are clearly seen on steady state protein levels truly affect "translation" or a post-translational effects.

Supplementary Figure 29 Lines 470-472 It is noteworthy that genetic removal of eif-2D in the worm does not appear to have an effect on: (a) canonical AUG-initiated translation (Supplementary Fig. 22), (b) animal locomotion, and (c) lifespan.

The authors should soften this language as they have not done sufficient work to make the global suggestion that eif-2D does not affect canonical AUG-initiated translation.

Discussion:

512-517 In the present study, however, eif-2A/eIF2A deficiency had no effect on poly-GA expression in *C. elegans*, while eIF2D knock-down in NSC-34 and HEK293 cells and eif-2D knock-out in *C. elegans* had a prominent effect on poly-GA expression. These results suggest that eIF2D plays a more

important role in RAN translation than eIF2A. It is possible, however, that eIF2A also plays a significant role in RAN translation, but only upon cellular stress since it can translate a subset of mRNAs when eIF2 α is phosphorylated^{17, 18, 59}.

A very recent publication in Human Molecular Genetics (Tusi et al., 2021) shows eIF2A plays an important role RAN translation of DM2 CAGG and CCUG repeats. This paper should be cited and discussed.

Supplementary 2e and a few other supplementary figure panels are cited out of order in the text. All figures and figure panels should be cited in the order that they are referred to in the text for ease of reading.

Reviewer #2 (Remarks to the Author):

The authors included many additional data and significantly improved the manuscript. There are still weaknesses and some of the concerns were not fully addressed.

1. The data about eIF2D on RAN translation of different reading frames is very confusing, and not consistent in different systems (original comment 2). In 293 cells, knockdown of eIF2D clearly reduced all GA, GP and GR. The authors added data using NSC-34 cells and showed knockdown of eIF2D reduced GA level. However, they did not show data on GP and GR, which should be included. Therefore hard to conclude there is specific effect on GA in this cell line. In *C.elegans*, the authors claimed eIF2D only affected GA significantly, a little on GP and no effect on GR. However, the GR level in C9ubi animal was not reliably measured (no significant difference from the control). Therefore it is not appropriate to conclude there are no changes of GR in eif-2D (-), as this is probably the problem of assay sensitivity. For GP, there is definitely robust downregulation in eif-2D (-). As negative control was not included in the same graph, and different y-axis units were used in different figures (norm/Nluc RNA in Fig 1g,h; arbitrary unit in sup fig 15), it is impossible to conclude how much GP was reduced without the baseline information. It is better to use consistent format/unit and include negative controls for all graphs. The authors also tried to test in patient iPS-neurons, but unfortunately were not able to measure GA, therefore had no evidence to show the knockdown of eIF2D is sufficient in this model. In general, negative data due to technical difficulties should not be used as evidence of no effects. In the four systems, there are more data supporting that eIF2D affects all the reading frames instead of specific effects on GA. The authors should be more cautious with the words and conclusions. Furthermore, there is no evidence to show the effects requires CUG codon.

2. It is nice the authors now included patient-derived iPS-neurons. But unfortunately they did not use this system to address the reviewer's concerns (original comments 5 and 6). The authors mentioned they were unable to identify endogenous genes using CUG for translation initiation in *C.elegans*. They could test the CUG transcripts in the human iPS-neurons as they already have the model. In addition, a puromycin incorporation assay could be used to test whether there is effect on global translation. Furthermore, the reviewer suggested to test the toxicity phenotype rescue effect of knocking known eIF2D in patient iPS-neurons. Although GA level was not detectable, they could still show eIF2D knockdown and measure the effects on neuron survival.

Reviewer #3 (Remarks to the Author):

It's lovely that the authors have so vigorously addressed concerns raised previously. All of my substantive points have been addressed. A few minor points remain:

Inaccurate/confusing sentence – “knockdown” does not have a role in poly-GA production
“Lastly, knockdown of eIF2A, an unconventional translation initiation factor used by cells under stress conditions^{17, 18, 19}, had a partial role in poly-GA production in HEK293 cells and neural cells of the chick embryo...”

In Figure legend 1 and elsewhere, Act-1 should not be capitalized,
"The G4C2 repeat mRNA (C9orf72 intronic mRNA) and Act-1 mRNA were assessed by RT-PCR..."

In Figure Legends 1 and 2, KasEx154 and KasEx155 should not be capitalized. Similarly, KasEx157 should not be capitalized in Figure Legend 4.

Multiple comparisons corrections should be applied to Figure 6C and D. Reporting just SEM is not sufficient when so many comparisons are made. The reviewer suspects that this will not change the conclusions to be drawn.

Draft Only

Reviewer #1 (Remarks to the Author):

This is a much-improved manuscript with substantial new data, which greatly strengthen the manuscript and which now provide mechanistic insight into the effects of eIF-2D on RAN translation. I do still have concerns with the way some of the experiments are described but these can be addressed by carefully editing the text.

Line 149. The authors should also list C9 Δ nLuc as being positive for mRNA. This is shown in Supplemental Fig 2B but was not mentioned in the text.

We have now changed the text to: First, we performed quantitative PCR with reverse transcription (RT-qPCR) and found that mRNA corresponding to intronic sequences that flank the G4C2 repeats is only detected in C9^{ubi}, UAG^{ubi}, Δ G4C2 repeats^{ubi}, and C9 Δ nLuc^{ubi} animals, whereas nLuc mRNA is only detected in C9^{ubi}, UAG^{ubi}, and Δ G4C2 repeats^{ubi} animals (**Fig. 1b, Supplementary Fig. 2b-c**).

Line 179-181: “Interestingly, translation (measured by the nLuc reporter) was initiated from this same CUG in Δ G4C2 repeats ubi animals, even though there were no G4C2 repeats (Supplementary Fig. 6).” This statement is not supported by the data since no experiment was done to show that the CUG is the initiation site for nLuc – all of the flanking sequence was deleted so it is not clear that translation initiation really in fact starts with the same CUG or at another site within the deleted sequence. This can be addressed by revising the text.

We completely agree. The new sentence is: “Interestingly, the nLuc reporter gene is translated in Δ G4C2 repeats^{ubi} animals, which lack G4C2 repeats but carry the intronic C9orf72 sequences including the upstream CUG (**Supplementary Fig. 6**). However, it remains unclear whether this CUG can initiate translation in the absence of G4C2 repeats.”

The authors should be careful to say that they are measuring steady-state levels of DPRs not DPR production. This is important because no assays have been done to assess protein production vs protein degradation. This is again a change that can be addressed by editing the text but it is a critical edit that needs to be made throughout the manuscript. I highlight a few examples below but there are many more instances as well and this needs to be changed throughout the manuscript.

We thank the reviewer for this important point. Throughout the manuscript, we have now changed “DPR production” to “DPR steady-state levels”.

Lines 235-237 “Our findings suggest that DPR (poly-GA) production from the G4C2 repeats exclusively in neurons is sufficient to trigger disease associated phenotypes, establishing C9 neuro animals as a model for C9orf72-mediated ALS/FTD.” This is not necessarily true that polyGA production is responsible for this change and should be rephrased to say that decreased levels of GA proteins are sufficient to improve the phenotypes.

We agree and have now corrected this conclusion: Our findings establish C9^{neuro} animals as a model for C9orf72-mediated ALS/FTD, and further suggest that decreased levels of poly-GA proteins in neurons are sufficient to improve ALS/FTD-associated phenotypes.

260-262 “Of note, we found a prominent decrease in poly-GA expression in eif-2D (gk904876); C9 ubi

compared to C9 ubi animals, but no effect in eif-2A (gk358198); C9 ubi animals (Fig. 4c-d), implicating eif-2D in DPR translation.”

It is important when discussing the results of this figure that that authors state that they see a decrease in steady state levels and to specify that this is for the GA protein and not the more general and incorrect statement of “DPR production”.

We have now corrected the text: “Of note, we found a prominent decrease in poly-GA steady-state levels in *eif-2D* (gk904876); C9^{ubi} compared to C9^{ubi} animals, but no effect in *eif-2A* (gk358198); C9^{ubi} animals (Fig. 4c-d), implicating *eif-2D* in poly-GA expression”.

270-271 “Since eif-2D is necessary for translation of at least two of the DPRs, loss of eif-2D gene activity may improve survival in C9 ubi animals.”

Needs to be reworded as the authors have not performed pulse-chase or other experiments to distinguish if eif-2D effects which are clearly seen on steady state protein levels truly affect “translation” or a post-translational effects.

We corrected the text: “Since *eif-2D* is necessary for steady-state levels of at least two of the DPRs, loss of eif-2D gene activity may improve survival in C9^{ubi} animals”.

Supplementary Figure 22 - Lines 470-472: It is noteworthy that genetic removal of eif-2D in the worm does not appear to have an effect on: (a) canonical AUG-initiated translation (Supplementary Fig. 22), (b) animal locomotion, and (c) lifespan.

The authors should soften this language as they have not done sufficient work to make the global suggestion that eif-2D does not affect canonical AUG-initiated translation.

Thank you for this comment. The text now reads: It is noteworthy that genetic removal of *eif-2D* in *C. elegans* does not appear to have an effect on: (a) canonical AUG-initiated translation of three tested mRNAs (Supplementary Fig. 22), (b) animal locomotion, and (c) lifespan.

Discussion:

512-517 In the present study, however, eif-2A/eIF2A deficiency had no effect on poly-GA expression in *C. elegans*, while eIF2D knock-down in NSC-34 and HEK293 cells and eif-2D knock-out in *C. elegans* had a prominent effect on poly-GA expression. These results suggest that eIF2D plays a more important role in RAN translation than eIF2A. It is possible, however, that eIF2A also plays a significant role in RAN translation, but only upon cellular stress since it can translate a subset of mRNAs when eIF2 α is phosphorylated^{17, 18, 59}.

A very recent publication in Human Molecular Genetics (Tusi et al., 2021) shows eIF2A plays an important role RAN translation of DM2 CAGG and CCUG repeats. This paper should be cited and discussed.

The revised Discussion now mentions this recent paper: “Our previous study demonstrated that knockdown of eIF2A in HEK293 cells and neural cells in the chick embryo led to a modest (30%) decrease in poly-GA expression from G4C2 expanded repeats²⁰. A recent study also implicated eIF2A in RAN translation from CCUG and CAGG repeats in myotonic dystrophy type 2 (DM2) (Tusi et al., 2021). In the present study, however, *eif-2A/eIF2A* deficiency had no effect on poly-GA expression in *C. elegans*, while eIF2D knockdown in NSC-34 and HEK293 cells and *eif-2D* knock-out in *C. elegans* had a prominent effect on poly-GA expression.”

Supplementary 2e and a few other supplementary figure panels are cited out of order in the text. All figures and figure panels should be cited in the order that they are referred to in the text for ease of reading.

All figures and figure panels are now cited in the order they are referred to in the text. Thank you.

Reviewer #2 (Remarks to the Author):

The authors included many additional data and significantly improved the manuscript. There are still weaknesses and some of the concerns were not fully addressed.

1. The data about eif2D on RAN translation of different reading frames is very confusing, and not consistent in different systems (original comment 2). In 293 cells, knockdown of eIF2D clearly reduced all GA, GP and GR. The authors added data using NSC-34 cells and showed knockdown of eIF2D reduced GA level. However, they did not show data on GP and GR, which should be included. Therefore hard to conclude there is specific effect on GA in this cell line.

In order to extend our *C. elegans* findings, we used three other cellular systems (NSC-34, HEK293, human iPSC-derived motor neurons). To help the reader and avoid the confusion pointed out by the reviewer, we have now included a new table (**Table 1**) that summarizes the observed effects on DPRs upon *eif-2D*/eIF2D knockdown *in vivo* (*C. elegans*) and *in vitro* (NSC-34, HEK293, human iPSC-derived motor neurons). Following the reviewer's suggestion, we conducted new experiments in NSC-34 cells. We again found that eIF2D knockdown affected poly-GA expression (as reported in the previous version of the manuscript), but no effects were observed in poly-GP and poly-GR expression, as evidenced by sensitive immunoassays (new panels **e** and **f** in **Figure 6**). Lastly, we have edited the Results section entitled "Knockdown of eIF2D suppresses poly-GA production in mammalian cells" to include these new data and also highlight Table 1.

In *C. elegans*, the authors claimed eif2D only affected GA significantly, a little on GP and no effect on GR. However, the GR level in C9ubi animal was not reliably measured (no significant difference from the control). Therefore it is not appropriate to conclude there are no changes of GR in eif-2D (-), as this is probably the problem of assay sensitivity. For GP, there is definitely robust downregulation in eif-2D (-). As negative control was not included in the same graph, and different y-axis units were used in different figures (norm/Nluc RNA in Fig 1g,h; arbitrary unit in sup fig 15), it is impossible to conclude how much GP was reduced without the baseline information. It is better to use consistent format/unit and include negative controls for all graphs.

We have now conducted new experiments and included the $\Delta C9^{ubi}$ negative control. Specifically, we generated lysates from $\Delta C9^{ubi}$, $C9^{ubi}$, and *eif-2D*(-); $C9^{ubi}$ *C. elegans* animals and conducted immunoassays for poly-GP and poly-GR. As we did for the immunoassays presented in the previous version, these new assays were also done in a blinded fashion. Compared to baseline ($\Delta C9^{ubi}$), we observed a significant increase in poly-GP and poly-GR levels in $C9^{ubi}$ animals. Genetic removal of *eif-2D* affected poly-GP, but not poly-GR, hence reproducing the results we reported in the previous submission. This new data is now presented in Supplementary Figure 15.

Following the reviewer's valuable suggestion, all DPR immunoassays are presented in arbitrary units, with the exception of Figure 1g-h. The rationale for this exception is that Figure 1g-h compares the effect of CUG > UAG mutation on poly-GP and poly-GR. Because two different multi-copy transgenes are compared ($C9^{ubi}$ versus UAG^{ubi}), the normalization to nLuc mRNA levels is important to ensure that the observed lack of an effect on poly-GP and poly-GR upon UAG mutation is not a consequence of different copy numbers of the $C9^{ubi}$ and UAG^{ubi} transgenes.

The authors also tried to test in patient iPSC-neurons, but unfortunately were not able to measure GA, therefore had no evidence to show the knockdown of eIF2D is sufficient in this model. In general, negative data due to technical difficulties should not be used as evidence of no effects.

We completely agree and now provide further details on this in the text. It is possible that different C9ORF72 iPSC lines (from different patients) express different levels of poly-GA. One possibility is that poly-GA production depends on the number of G4C2 repeats, which is not known for the C9ORF72 iPSC lines we used. As noted in Methods, the number of G4C2 repeats in these lines is > 24. We have modified the text to include these important details: “Next, we employed an *in vitro* platform to differentiate iPSC-derived motor neurons from a healthy individual and two ALS patients with > 24 G4C2 repeat expansions. Despite using a sensitive ELISA method, we were unable to detect poly-GA expression above background levels in the iPSC-derived motor neurons from these specific ALS patients, preventing us from testing the role of eIF2D in poly-GA translation in these cells (**Supplementary Fig. 25**). Although we could not detect poly-GA, we did detect poly-GR and poly-GP in these same samples.”

In the four systems, there are more data supporting that eIF2D affects all the reading frames instead of specific effects on GA. The authors should be more cautious with the words and conclusions.

Based on **Table 1**, poly-GA is affected upon eIF2D knockdown in *C. elegans*, NSC-34, and HEK293 cells. Poly-GP is also affected in *C. elegans* and HEK293 cells, and whereas poly-GR is only affected in HEK293 cells. Because the HEK293 were transfected with a bicistronic construct (line 362), the way DPR translation is initiated in this construct is likely quite different from the monocistronic constructs used in the worm and NSC-34 cells, and from the endogenous human C9ORF72 locus (please see lines 372-375 in manuscript). As the reviewer suggested, we have carefully edited the text to precisely describe what we observed, that is, a consistent decrease of poly-GA in three systems, a decrease of poly-GP in two systems, and a decrease of poly-GR in one system. Hopefully, the addition of Table 1 will help the readers.

Furthermore, there is no evidence to show the effects requires CUG codon.

We completely agree. We have modified the text accordingly. In the Discussion section entitled “A novel role for *eif-2D/eIF2D* in regulating poly-GA expression from G4C2 nucleotide repeats” we have softened our conclusions and also acknowledged that eIF2D could affect DPR expression through different mechanisms.

2. It is nice the authors now included patient-derived iPSC-neurons. But unfortunately they did not use this system to address the reviewer’s concerns (original comments 5 and 6). The authors mentioned they were unable to identify endogenous genes using CUG for translation initiation in *C.elegans*. They could test the CUG transcripts in the human iPSC-neurons as they already have the model. In addition, a puromycin incorporation assay could be used to test whether there is effect on global translation. Furthermore, the reviewer suggested to test the toxicity phenotype rescue effect of knocking known eIF2D in patient iPSC-neurons. Although GA level was not detectable, they could still show eIF2D

knockdown and measure the effects on neuron survival.

We thank the reviewer for these valuable suggestions, which we will follow up in a future manuscript. Following the editor's suggestion, we refrained from performing these experiments, and edited the text in a way that does not overstate the relevance of our experiments to human disease.

Reviewer #3 (Remarks to the Author):

It's lovely that the authors have so vigorously addressed concerns raised previously.

We are glad the reviewer finds that the new experiments have strengthened the manuscript.

All of my substantive points have been addressed. A few minor points remain:

Inaccurate/confusing sentence – “knockdown” does not have a role in poly-GA production
“Lastly, knockdown of eIF2A, an unconventional translation initiation factor used by cells under stress conditions^{17, 18, 19}, had a partial role in poly-GA production in HEK293 cells and neural cells of the chick embryo...”

The new sentence reads: “Lastly, knockdown of eIF2A, an unconventional translation initiation factor used by cells under stress conditions^{17, 18, 19}, partially affected poly-GA production in HEK293 cells and neural cells of the chick embryo...”

In Figure legend 1 and elsewhere, Act-1 should not be capitalized,
“The G4C2 repeat mRNA (C9orf72 intronic mRNA) and Act-1 mRNA were assessed by RT-PCR...”

This has been fixed now throughout the text to *act-1*. Thank you.

In Figure Legends 1 and 2, KasEx154 and KasEx155 should not be capitalized. Similarly, KasEx157 should not be capitalized in Figure Legend 4.

Fixed. Thank you.

Multiple comparisons corrections should be applied to Figure 6C and D. Reporting just SEM is not sufficient when so many comparisons are made. The reviewer suspects that this will not change the conclusions to be drawn.

In Figure 6, we have performed Turkey’s multiple comparisons test to calculate P values. This information is now included in the legend. In **Figure 6c-f**, we are showing the p values for two comparisons: a) eIF2D shRNA compared to pcDNA, and b) eIF2D shRNA compared to Control shRNA).

REVIEWER COMMENTS

Reviewer #1 (Remarks to the Author):

The authors have addressed my remaining concerns.

Reviewer #2 (Remarks to the Author):

Most of the concerns were addressed and the reviewer agree that the toxicity assay in iPSCs is not essential if not overstated. However, there is one major point remaining about the iPSC experiment:

It is understandable that the monocistronic and bicistronic reporters have different features. But it is hard to conclude which one is more close to endogenous RAN, as the repeat is located in the intron, not similar to either reporter. That is exactly why the reviewer asked it is essential to have endogenous GA quantification and show the effect of eIF2D in patient cells. This is the key experiment to support the findings of this manuscript. GA has been always shown to have highest level among all the three DPRs, and GR is usually the most difficult to detect. As the authors can detect GP and GR, it should be achievable to optimize the GA assay or try different GA antibodies. The authors guessed "one possibility is that poly-GA production depends on the number of G4C2 repeats, which is not known for the C9ORF72 iPSC lines we used". However, there is no evidence supporting this and the chance that both lines have very short repeats is low. The authors could also consider other cell types from patients to validate the effect of eIF2D on endogenous GA.

For the iPSC lines, as this is a critical experiment, and if this is the first time to publish the lines, the authors should characterize and show the approximate repeat length by southern blot, instead of guessing >24. For C9 #2 (ND10689): Coriell, Cat# ND10689, it is listed as B-lymphocyte, not iPSC line. If the authors converted to iPSC, data of the pluripotency markers should be included. The neuron differentiation markers of the lines should be included in supplement as well. If the lines have been published, references need to be added.

Lines 375-377, it is claimed that "consistent with this notion, high poly-GR levels are present in these HEK293 cells (Fig. 6m), and low poly-GR levels are present in our worm model (Supplementary Fig. 15) despite using the same detection method (ELISA)." This claim is not accurate because the poly-GP levels are also much lower in worm model than 293 cells. Actually the relative GR level compared to GP is higher in worm than 293. The different effect on GR in 293 cells cannot be due to the low expression level as the authors claimed. This explanation cannot hold.

RESPONSE TO REVIEWER COMMENTS

Reviewer #1 (Remarks to the Author):

The authors have addressed my remaining concerns.

We are glad all concerns of Reviewer #1 are addressed. Of note, all concerns of Reviewer #3 were addressed in the previous round of revision.

Reviewer #2 (Remarks to the Author):

Most of the concerns were addressed and the reviewer agree that the toxicity assay in iPSCs is not essential if not overstated.

We are glad most of Reviewer 2 concerns have been addressed. We thank the reviewer for their additional comments.

However, there is one major point remaining about the iPSC experiment: It is understandable that the monocistronic and bicistronic reporters have different features. But it is hard to conclude which one is more close to endogenous RAN, as the repeat is located in the intron, not similar to either reporter. That is exactly why the reviewer asked it is essential to have endogenous GA quantification and show the effect of eIF2D in patient cells. This is the key experiment to support the findings of this manuscript.

We agree with the reviewer. As mentioned in previous rounds of revision, we conducted this experiment in the context of collaborations with experts in the field of iPSC technology (Dr. Evangelos Kiskinis) and DPR detection (Fen-Biao Gao). Unfortunately, we were unable to detect poly-GA in motor neurons derived from these two specific iPSC lines of ALS patients (Supplementary Figure 26). Additional iPSC experiments from different ALS patients could be the focus of a future study. We therefore added the following sentence in Results *“Future experiments with additional iPSC lines from other ALS patients carrying long G4C2 could help clarify the role of human eIF2D in poly-GA production”*. Lastly, we have toned down the disease relevance throughout the manuscript, as the editor suggested.

GA has been always shown to have highest level among all the three DPRs, and GR is usually the most difficult to detect. As the authors can detect GP and GR, it should be achievable to optimize the GA assay or try different GA antibodies. The authors guessed “one possibility is that poly-GA production depends on the number of G4C2 repeats, which is not known for the C9ORF72 iPSC lines we used”. However, there is no evidence supporting this and the chance that both lines have very short repeats is low. The authors could also consider other cell types from patients to validate the effect of eIF2D on endogenous GA.

In the Decision Letter, the editor writes: *“Editorially we feel additional experiments to show endogenous polyGA-level in patient cell are not required”*. Following the editor’s suggestion, we refrained from performing additional iPSC experiments. Optimization of the GA assay with different antibodies is an open-ended experiment, which would delay at least by several months the publication of this manuscript. Moreover, we now mention in Results: *“Future experiments with additional iPSC lines from other ALS patients carrying long G4C2 repeats could help clarify the role of human eIF2D in poly-GA production”*.

Regarding the GA versus GR comparison, detection sensitivity depends on how good the GA and GR antibodies are (specificity, binding affinity etc.), and is not simply proportional to the relative abundance of GA and GR. It is possible in the 2 iPSC lines we used, GA level is below detection threshold.

Regarding number of G4C2 repeats, please see below.

For the iPSC lines, as this is a critical experiment, and if this is the first time to publish the lines, the

authors should characterize and show the approximate repeat length by southern blot, instead of guessing >24. For C9 #2 (ND10689): Coriell. Cat# ND10689, it is listed as B-lymphocyte, not iPSC line. If the authors converted to iPSC, data of the pluripotency markers should be included. The neuron differentiation markers of the lines should be included in supplement as well. If the lines have been published, references need to be added.

Three iPSC lines are used in our study. All three have been previously published and the relevant papers are now cited. In Materials and Methods (pages 33-34), we now provide additional information on pluripotency and G4C2 repeat length as shown below:

“ALS Patient iPSC Line Information:

Control (18A) line previously characterized for pluripotency (Boulting et al., Nat Biotechnology, 2011)⁶⁸. Harvard University; RRID:CVCL_8993. Female, 48 years old.

*C9 #1 (TALS9-11.2) line obtained from Target ALS. Cat# ND50008; RRID:CVCL_FA04. Female, 61 years old. Pluripotency certificate is provided in **Supplementary File 1**. This line is previously described in Ortega et al. Neuron, 2020⁶⁷, where repeat-primed PCR confirmed a large number of G4C2 repeats.*

C9 #2 (ND10689) iPSC line was reprogrammed by and received from the Ichida Lab (University of Southern California). B-lymphocytes were originally purchased from Coriell. Cat# ND10689; RRID:CVCL_BE81. Female, 51 years old. Pluripotency status and presence of a G4C2 repeat expansion were previously described in Shi et al. Nature Medicine, 2018⁶⁹. In the latter study, the G4C2 repeat size is approximately 14kb for this line. Repeat-primed PCR in Ortega et al. Neuron, 2020⁶⁷ also confirmed a large number of G4C2 repeats”.

Prompted by the reviewer, we have also conducted new experiments to confirm the expression of neuron differentiation markers (ChAT, ISLET1/2, MAP2) in motor neurons derived from all three iPSC lines (new **Supplementary Figure 25**). Lastly, previous work using the same differentiation protocol confirmed the presence of secreted acetylcholine (assessed by mass spectrometry) in iPSC-MNs generated using this protocol (Ziller et al., Cell Stem Cell, 2018, PMID: 29551301).

Lines 375-377, it is claimed that “consistent with this notion, high poly-GR levels are present in these HEK293 cells (Fig. 6m), and low poly-GR levels are present in our worm model (Supplementary Fig. 15) despite using the same detection method (ELISA).” This claim is not accurate because the poly-GP levels are also much lower in worm model than 293 cells. Actually the relative GR level compared to GP is higher in worm than 293. The different effect on GR in 293 cells cannot be due to the low expression level as the authors claimed. This explanation cannot hold.

We agree with the reviewer and recognize the confusion. The GP and GR comparison between HEK293 cells and the worm model is further complicated by the fact that detection sensitivity depends on how good these antibodies are (specificity, binding affinity etc.). Therefore, we decided to delete this sentence (lines 375-377) altogether from the manuscript.